


# Gas phase composition and secondary organic aerosol formation from gasoline direct injection vehicles investigated in a batch and flow reactor: effects of prototype gasoline particle filters.

Simone M. Pieber[1], Nivedita K. Kumar[1], Felix Klein[1], Pierre Comte[2], Deepika Bhattu[1], Josef Dommen[1], Emily A. Bruns[1], Dogushan Kilic[1], Imad El Haddad[1], Alejandro Keller[3], Jan Czerwinski[2], Norbert Heeb[4], Urs Baltensperger[1], Jay G. Slowik[1] and André S. H. Prévôt[1]

[1]Paul Scherrer Institute, Laboratory of Atmospheric Chemistry, CH-5232 Villigen, Switzerland
[2]Bern University of Applied Sciences, CH-2560 Nidau, Switzerland
[3]University of Applied Sciences Northwestern Switzerland; CH-5210 Windisch, Switzerland
[4]Empa Material Science and Technology; CH-8600 Dübendorf, Switzerland

*Correspondence to*: andre.prevot@psi.ch

**Abstract.** Gasoline direct injection (GDI) vehicles have recently been identified as a significant source of carbonaceous aerosol, of both primary and secondary origin. Here we investigated primary emissions and secondary organic aerosol (SOA) formation from GDI vehicle exhaust for multiple vehicles and driving test cycles, and novel GDI after-treatment systems. Emissions were characterized by proton transfer reaction time-of-flight mass spectrometry (gaseous non-methane organic compounds, NMOCs), aerosol mass spectrometry (sub-micron non-refractory particles), and light attenuation measurements (equivalent black carbon (eBC) determination using Aethalometer measurements) together with supporting instrumentation. We evaluated the effect of retrofitted prototype gasoline particle filters (GPFs) on primary eBC, organic aerosol (OA), NMOCs, as well as SOA formation. Two regulatory driving test cycles were investigated, and the importance of distinct phases within these cycles (e.g. cold engine start, hot engine start, high speed driving) to primary emissions and secondary products was evaluated. Atmospheric processing was simulated using both the PSI mobile smog chamber (SC) and the potential aerosol mass oxidation flow reactor (OFR). GPF retrofitting was found to greatly decrease primary particulate matter (PM) through removal of eBC, but showed limited partial removal of the minor POA fraction, and had no detectable effect on either NMOC emissions (absolute emission factors or relative composition) or SOA production. In all tests, overall primary and secondary PM and NMOC emissions were dominated by the engine cold start, i.e. before thermal activation of the catalytic after-treatment system. Differences were found in the bulk compositional properties of SOA produced by the OFR and the SC (O:C and H:C ratios), while the SOA yields agree within our uncertainties, with a tendency for lower SOA yields in SC experiments. A few aromatic compounds are found to dominate the NMOC emissions (primarily benzene, toluene, xylene isomers and C3-benzenes). A large fraction (> 0.5) of the SOA production was explained by those compounds, based on investigation of reacted NMOC mass and comparison with SOA yield curves of toluene, *o*-xylene and 1,2,4-trimethylbenzene determined in our OFR within this study. Remaining differences in the obtained SOA yields may



result from diverse reasons including aging conditions, unaccounted-for precursors and differences in SOA yields of aromatic hydrocarbons with different degrees of substitution, as well as experimental uncertainties in the assessment of particle and vapor wall losses.

**List of selected abbreviations/definitions**

| | | |
|---|---|---|
| 5 | AMS = | Aerosol mass spectrometer |
| | ArHC = | Aromatic hydrocarbons (including functionalized aromatic hydrocarbons) |
| | catGPF = | Catalytically active gasoline particle filter |
| | cE = | Cold-started EDC vehicle test |
| | cW = | Cold-started WLTC vehicle test |
| 10 | eBC = | Equivalent black carbon, as determined by Aethalometer measurements |
| | EDC = | (New) European Driving Cycle |
| | FID = | Flame ionization detector |
| | GDI = | Gasoline direct injection vehicle |
| | GPF = | Gasoline particle filter |
| 15 | hE = | Hot-started EDC vehicle test |
| | hW = | Hot-started WLTC vehicle test |
| | NMHC = | Non-methane hydrocarbons, i.e. gaseous organic compounds (hydrocarbons) as measured by FID |
| | NMOC = | Non-methane organic compounds, i.e. gaseous organic compounds as measured by PTR-ToF-MS |
| | OFR = | Oxidation flow reactor (a potential aerosol mass, PAM, reactor) |
| 20 | OFR-from-SC = | Also referred to as "batch OFR", OFR continuously sampling from a batch sample previously collected in the SC |
| | Online OFR = | OFR deployed online during a driving cycle, connected directly to diluted exhaust |
| | Ph 1 = | First phase of WLTC, Ph 1 (cW) refers to first phase of cold-started WLTC |
| | Ph 2-4 = | Second to fourth phase of WLTC, Ph 2-4 (cW) refers to the $2^{nd}$ to $4^{th}$ phase of cold-started WLTC, these |
| 25 | | are quasi hot engine conditions |
| | POA = | Primary organic aerosol |
| | PTR-ToF-MS = | Proton transfer reaction time-of-flight mass spectrometer |
| | SC = | Smog chamber |
| | SOA = | Secondary organic aerosol |
| 30 | WLTC = | World-wide light duty test cycle |



## 1 Introduction

Vehicular emissions are a significant source of air pollution in many urban areas ((Platt et al., 2014);(Zotter et al., 2014);(Bahreini et al., 2012);(Borbon et al., 2013);(May et al., 2014);(Worton et al., 2014);(Gentner et al., 2017)). Depending on vehicle fleet technology, emissions may include fine particulate matter (PM), which consists mainly of sub-

micron primary organic aerosol (POA) and black carbon (BC), as well as reactive gases such as nitrogen oxides ($NO_x$), and organic vapors. (Note that we refer to organic gas phase compounds as non-methane organic compounds, NMOCs. Measurements by proton transfer reaction mass spectrometry are also referred to as NMOCs herein. Instead, when referring to measurements by flame-ionization technique, we refer to organic gas phase compounds as non-methane hydrocarbons (NMHCs) instead.)

The NMOCs react in the atmosphere and can form secondary organic aerosol (SOA) (Hallquist et al., 2009). Human health is known to be impacted by $NO_x$ emissions, the associated ozone ($O_3$) formation, and especially by fine PM emitted from combustion processes. Fine PM penetrates deep into the human body and can damage lung tissue ((Kunzi et al., 2015)), and even damage the brain ((Calderon-Garciduenas and Villarreal-Rios, 2017)). Therefore, numerous strategies have been developed to decrease PM and $NO_x$ emissions from on-road vehicles, including optimization of engine settings and

implementation of after-treatment systems. Examples of such systems include oxidation catalysts that oxidize gas phase products of incomplete combustion (CO, NMOC) to $CO_2$, three-way-catalysts (TWC) (for gasoline on-road vehicles) and selective catalytic reduction (SCR) systems (for heavy duty diesel engines and large diesel passenger cars) to convert $NO_x$ emissions to $N_2$ and $O_2$, and (catalyzed) diesel particle filters (DPFs) to reduce primary PM emissions from diesel vehicles. Historically, diesel-fueled vehicles have been recognized as a significant source of primary PM, especially BC (Bond et al.,

2004). Accordingly, the use of older-generation diesel vehicles may be restricted in cities and modern diesel vehicles are subject to stringent primary PM limits. To achieve these limits, they are equipped with both diesel oxidation catalysts (DOCs) and DPFs, which have trapping efficiencies for refractory material of up to 99% (Gordon et al., 2013a). Due to the regulatory attention and the improved after-treatment systems, diesel PM emissions have been greatly reduced. However, $NO_x$ emissions from diesel vehicles have not been addressed as successfully and remain a topic of debate (e.g. (Barrett et al.,

2015);(Wang et al., 2016);(di Rattalma and Perotti, 2017)).

In contrast, modern gasoline light-duty vehicles have recently been engineered towards better fuel economy and reduced carbon dioxide ($CO_2$) emissions to satisfy regulations aimed at mitigating climate change (Karjalainen et al., 2014). However, recent research indicates that some of the methods used to attain these emission goals (including smaller engines, leaner combustion, and gasoline direct injection (GDI) systems mimicking the lower fuel consumption and decreased $CO_2$

emission factors of diesel vehicles) lead to an increase in the primary carbonaceous emissions (especially BC), among gasoline systems ((Karjalainen et al., 2014);(Zhu et al., 2016);(Platt et al., 2017);(Saliba et al., 2017)). Modern gasoline





light-duty vehicles have higher mass-based emission factors of these pollutants than do modern diesel vehicles (Platt et al., 2017), and additionally emit ammonia ($NH_3$) ((Heeb et al., 2006);(Suarez-Bertoa et al., 2014)) from the gasoline TWC. These emissions are released predominantly at engine start-up, when catalytic after-treatment systems are still cold, as well as during acceleration and deceleration ((Platt et al., 2017);(Gentner et al., 2017)). In the light of increasingly stringent

legislations for gasoline vehicles, automobile manufacturers have recently considered equipping modern gasoline light-duty vehicles with gasoline particulate filters (GPFs) to reduce primary PM emissions, and first results are promising ((Chan et al., 2014); (Demuynck, 2017)). Although GPFs are likely to be similarly effective as DPFs in reducing primary emissions (POA and BC), recent research indicates that the dominant fraction of the total PM from modern gasoline vehicles is secondary ((Platt et al., 2017);(Platt et al., 2013);(Nordin et al., 2013);(Gordon et al., 2014);(Gordon et al., 2013b);(Gentner

et al., 2017)). Dominant secondary species include SOA and ammonium nitrate ($NH_4NO_3$), which are formed by the reaction of emitted organic gases and $NO_x$ (in the presence of $NH_3$), respectively, with atmospheric oxidants such as hydroxyl (OH) radicals. The gaseous precursors leading to secondary aerosol are unlikely to be removed by GPF systems alone. Laboratory results of the GPF effect on NMOC emissions and the associated SOA formation are, however, missing so far.

Detailed investigations of SOA formation are typically performed in smog chambers (SC), where the emitted gases are

oxidized in batch-style experiments lasting several hours under close-to-tropospheric conditions. The poor time resolution of such experiments prevents efficient study of SOA formation as a function of driving conditions (e.g. engine load or catalyst temperature), which as noted above is a critical consideration for gasoline vehicles. In contrast, oxidation flow reactors (OFR) ((Kang et al., 2007);(Li et al., 2015)) are based on flow-through systems, allowing for investigation of SOA formation from time-varying emissions. They utilize higher-than-ambient oxidant concentrations to simulate hours to days of

atmospheric aging in only a few minutes of experimental time. However, few studies ((Karjalainen et al., 2015);(Bruns et al., 2015);(Tkacik et al., 2014);(Ortega et al., 2013)) have attempted the quantitative application of OFR systems to complex combustion emissions, and the differences between OFR and atmospheric oxidation conditions (e.g. high oxidant concentrations, short-wavelength light spectrum, and high wall surface-to-volume ratios) require further investigation ((Lambe et al., 2011);(Lambe et al., 2015);(Peng et al., 2015);(Peng et al., 2016);(Li et al., 2015);(Lambe et al., 2017);(Palm

et al., 2016)).

Despite numerous recent investigations of SOA formation from gasoline vehicle exhaust ((Platt et al., 2013);(Gordon et al., 2013b);(Gordon et al., 2014);(Nordin et al., 2013);(Platt et al., 2017)), the SOA formation processes and the role of relevant precursors and their SOA yields in simulated aging experiments remain a subject of debate. A wide range of ratios of secondary-to-primary OA (SOA/POA), and SOA yields (mass of SOA formed per organic vapors reacted) has been reported

despite standardized and repeatable testing procedures ((Jathar et al., 2014);(Gentner et al., 2017)). This is in part due to the high uncertainty related to experimental considerations, including NMOC levels, NO concentrations, OH exposure, particle and vapor wall losses and emissions sampling. Further, due to limitations of the previously applied techniques to study complex combustion emissions (such as offline GC-MS and GC-FID analysis of the total hydrocarbons (THC), or



quadrupole proton transfer reaction mass spectrometry (Q-PTR-MS) ((de Gouw et al., 2003);(Lindinger and Jordan, 1998)) allowing only for online monitoring of selected compounds having no significant interferences at the same integer $m/z$). Recently, (Zhao et al., 2016) suggested that the precursors are dominantly volatile organic compounds (VOCs) with a saturation mass concentration, $C^*$, above $10^6$ µg m$^{-3}$) and should hence allow for investigation with modern online

instrumentation, such as the high resolution time-of-flight PTR-MS (PTR-ToF-MS).

Here, we investigate primary NMOC, POA, eBC emissions and SOA formation from Euro 4 and 5 direct injection gasoline (GDI) vehicle exhaust, including vehicles retrofitted with prototype gasoline particle filter (GPFs). Vehicles were tested on a chassis dynamometer during a modern regulatory driving cycle (world-wide light duty test cycle, WLTC class-3) and older low-load European driving cycle (EDC), under cold- and hot-started engine conditions. SOA formation was investigated

through batch-style aging of collected emissions in (1) the PSI mobile smog chamber (SC) (Platt et al., 2013), and (2) the potential aerosol mass (PAM) oxidation flow reactor (OFR) (Bruns et al., 2015;Lambe et al., 2011;Lambe et al., 2015), used to study oxidation of batch emissions and single precursors. Further, time-resolved analysis of aged emissions during a driving cycle using the OFR for cold- and hot-started emissions was performed. Relevant SOA precursors were characterized using a PTR-ToF-MS, and the consumption of both individual and total NMOC was related to SOA formation.

The SOA mass and its bulk chemical composition were characterized by HR-ToF-AMS measurements.

## 2 Experimental

Two sets of experiments to study vehicle emissions were conducted (experiment set I in 2014, experiment set II in 2015). In addition, selected SOA precursor NMOCs (toluene, $o$-xylene, and 1,2,4-trimethylbenzene (TMB)) were injected into the OFR at different concentrations for comparison with the vehicle exhaust aging experiments (experiments conducted in

2016). In the following we describe vehicle testing (Section 2.1), non-regulatory and photochemistry experiments (including SC and OFR description and data correction) (Section 2.2), and instrumentation (including data processing) (Section 2.3).

### 2.1 Vehicle testing

Vehicles were operated on a chassis dynamometer with standard testing equipment at the "Laboratories for IC-Engines and

Exhaust Emission Control of the Berne University of Applied Sciences in Biel (Switzerland)", which includes a roller dynamometer (Schenck 500 GS60), a driver conductor system (Tornado, version 3.3), a CVS dilution system (Horiba CVS-9500T with Roots blower), and automatic air conditioning in the hall, (intake- and dilution air), which was maintained at a temperature of 20 - 30°C and an absolute humidity of 5.5-12.2 g kg$^{-1}$. The driving resistances of the test bench and the braking resistances were set according to legal prescriptions without elevation change. This equipment fulfills the

requirements of the Swiss and European exhaust gas legislation. The dilution ratio in the CVS-dilution tunnel is variable and




assessed by measurement of the $CO_2$-analysis. In addition, an FTIR instrument sampled undiluted exhaust at the tailpipe of the vehicles.

### 2.1.1 Vehicles, GPFs and fuels

The vehicles tested are listed in Table 1 and Table S1 (Supporting Information). In 2014, two vehicles were tested: a modern GDI Opel Insignia (denoted GDI1), as well as a Volvo V60 (denoted GDI4). GDI1 was investigated i) in standard configuration, and ii) equipped with a prototype gasoline particle filter (GPF (cordierite, porosity 50%, pore size 19 µm, 2000 cells per square inch)), installed at the muffler ("underfloor"). The GPF filtration quality is equivalent to the best available technology for DPFs (personal communication by the manufacturer). In 2015, two additional GDI vehicles

(denoted GDI2, and GDI3) in standard configuration were tested (no retrofitted after-treatment system). Tests with GDI4 (Volvo V60) in standard configuration were repeated in 2015; further, GDI4 was retrofitted with i) the previously tested GPF (as above: cordierite, porosity 50%, and pore size 19 µm, 2000 cells per square inch), as well as ii) a Pd/Rh catalytically coated GPF (catGPF) (installed at the muffler, underfloor, while keeping the original TWC in the original position). For the retrofitted catGPF, the primary purpose of the catalytically active coating is the constant self-cleaning of deposited

carbonaceous material on the particle filter (personal communication with manufacturer). In future applications, such catalytic coating on a GPF might replace the existing TWC in GDI vehicles, or specifically, the TWC can be replaced with a GPF carrying the TWC coating. All vehicles were fueled with gasoline from the Swiss market, RON 95, according to SN EN228. It contains 35% of aromatic hydrocarbons, below 1% alkenes, 5% methyl-tert-butyl-ether (MTBE) (in 2014, ~8% in 2015) added as anti-knocking agent,and < 0.5% ethanol, all on a volumetric basis.

### 2.1.2 Test cycles

We used dynamic driving cycles: the world-wide light duty test cycle (WLTC-class 3), and for reference the common but less representative EDC (European driving cycle) (speed profiles in Figure 1 and Figure S2). While the EDC is characterized by two phases (urban and extra-urban phase of highly repetitive characteristics) and lasts 20 min, the WLTC is characterized

by four phases at different speed levels (referred to as Phase (Ph) 1-4, i.e. low, medium, high, extra-high speed); it contains patterns of disruptive acceleration and deceleration and lasts 30 min. Engines were started either after a soaking time of at least 6 hours at the test temperature (typically between 20-25°C, referred to as "cold-started"), or after warming up the engine and after-treatment system by driving for 3 min at a steady-state speed of 80 km h$^{-1}$ ("hot-started"). Cycles are classified as cold-started WLTC (denoted cW), and hot-started WLTC (hW), cold-started EDC (cE), and hot-started EDC

(hE) throughout the manuscript.





### 2.2 Non-regulatory and photochemistry experiments

In parallel to the regulatory CVS sampling and tailpipe FTIR measurements, emissions were sampled using 1 or 2 Dekati ejector dilutors in series for characterization by non-regulatory equipment and photochemistry experiments. Figure 1 gives a scheme of the set-up, including our non-regulatory equipment, the SC and OFR. Sampling was performed similar to the
description in (Platt et al., 2017) and (Platt et al., 2013), which demonstrated good agreement between batch-sampled emissions and 1) online measurements of gaseous emission at the tailpipe (Platt et al., 2013) and 2) gravimetric PM samples from the CVS (Platt et al., 2017). Clean air to operate the sampling and dilution system, as well as the SC and OFR, was provided by a compressor (Atlas Copco SF 1 oil-free scroll compressor with 270 L container, Atlas Copco AG, Switzerland) combined with an air purifier (AADCO 250 series, AADCO Instruments, Inc., USA).

### 2.2.1 Experimental procedure

Experiments were conducted in three configurations: 1) time-resolved measurements of primary emissions and time-resolved aging in the OFR, 2) OFR photochemical aging from SC batch samples, 3) SC photochemical aging of SC batch sample. Experiments were conducted as follows. First, diluted emissions from the cold-started vehicle tests were sampled online
during the test bench driving cycle and characterized in real-time, either fresh ("primary"), or photo chemically aged in the OFR ("secondary"). In parallel, the emissions from the cold-started driving cycle were sampled into the SC for a later photochemical batch experiment, either over the full cycle (cW and cE), the first (Ph 1, cW) or the aggregated second through fourth phases (Ph 2-4, cW). Thereafter, a second vehicle test was performed, for which the vehicle was warmed up for 3 min at 80 km h$^{-1}$ steady state driving. The hot-started test emissions were sampled and characterized in real-time fresh
or aged ("OFR online"). No sampling of hot-started driving cycle emissions into the SC was performed. When both driving tests were completed, the previously sampled cold-started emissions were characterized and photochemically aged in the OFR by sampling the batch collection from the SC (OFR-from-SC sampling experiments, also referred to as "batch OFR" herein). Then, photochemical aging was performed in the SC. In addition to the vehicle tests (online OFR and OFR-from-SC experiments explained above), single NMOC (toluene, o-xylene, 1,2,4-TMB) species from a liquid injection system were
aged in the OFR as reference measurements.

At the start of each experiment the SC was filled to approximately two thirds full with humidified air, with the remaining volume available for sample injection. After sample injection, the chamber volume was filled up to its maximum with pure air, and the relative humidity (RH) was adjusted to 50%. To quantify OH exposure during the experiments, 1 μL of 9-times deuterated BuOH (BuOH-D9, purchased from Cambridge Isotope Laboratories) was added (Barmet et al., 2012). Once the
monitored emissions parameters and the BuOH-D9 signal stabilized and indicated a well-mixed chamber, primary emissions were characterized and sampled into the OFR for photochemical aging. The OFR was operated at different OH exposures determined by UV lamp intensity (denoted 100%, 70% and 50%), followed by a UV off (OFR dark) period. Once OFR-





from-SC sampling was completed, $O_3$ was injected into the SC to titrate NO to $NO_2$. HONO, used as an OH precursor, was injected continuously for the remainder of the experiment and photochemistry was initiated by illuminating the SC with the UV lights for a period of 2 hours. The temperature around the SC was kept approximately at 25°C, but reached up to 30°C with UV lights switched on. The OFR likely also has slightly higher than ambient temperatures close to the UV sources, due

to heating from the lamps.

### 2.2.2 PSI mobile smog chamber (SC)

The PSI mobile SC (Platt et al., 2013) is an approximately 12 $m^3$, 125 μm thick collapsible Teflon bag (DuPont Teflon fluorocarbon film (FEP), type 500A, Foiltec GmbH, Germany) suspended from a mobile aluminum frame (2.3×2×2.5 m,

L×W×H) with a battery of 40×100W UV lights (Cleo Performance solarium lamps, Philips). It is equipped with an injection system for purified air, water vapor, and gases ($O_3$, NO, $NO_2$, $SO_2$, propene ($C_3H_6$)). OH radicals used as the primary oxidant are generated by photolysis of HONO injected continuously into the chamber and generated as described in (Platt et al., 2013);(Taira and Kanda, 1990)). The SC was cleaned prior to each experiment by filling with humidified air and $O_3$ and irradiating with UV light for at least 1 h, followed by flushing with dry, pure air for at least 10 h. Background measurements

of the clean chamber were conducted prior to each experiment. Control experiments were conducted regularly in the SC to estimate the contribution of the SC background to SOA formation.

### 2.2.3 Oxidation flow reactor (OFR)

Experiments herein utilize the potential aerosol mass (PAM) OFR, of which several different configurations currently exist

((Bruns et al., 2015);(Lambe et al., 2011);(Kang et al., 2007);(Lambe et al., 2015);(Lambe and Jimenez)). Our OFR was previously described by (Bruns et al., 2015) and consists of a 0.015 $m^3$, cylindrical glass chamber (0.46 m length, 0.22 m diameter) containing two low pressure mercury UV lamps, each with discrete emission lines at 185 and 254 nm (BHK Inc.) ((Li et al., 2015);(Peng et al., 2015);(Peng et al., 2016)). The lamps are cooled by a constant flow of air. The incoming reactant flow is mixed radially dispersed by a perforated mesh screen at the inlet flange. In our experiments, the flow through

the OFR was regulated by the flow pulled by instruments and pumps behind the reactor, and was set to ~8-9 L $min^{-1}$, corresponding to a plug flow residence time of 90-100 s. A small fraction of the total flow (0.5-1 L $min^{-1}$) was sampled behind a second perforated mesh (often termed "ring-flow") and discarded. The OFR was equipped with an injection system for water vapor (a Nafion humidifier) and organic compounds (BuOH-D9 as an OH tracer, and toluene, *o*-xylene and 1,2,4-TMB purchased from Sigma-Aldrich (p.a.) for single precursor tests (Figure S1, Supporting Information). During "online"

(time-resolved) operation of the OFR, the diluted exhaust (either 1 or 2 ejector dilutors) was mixed with humidified air up to 50% of the total flow through the reactor, leading to an additional dilution of up to a factor 2. When sampling from the SC



(OFR-from-SC experiments) instead, no separate addition of water vapor or BuOH-D9 was required. OH radicals in the OFR are produced by photolysis of water vapor ($H_2O$) at 185 nm, or by production of atomic oxygen in excited state $O(^1D)$ from photolysis of ozone ($O_3$) at 254 nm, which can react with $H_2O$ to form OH. $O_3$ itself is produced by reaction of atomic oxygen in ground state, $O(^3P)$, with $O_2$. $O(^3P)$ itself is formed by photolysis of $O_2$ at 185 nm. Lamp power can be regulated

between 0 and 100%, with lower intensities lowering both, $O_3$ and OH production. The ratio of $OH/O_3$ remained relatively constant at our test points (($1.4\text{-}2.6)\times10^{-5}$ at 100%, ($1.9\text{-}3.0)\times10^{-5}$ at 70%, and ($1.7\text{-}2.6)\times10^{-5}$ at 50%). The OFR was cleaned prior to each experiment by flushing it with humidified, pure air, while keeping the UV lights on for at least 10 min.

### 2.2.4 Particle losses in SC and OFR

Loss of particulate (and gaseous) material to reactor walls are one of the largest uncertainties in simulations of atmospheric aging and are variable between systems ((Zhang et al., 2014);(Lambe et al., 2011);(McMurry and Grosjean, 1985)). Minimizing the surface area to volume ratios (i.e building bigger chambers), using inert wall materials (such as Teflon), and attempting to isolate the sampled flow from the walls (as in our OFR), help to reduce losses. The main losses of particles are due to (1) diffusion, (2) electrostatic deposition and (3) gravitational settling. Wall losses of particles in the SC were

accounted for using the method described in (Weitkamp et al., 2007) and (Hildebrandt et al., 2009). The suspended OA concentration, $C_{OA,suspended}$, was corrected to yield $C_{OA,wlc}$ according to Eq. (1) (Hildebrandt et al., 2009), for vapor losses see discussion in section 2.2.5). Particulate wall-loss rates, $k_w$, were determined as the exponential decay constant from an exponential fit of the decrease in eBC mass (determined from optical absorption at $\lambda$=950 nm) with time. When eBC was below the detection limit (e.g., for experiments with vehicles equipped with GPF), an average of the decay constants

determined from the other experiments was applied ($k_w$=5.6x$10^{-5}$ s$^{-1}$). Diffusional losses of particles vary with particle size (McMurry and Grosjean, 1985). Our correction implicitly assumes internally mixed OA/eBC particles, and does not account separately for size-dependent effects.

$$C_{OA,wlc}(t) = C_{OA,suspended}(t) + \int_0^t k_w * C_{OA,suspended}(t) * dt \qquad (1)$$

A comparison of eBC mass before and after the OFR indicated no significant loss, and consequently no correction for particle losses was applied to OFR data. Additionally, particle wall losses in the OFR have been quantified previously by (Lambe et al., 2011), who reported above 80% transmission efficiency through the OFR for particles of mobility diameter ($d_m$) > 150 nm (Lambe et al., 2011). The particles measured behind the OFR in the current study had a vacuum aerodynamic

diameter ($d_{va}$) of approximately 200 - 400 nm based on HR-ToF-AMS (DeCarlo et al., 2006) measurements (size




distributions are provided in Figure S9, Supporting Information), corresponding to a $d_m > 150$ nm when assuming spherical particles and an OA density of 1.2 g cm⁻³. Therefore a transmission efficiency of at least 80% can be assumed.

### 2.2.5 Vapor losses in SC and OFR (walls and non-OH processes)

Low-volatility vapors (especially semi-volatile (SVOC), and low volatility organic compounds (LVOC) are prone to losses on clean reactor walls and deposited OA particles, which compete with partitioning to suspended OA particles. Numerous publications discuss potential vapor wall losses in these systems (e.g. (Zhang et al., 2014);(Hildebrandt et al., 2009)), highlighting that these losses may be important for under-predicted SOA yield estimates. However, a robust strategy for their determination and correction remains challenging. In previous work, we estimated that vapor wall losses may cause SOA

yields to be underestimated for the SC used herein (assessed based on gasoline vehicle exhaust SOA, see (Platt et al., 2017) by roughly a factor 1.5-2 for our experimental conditions), comparable to other systems (Zhang et al., 2014). As such correction is not widely used, it is not applied herein to facilitate comparison with previously published data.

(Palm et al., 2016) estimated LVOC losses in the OFR due to vapor wall losses, insufficient residence time for partitioning to the particle phase (before vapors exit the OFR), and fragmentation due to multiple OH reactions prior to vapor condensation

on suspended OA. We tested the loss rate of vapors in our system based on this model during batch mode operation of the OFR. Given the high SOA concentration and hence seed surface (1000 - 5000 nm² cm⁻³ based on the SMPS size distribution), 80% of the formed LVOC was calculated to partition to the pre-existing OA mass. Presented data are not corrected for this potential underestimation of SOA yields (which would increase them by a factor of 1.25), for comparison with previous data.

Non-OH reaction processes in the OFR can be another pathway by which primary vapors can be lost. These processes have been paramterized by (Peng et al., 2016) as a function of residence time, photon-flux or $O_3$ measurements, water vapor availability, and external OH reactivity ($OHR_{ext}$) defined as the product of the available OH-reactive material and its respective OH rate constant. In addition to OH, also photons (185 nm, 254 nm), and oxygen allotropes (excited oxygen atoms ($O(^1D)$), ground state oxygen atoms ($O(^3P)$), ozone ($O_3$)) were identified as relevant loss processes to precursor

molecules, dependent on their chemical identity. To estimate the influence of these parameters we applied the model of (Peng et al., 2016) to the current study as discussed in the Supporting Information. For OFR-from-SC experiments we predict an influence of non-OH loss processes (up to 25% UV loss for benzene and 10% for toluene). For time-resolved OFR experiments, the model predicted significant losses at low dilution ratios (1 ejector dilutor, which applies to experiments performed in 2014), but smaller influences for experiments conducted with double dilution (2 ejector dilutors, which applies

to experiments performed in 2015). Time-resolved OFR experiments from 2014 were also impacted by OH suppression and $NO_x$ levels, for which reason they were not used quantitatively within this publication (see discussion in SI and Section 3.3).





### 2.2.6 SOA yields

SOA yields analysis is based on SC and OFR-from-SC experiments with GDI1-3. An *effective* SOA yield ($Y_e$), was calculated as the ratio of the SOA mass to the reacted SOA-forming mass, $\Delta NMOC_{reacted}$ (in $\Delta \mu g\ m^{-3}$, Eq. (2)). The *effective* SOA yield provides the SOA mass formed via the identified SOA precursors (*i*), neglecting non-reactive and non-SOA

forming precursors and assuming that all relevant SOA precursors are measured. Identified SOA precursors *i* refers to the 8 dominant aromatic hydrocarbons presented in Figure 4d.

$$Ye\ =\frac{\Delta SOA}{\sum_i \Delta NMOC_{i,reacted}} \hspace{4cm} (2)$$

The SOA-forming precursor mass was determined by identifying and quantifying relevant SOA-precursor NMOCs by PTR-ToF-MS. SOA yields are presented as a function of the suspended (i.e. non particle wall loss corrected) organic aerosol mass (POA+SOA), for consistency with the wall loss correction method described above (i.e. neglecting vapor-wall interactions). In practice, this has little effect on the obtained SOA yield curves, as particle wall losses were limited due to the short SC experiment (lasting 2 hours). The yield can be calculated for each point in time since initiation of photochemistry in the SC

or in the OFR sampling from the SC (OFR-from-SC). This results in a yield as a function of OH exposure, and also as a function of suspended OA.

### 2.2.7 OH exposure estimation

The time-integrated OH exposure (molec cm$^{-3}$ s), defined as the integrated OH concentration ([OH]) over the reaction time

(*t*) is calculated from the decay of BuOH-D9 as described by (Barmet et al., 2012) for both the OFR and SC experiments. The obtained OH exposure can be related to an approximate ambient aging time by assuming a mean atmospheric [OH] (e.g. $1 \times 10^6$ molec cm$^{-3}$ for a global 24h daytime average, or applying a 12h average of $2 \times 10^6$ molec cm$^{-3}$, taking into account that OH radicals are predominantly available during daytime (average value taken from (Finlayson-Pitts and Pitts))). OH concentration and exposure are also predicted by (Peng et al., 2016) as presented in the SI.

### 2.3. Instrumentation and data processing

### 2.3.1 Test bench instrumentation

Gaseous components were monitored with an exhaust gas measuring system Horiba MEXA-9400H, including measurements of CO and $CO_2$ by infrared analyzers (IR), hydrocarbons by flame ionization detector (FID) for total hydrocarbon (THC) and





non-methane hydrocarbon (NMHC) measurements, NO/$NO_x$ with a chemo luminescence analyzer (CLA) which was not heated and applicable only for diluted gas, and $O_2$ (Magnos). The dilution ratio in the CVS-dilution tunnel is variable and was controlled by means of the $CO_2$-analysis. Non-legislated gaseous emission components were analyzed by FTIR (Fourier Transform Infrared Spectrometer, AVL SESAM) at the exhaust tailpipe, offering time-resolved measurement of approx. 30

emission components, including NO, $NO_2$, $NO_x$, $NH_3$, $N_2O$, HCN, HNCO, HCHO. Number concentration of non-volatile particles were measured with condensation particle counters (CPC) behind a thermo-conditioner heating the sample to 300°C.

### 2.3.2 Non-regulatory equipment for photochemistry experiments

Along with a suite of basic gas-phase monitors for measurements of $CO_2$, CO and $CH_4$ (CRDS, Picarro), THC, $CH_4$ and NMHC (FID, Horiba), NO, $NO_2$, $O_3$ and particle-phase instruments (CPC and SMPS for particle number and size measurements, and 7-wavelength aethalometers for eBC determination (Drinovec, 2015) (Aerosol d.o.o), high resolution time-of-flight mass spectrometers were applied to investigate the chemical composition of the fresh and aged exhaust (instruments are listed Table S2-S3, Supporting Information).

### 2.3.3 PTR-ToF-MS

A high resolution proton transfer reaction time-of-flight mass spectrometer ((Jordan et al., 2009);(Graus et al., 2010)) (PTR-ToF-MS), (PTR-TOF-8000, Ionicon Analytik Ges.m.b.H., Innsbruck, Austria), was used to study the chemical composition of the gaseous non-methane organic compounds (NMOC) in fresh and aged emissions. The PTR-ToF-MS uses hydronium

ions ($H_3O^+$) as the primary reagent to protonate gaseous organic molecules having a proton affinity higher than that of water (691 kJ mol$^{-1}$, (Gueneron et al., 2015)). Water clusters $(H_3O)(H_2O)^+$ were below 5% of the $H_3O^+$ ion and were not considered for the calculations. Detected compounds included most aromatic hydrocarbons of interest, alkanes (above $C_{10}$) and alkenes (above $C_2$), as well as oxygenated compounds (aldehydes, ketones and carboxylic acids) and thus includes many organic molecules expected in GDI vehicle exhaust ((Gueneron et al., 2015);(Schauer et al., 2002)). The sample was introduced into

a drift tube and mixed with $H_3O^+$ ions produced from water vapor in a hollow cathode ion source, leading to protonation of the analyte gas molecules, which were detected by time-of-flight mass spectrometry ((Jordan et al., 2009);(Graus et al., 2010)). For experimental set I (2014), the PTR-ToF-MS operated with a drift voltage of 545 V, a chamber temperature of 90 °C, and a drift pressure of 2.2 mbar resulting in a reduced electric field (*E/N*) of about 140 Td. In experimental set II (2015) and for single precursor experiments (2016), the drift voltage was 545 V, the drift tube temperature was 60 °C and the drift

pressure was 2.1-2.2 mbar resulting in an *E/N* of 130 Td. The mass resolution, as well as the mass accuracy and the relative transmission efficiency ((De Gouw and Warneke, 2007);(Müller et al., 2014)), were routinely verified using a 12-compound





gas standard (Carbagas, protonated integer *m/z* 45 to 181, containing alcohols, carbonyls, alkenes, aromatic hydrocarbons (ArHC) and terpenes). Additionally, we use an internal calibrant (diiodobenzene, $C_6H_4I_2$) for mass calibration (protonated integer *m/z* 331), to support mass calibration at higher *m/z*. Data were analyzed using the Tofware post-processing software (version 2.4.2, TOFWERK AG, Thun, Switzerland; PTR module as distributed by Ionicon Analytik GmbH, Innsbruck,

Austria), running in the Igor Pro 6.3 environment (Wavemetrics Inc., Lake Oswego, OR, U.S.A.). In the absence of fragmentation (discussed below), ions are observed at mass-to-charge (*m/z*) ratios corresponding to the neutral parent molecule shifted by the mass of one proton (denoted $[NMOC+H]^+$). The exact mass was used to determine the elemental composition of an ion and was combined with previous reports of compounds identified in combustion emissions ((Schauer et al., 2002);(Schauer et al., 1999);(Gueneron et al., 2015);(Erickson et al., 2014)) to propose likely molecular structures.

NMOC concentrations were derived from the $H_3O^+$ normalized ion signal of $[NMOC+H]^+$, the appropriate reaction rate ((Cappellin et al., 2012);(Cappellin et al., 2010)) towards $H_3O^+$ ($k_{H3O+}$) and the residence time in the drift tube, following standard procedures. While ideally the molecular sum formula can be approximated by the exact mass of $[NMOC+H]^+$ (see discussion on fragmentation below), isomers, such as e.g. *o*-, *p*-, *m*-xylenes and ethylbenzene, cannot be resolved and the selection of $k_{H3O+}$ may thus be somewhat uncertain. If available and applicable based on identification, we used the exact

reaction rate reported in literature ((Cappellin et al., 2012);(Cappellin et al., 2010)), otherwise we assumed the collisional rate constant of $2 \times 10^{-9}$ cm$^3$ s$^{-1}$. Although protonation with $H_3O^+$ is considered a soft ionization technique, fragmentation occurs for certain compounds including aldehydes, alcohols, alkanes, alkenes and substituted aromatics, with the non-oxygen-containing species being of particular importance for the current study ((Gueneron et al., 2015);(Erickson et al., 2014);(Buhr et al., 2002)). Fragments were observed here, but they constituted only a small fraction of the total signal in our

analysis (see Figure 4), and therefore no corrections were applied. Fragments that could not be attributed to an $[NMOC+H]^+$ parent are reported as "structurally unassigned".

### 2.3.4 HR-ToF-AMS

Quantitative, size-resolved mass spectra of the non-refractory sub-micron particle composition were provided by use of a

high resolution time-of-flight aerosol mass spectrometer (HR-ToF-AMS, Aerodyne, (DeCarlo et al., 2006)). Particles are continuously sampled into the HR-ToF-AMS through a PM$_1$ aerodynamic lens and focused onto a heated porous tungsten vaporizer ($T_{vap}$=600 °C) in high vacuum ($10^{-5}$ Pa). The non-refractory particle components flash-vaporize and the resulting gas is ionized by electron ionization (EI, 70 eV), and then classified by a time-of-flight mass spectrometer (ToF-MS). The particle beam is alternately blocked ("*closed*") and unblocked ("*open*"), and all data presented herein are *open minus closed*

signals derived from high resolution analysis fitting procedures (SQUIRREL1.51H, PIKA 1.10H), running in the Igor Pro 6.3 environment (Wavemetrics Inc., Lake Oswego, OR, U.S.A.). Following standard procedures (Canagaratna et al., 2007), the instrument ionization efficiency (IE) and particle size measurement were calibrated using size-selected NH$_4$NO$_3$ particles





and polystyrene latex spheres (PSLs), respectively. A relative ionization efficiency (RIE) of 1.4 for organic material and a collection efficiency (CE) of 1 was applied to the data. No corrections for lens transmission were performed. HR-ToF-AMS data were corrected for background gas-phase $CO_2$ in the emissions by subtracting a $CO_2$-signal measured in a particle-free sample. The interaction of inorganic salts with pre-deposited carbon on the tungsten vaporizer can lead to the generation of

$CO_2^+$ signal in the *open minus closed* HR-ToF-AMS mass spectra (Pieber et al., 2016). Here photochemical aging of the exhaust resulted in significant $NH_4NO_3$ formation, reaching $NO_3$/OA ratios of 5. A $CO_2^+$ signal at 3.5% to $NO_3$ was determined by calibration (see Figure S3, Supporting Information) and corrected according to (Pieber et al., 2016).

### 3 Results and discussion

### 3.1 Pollutants as function of vehicle technology and driving cycle

Figure 2 summarizes emission factors (EFs) of pollutants across all vehicles and conditions tested. Investigation of THC, NMHC and gravimetric PM of time-resolved emissions for cold- and hot-started WLTC and EDC tests using GDI1-3 demonstrated significant THC and NMHC emissions during cold-engine tests, with emission factors (EFs) reduced by a factor of 90 during hot-started cycles (Figure 2, panel a and c). Median NMHC EFs were 1132 mg $kg_{fuel}^{-1}$ (cW) and 12.9 mg

$kg_{fuel}^{-1}$ (hW). EFs from cold-started WLTC (cW) for GDI1-3 were dominated by Ph 1 (cW, 4663 mg $kg_{fuel}^{-1}$), which exceeded all other phases of cold- and hot-started WLTC by 2 to 4 orders of magnitude (median, GDI1-3). GDI4 had lower total emissions during cold-started cycles compared to other vehicles (~factor 3 lower, median NMHC EF (cW): 434 mg $kg_{fuel}^{-1}$) and a smaller difference between cold- and hot-started cycles (GDI4 cW NMHC is only 8 times higher than hW, rather than 90 times as for GDI1-3; median NMHC EF for hW (GDI4): 55.7 mg $kg_{fuel}^{-1}$). When looking at individual phases

of the driving cycle (e.g. comparing Ph 1 of cW and hW vs. Ph 2-4 of cW and hW) for different vehicles (GDI1-3 vs. GDI4), we find that while Ph 1 (cW) NMHC emissions for GDI4 are significantly lower compared to GDI1-3 (by a factor 3), NMHC EF for GDI4 during all other phases appeared higher than those of GDI1-3 (factor 2-30, with the biggest difference found for Ph 2-4 (hW). The corresponding median data are 4663, 0.1, 23.8, 1.6 (for GDI1-3, Ph 1 (cW), Ph 1 (hW), Ph 2-4 (cW) and Ph 2-4 (hW) respectively), and 1507, 2.2, 56.8, 41.1 mg $kg_{fuel}^{-1}$ (for GDI4).

Lower cold start emissions of GDI4 compared to other vehicles may be explained by differences in the catalytic after-treatment system, the location of the catalyst as well as reduced cold start enrichment. In terms of NMHC/THC EFs, GDI4 can be considered in line with Euro 6 vehicles, for which regulation also focuses on the reduction of the cold-start HC emissions. No influence of GPF installation on the NMHC EFs was observed for either GDI1 or 4 (GDI2 and 3 were not tested), as further discussed below (Section 3.2). Primary PM emissions are less affected by the differences between cold-

and hot-started cycles and among vehicles (Figure 2a). The largest difference was induced by the application of GPFs as discussed further below (Section 3.2). The total PM emitted by vehicles in standard configuration is dominated by eBC



rather than POA (Figure 2b), and the low POA-to-eBC ratio is similar to diesel engines not equipped with DPFs, as also found by (Saliba et al., 2017). PM measured in the batch samples (sum of eBC and POA, Figure 2b) compares generally well with the gravimetric PM analysis of filters sampled from the CVS (Figure 2a). Selective sampling of phases of the cold-started cW into the SC (Figure 2d) and time-resolved measurements (Figure 3) indicate that significant eBC is emitted

during cold-engine start-up (Ph 1 cW). Equivalent BC emissions are however not as strongly reduced during hot-engine (Ph 2-4 from cold-started cycle) emissions, and during hot-started cycles (hW, Figure 2a for total PM (CVS), Figure 3 for eBC) as e.g. the NMHC EFs under hot engine conditions.

Emissions of all cold-started vehicles, technologies and driving tests showed significant SOA formation upon photochemical oxidation (Figure 2b), in line with previous findings ((Platt et al., 2017);(Gordon et al., 2014);(Nordin et al., 2013)). This is

consistent with the above observation that NMHC and NMOC emissions (determined by the PTR-ToF-MS, see Figure 2d) are greatly elevated during cold-started cycles. The SOA production in mg kg$^{-1}_{fuel}$ lies within the range of previous aggregated data ((Jathar et al., 2014);(Platt et al., 2017)) (median 60, range ~10-400 mg kg$^{-1}_{fuel}$) (Jathar et al., 2014) for the SC experiments with vehicles in standard configuration (as well as equipped with GPF, discussed in Section 3.2). Similar to the observations for NMHC EFs, SOA production factors for GDI4 (median: 12 mg kg$^{-1}_{fuel}$) are around a factor 20 lower

compared to average SOA production factors for GDI1-3 (Figure 2b) (median: 222 mg kg$^{-1}_{fuel}$).

OFR experiments typically result in higher SOA production than the SC experiments, which can be explained by higher OH exposure, leading to reaction of more precursor mass and higher OA loadings and hence an influence of partitioning effects ((Pankow, 1994);(Donahue et al., 2006)), as discussed later (see SOA yield curve analysis, Figure 6). Photo-chemical aging of Ph 2-4, sampled into the SC and OFR-from-SC from a cold-started driving test, showed significantly lower SOA

production factors (Figure 2d), in analogy to the lower NMHC and NMOC emission factors for hot engine conditions, which are discussed in detail later.

### 3.2 Effect of gasoline particle filters (GPF) on pollutants

Gravimetric PM and eBC were greatly reduced by the retrofitted GPFs (GPF tested on GDI1 and GDI4; with GPF

performance apparently compromised on GDI4, potentially due to aging of the filter; catGPF tested on GDI4 also performed poorly). While the retrofitted GPFs efficiently remove the non-volatile eBC and thus significantly reduce total primary PM (Figure 2 panel a,b,d), the effect on POA is more complex. POA has a wide range of volatilities and may thus encounter a particle filter in either vapor or particle phase. Thus GPFs can only efficiently remove the low volatility POA fraction, while more volatile POA passes through the filter as vapor and will condense when the exhaust is cooled in the ambient air. Within

experimental uncertainty, retrofitted GPFs (including catGPF behind the standard TWC) did not affect the POA fraction. Further, GPFs did not affect FID-based NMHC (Figure 2a) and PTR-ToF-MS-based NMOC EFs (Figure 2d), or NMOC composition as discussed later in Figure 4. The retrofitted GPFs did neither reduce the produced SOA mass (Figure 2, panel



b,d, and Figure S14). SOA reduction requires additional after-treatments to remove NMHCs/NMOCs, such as engine or catalyst pre-heating, indicated by significantly lowered SOA formation during Ph 2-4 (when engine and after-treatment systems are already hot), compared to Ph 1 emissions (when engine/after-treatment systems are cold), as discussed above in Section 3.1.

### 3.3 Time-resolved SOA formation (OFR)

Investigation of CVS and batch sampling of the individual phases of cold-started WLTC indicates the highest emission of SOA precursors and SOA formation from cold-started Ph 1 (cW), consistent with the discussion in Section 3.1. This is confirmed by time-resolved SOA profiles from aging of the emissions in the OFR online during the driving cycles. Figure 3

shows the time-resolved aged emissions for cold- and hot-started WLTC tests using GDI1 (standard configuration). The emissions are exposed to photochemistry in the OFR, with UV intensity at 100%. OA and nitrate (denoted $NO_3$) are monitored behind the OFR by the HR-ToF-AMS; for the cold-started cycle, the POA signal measured during a separate experiment (with OFR UV off) is shown for reference. The large difference between the OA and POA traces indicates that the observed OA is predominantly SOA. During the cold-started cycle, significant SOA formation is observed by the HR-

ToF-AMS during Ph 1 (i.e. start and low speed) and to a lesser extent during Ph 2-4 (simulated highway driving). The peak at engine start is observed during all cold-start vehicle tests, regardless of vehicle, driving cycle or presence/absence of GPFs, while the small peak at the end of high-speed/extra urban driving is finished appears inconsistently. The latter is related to a delay of the OFR signal by the residence time in the reactor. The cold-start SOA signal correlates with THC measurements at the OFR inlet (Figure S5-S8, Supporting Information), and is not evident during the hot-started cycle.

These trends are consistent with the regulatory test bench measurements described above and EFs calculated from batch samples in the SC. The duration of the SOA peak observed at the engine start is likely artificially increased by OFR residence/response timescales and reflects the first few seconds to minutes, prior to catalyst light-off, rather than representing consistently high emissions throughout Ph 1 (Figure S5-S8, Supporting Information). Supporting this explanation, the hot-started cycle (in which the catalyst operates efficiently from the beginning of the test) does not exhibit any significant

emission of NMHC (Figure 2c), and leads to very little SOA formation also when investigated online. Hence, the cold-start emissions dominate the total GDI SOA burden, and are selected below for investigation of relevant SOA precursors, SOA potential and yields in OFR-from-SC and SC photochemistry experiments. Time-resolved SOA data from 2014 are not used quantitatively herein, due to instabilities with the OH exposure throughout the driving cycle (lower OH exposure during high emissions period as well as impact by photolysis and competing non-OH processes (i.e. high external OH reactivity, see SI

Eq. S2 and Figure S11-S12, ((Peng et al., 2015);(Peng et al., 2016); (Li et al., 2015)) and potential impact of NO on the oxidation regime (high vs. low NO levels, $NO_3$ radical formation, see (Peng and Jimenez, 2017)). This is caused by the low dilution applied (1 ejector dilutor, 1:8, and additional 1:2 at OFR entrance) in 2014. For the experiments conducted in 2015,



such experimental artefacts were reduced by the use of a higher dilution ratio (2 ejector dilutors in series, each 1:8 and additional 1:2 at OFR entrance). Time-resolved data from 2015 collected with GDI4 were integrated to derived EFs labelled "Online, OFR100%" in section 3.1, 3.2 and 3.4 (Figure 2b, Figure 4) and are comparable to data derived from GDI4 SC experiments. While we don't rely on an absolute quantitative use of our 2014 data from time-resolved measurements, the

relative profile (SOA dominated by cold start as presented in Figure 3) holds true regardless of those effects and is confirmed in the 2015 data set (Figure S14) showing the same trends.

### 3.4 Primary NMOC composition investigated by PTR-ToF-MS

Figure 4a shows the average NMOC mass spectrum obtained by the PTR-ToF-MS for exhaust from vehicle GDI1 over a full cold-started WLTC. The relative NMOC composition over all test conditions (driving cycles and phases, vehicle

configuration) is given in Figure 4b. A detailed description is provided later. In summary, while gasoline as a fuel is mainly composed of aliphatic compounds and ArHC having between 7 and 10 carbons (making up roughly 35% of the fuel volume), the exhaust mass spectral composition from cold-started driving tests appears to be dominated by surviving fuel additives (ArHC, and methyl-tert-butyl-ether (MTBE)), together with newly formed ArHC and short chain aliphatics, which which are incomplete combustion products. The composition depends strongly on the driving cycle phase, with ArHC dominating the

emissions in Ph 1 (cW) and the full cycles (cW, cE), while constituting a smaller fraction of Ph 2-4 (cW). As discussed above (Figure 2), NMHC EFs during Ph 1 (cW) are far higher than Ph 2-4 (cW), with the resulting effect that the emissions composition for the full WLTC closely resemble those of Ph 1 (i.e., dominated by ArHC) also from a chemical composition perspective. (Note that the NMOC concentrations for Ph 2-4 (cW) are close to our background measurements (signal not significantly different from 3 standard deviations of the background measurement)). We showed above that GPF installation

does not reduce NMHC EFs (Figure 2, Figure 4c); in addition, it has no obvious influence on the gaseous NMOC composition (Figure 4b).

In detail, the mass spectrum and relative composition for full WLTC and Ph 1 (cW) experiments is dominated by a small number of ArHC ions, specifically: benzene ($[C_6H_6+H]^+$, integer $m/z$ 79, denoted BENZ), toluene ($[C_7H_8+H]^+$, $m/z$ 93, denoted TOL), $o$-/$m$-/$p$-xylene or ethylbenzene ($[C_8H_{10}+H]^+$, $m/z$ 107, denoted XYL/EBENZ) as well as $C_3$-benzenes

($C_9H_{12}+H^+$, $m/z$ 121, denoted C3BENZ). Reaction rates of the above compounds are shown in Table 2. The most important additional aromatic HC peaks in the spectra correspond to $C_4$-benzenes ($[C_{10}H_{14}+H]^+$, $m/z$ 135, denoted C4BENZ), naphthalene ($[C_{10}H_8+H]^+$, $m/z$ 129, denoted NAPH), styrene ($[C_8H_8+H]^+$, $m/z$ 105, denoted STY) and methyl-styrene ($[C_9H_{10}+H]^+$, $m/z$ 119, denoted C1STY). These 8 ArHC ions comprise 96.7±3.3% of the total ArHC signal in µg m$^{-3}$ and correspond to 69.5±19.7% of the total NMOC signal for full cW, cE and Ph 1 (cW) experiments (Figure 4; Ph 2-4 (cW):

65.2±9.8% and 13.9±12.1%). Oxygenated ArHC (such as phenolic compounds and benzaldehyde) make up an additional 1.2±2.0% contribution to the total ArHC fraction for cold-started conditions (cW, cE, Ph 1 (cW)). Their relative contribution




increases when engine conditions are hot (Ph 2-4 (cW): 5.9±1.2%). Also GDI4 shows enhanced contribution of oxygenated ArHC to the total NMOC compared to GDI1-3, which is in line with relatively enhanced hot engine emissions.

While the primary ionization pathway in the PTR-ToF-MS is proton transfer reaction by $H_3O^+$ ions, the ion source produces up to 5% of unwanted $O_2^+$. $O_2^+$ can lead to charge transfer or hydride abstraction reactions ((Amador Muñoz et al.,

2016);(Jordan et al.);(Knighton et al., 2009)). Signals at $[C_6H_6]^+$ (*m/z* 78), $[C_7H_8]^+$ (*m/z* 92) and $[C_8H_{10}]^+$ (*m/z* 106) likely derive from $O_2^+$ charged ions of ArHC, and are hence excluded from the analysis of the total mass (but support peak identification by correlation with their corresponding protonated ion at ~5% of the protonated signal). Other ions deriving from $O_2^+$ ionization are insignificant contributors to the total mass. The carbon content of the quantified ArHC corresponds to 48.8±7.6% of the FID NMHC signal (assuming equal response factors on the FID) for full cW, cE and Ph 1 (cW) (Note,

that the ratio of total NMOC mass (µgC) determined by the PTR-ToF-MS to NMHC measured by the FID (after subtraction of $CH_4$ as measured by the Picarro CRDS) is 0.65±0.15 as average of cW,cE, Ph 1 (cW) (NMHC/NMOC comparison for data for Ph2-4 are not presented due to interferences on FID in measurements of oxygen-containing hydrocarbons)). Figure 4c summarizes the ArHC emission factors, and Figure 4d gives the relative composition of the most dominant ArHC.

Of the non-aromatic peaks in Figure 4a, the largest signals occur at integer *m/z* 57 ($[C_4H_9]^+$), followed by 41 ($[C_3H_5]^+$) and

43 ($[C_3H_7]^+$), which taken together make up 7.9±4.8% of the signal for the full cycle (cW, cE) as well as for Ph 1 (cW). A larger fraction (13.2±11.9%) is observed when investigating Ph 2-4 (cW) (i.e. hot engine conditions). These ions are often fragments of larger molecules and hence not straight-forward to assign. Thus, they are included in the category of structurally unassigned hydrocarbons in Figure 4. Often, $[C_3H_5]^+$ and $[C_3H_7]^+$ are considered fragments of oxygenated parent molecules. In our experiments, however, these ions may dominantly derive from propene ($C_3H_6$), for which protonation

leads to $[C_3H_6+H]^+$, and a subsequent loss of $H_2$ leads to $[C_3H_5]^+$. The observed ratio of $[C_3H_5]^+$ and $[C_3H_7]^+$ is consistent with the ratio seen for pure propene ($C_3H_6$) injected into the instrument as reference (Figure S15). In analogy to $O_2^+$ ionization of ArHC, we find $[C_3H_6]^+$ in the spectra as insignificant signal (5% of $[C_3H_6+H]^+$). It is likely related to an $O_2^+$ charge transfer to propene ((Amador Muñoz et al., 2016);(Jordan et al.);(Knighton et al., 2009)), and supports the peak identification. The fuel contains 5%$_{vol}$ (2014) to 8%$_{vol}$ (2015) of methyl-tert-butyl-ether (MTBE), as an anti-knocking agent.

Fragmentation by proton transfer reactions of MTBE can lead to a significant signal at *m/z* 57 ($[C_4H_9]^+$). Protonated butene would also yield $[C_4H_9]^+$, but analogous to the ArHC and propene, should also give a correlated signal at $[C_4H_8]^+$ at approximately 5% of $[C_4H_9]^+$, which is not observed. The carbon content of unspecific fragments ($[C_3H_5]^+$ (*m/z* 41), $[C_3H_7]^+$ (*m/z* 43), $[C_4H_9]^+$ (*m/z* 57)) accounts for additional 4.4±3.0% of the FID NMHC signal (full cW, cE, and Ph 1 (cW)). Based on the literature reports of e.g. ((Platt et al., 2013) and (Schauer et al., 2002)) we expect a significant contribution of ethene

($C_2H_4$) to the exhaust hydrocarbons, which however, cannot be quantified by proton transfer reaction (Gueneron et al., 2015), and together with short-chain alkanes contributes in parts to the difference between the NMOC and NMHC signal. Other possibilities for parents of those potential fragments (41, 43, 57, and further $C_nH_{2n+1}^+$) are alkyl-substituted mono-aromatics, alkenes with >$C_4$, or alkanes (>$C_{10}$, potentially >$C_6$ if cyclic) ((Gueneron et al., 2015);(Erickson et al.,





2014);(Buhr et al., 2002)). While small intensities at the masses corresponding to $C_nH_{2n+1}^+$ (e.g. 71, 85, 99) are detected, we do not observe significant signals corresponding to aliphatic fragmentation patterns above $m/z$ 57. Signals indicating larger cycloalkanes or alkenes (e.g. most abundant fragments at $m/z$ 69 for substituted cyclohexane) ((Gueneron et al., 2015);(Erickson et al., 2014)) are also not abundant in our spectra, although their presence has been reported by gas-

chromatographic MS techniques in other experiments (e.g. (Saliba et al., 2017); (Zhao et al., 2016)). We cannot exclude the presence of those compounds, however, due to the limitations of our measurement principle. Further missing carbon mass in the NMOC and NMHC measurements (may result from alkyl-substituted mono-aromatics which can also lead to fragments at $m/z$ 41, 43, 57, and further $C_nH_{2n+1}^+$ as already mentioned above. The fragmentation process would result into a significant mass loss, as the aromatic ring would remain predominantly neutral (especially for mono-aromatics with long alkyl-

substituents (Gueneron et al., 2015)). For example, only 22% of the ion signal generated from n-pentylbenzene fragmentation retains the aromatic ring (19% M+H$^+$, 3% protonated benzene ring), and 88% is found at non-aromatic ions $m/z$ 41 or 43). A small contribution from oxygenated species (such as small acids and carbonyls) is found, while larger oxygenated molecules are not detected in significant amounts except for traces of benzaldehyde ($[C_7H_6O+H]^+$) and methyl-benzaldehyde ($[C_8H_8O+H]^+$). Nitrogen is found only in very few species, of which the dominant one is assigned to

acetonitrile ($CH_3CN$). Due to challenges in its quantification without proper calibration of the PTR-ToF-MS, and its unknown source (including potential outgassing from Teflon sampling lines), it was excluded from our analysis. The carbon content of oxygenated compounds found in the NMOC fraction, which have a lower response in the FID, would make up only 3.6±3.9% of the FID signal assuming a response equal to pure HCs for cW, cE and Ph 1 (cW). Hence, even if oxygenated species have a limited response in FID measurements, they do not bias the total FID NMHC measurements

substantially (assuming that the PTR-ToF-MS is able to detect and quantify all oxygenated species present). Summarizing, our above interpretation of the NMHC and NMOC closure holds for full cW and cE, and Ph 1 (cW) experiments, summing all these species and accounting for the uncertainties introduced by response factors and $k_{H3O+}$ rates of fragments, as well as species that the PTR-ToF-MS is unable to detect.

**3.5 SOA formation in OFR and SC: oxidation conditions and reacted SOA precursors**

Figure 5 shows a typical experiment during which collected primary emissions were sampled from the SC through the OFR (OFR-from-SC) and exposed to photochemistry at UV light settings of 100%, 70%, 50% in the OFR, and characterized in dark conditions (Figure 5a). After investigating OFR-from-SC aging, typically photochemistry was initiated in the SC (Figure 5b). Reactive NMOC (displayed are the dominantly observed ArHC), decay upon exposure to OH radicals, and OA mass increases. The emissions of vehicle exhaust contain NO, which can influence the chemical pathways during

atmospheric processing. In the OFR, nitrogen monoxide (NO) is converted rapidly to $NO_2$ and further to $HNO_3$, hence the OFR aging conditions when sampling from diluted exhaust (OFR-from-SC) can be considered "low NO conditions" ((Lambe et al., 2017); (Peng and Jimenez, 2017)). Only at elevated NO levels (such as during online operation of the OFR



during our 2014 measurements as discussed also in Section 3.3 and the SI), "high NO" conditions may be reached in the OFR ((Peng and Jimenez, 2017)). The dominance of the $RO_2$-NO or $RO_2$-$RO_2$ reactions is driven by the NO levels. Based on (Platt et al., 2014), $RO_2$ radicals predominantly react with NO, when the concentration of NO is higher than only 1 ppb in the SC. Before starting the SC aging by injecting HONO and initiating photo-chemistry, we titrate NO present in the SC to $NO_2$

using $O_3$. NO levels in the SC were typically below 5 ppb when photochemistry was initiated, and dropped to the detection limit within few minutes. The total $NO_y$ signal increases with time of SC experiment, which we relate to the formation of nitric acid ($HNO_3$) from primary $NO_x$ and continuous injection of nitrous acid (HONO) (see also particulate nitrate signal in Figure S10, Supporting Information), although the presence of $NO_2$ cannot be unambiguously quantified. We classify our SC experiments as "low NO" conditions, albeit NO concentrations can be higher than in the corresponding OFR experiments.

Reduced SOA yields as a function of higher NO concentrations in gasoline exhaust have been recently discussed by (Zhao et al., 2017).

While in terms of abundance of potentially SOA-forming precursors toluene (TOL) and xylenes and ethylbenzene (XYL/EBENZ) dominate over benzene (BENZ) and C3-benzenes (C3BENZ), OH reaction rates (Table 2), have the opposite trend for these compounds (C3BENZ > XYL/EBENZ > TOL > BENZ). The reacted ArHC, which is the quantity relevant

for SOA formation, is governed by the combination of abundance and reaction kinetics. Reacted ArHC (OH exposure (1.4-5.8)x$10^{11}$ molec $cm^{-3}$ s) is dominated by XYL/EBENZ (41±3%), which together with TOL (33±4%) comprises more than 70% of the reacted ArHC fraction. C3BENZ (13±2%) and BENZ (7±3%) provide smaller contributions in our cW, cE and Ph 1 (cW) experiments. C4BENZ, STY, C1STY and NAPH account for additional ~5% to the reacted ArHC fraction; other ArHC were not considered. (Time-series of typical experiments are provided in Figure 5a,b (middle); averaged contributions

of reacted ArHC as noted above in the text are displayed in Figure S4, Supporting Information, OH exposure data at the end point of SC experiments and for the OFR are provided in Figure 6 and Figure 7).

SOA mass was then predicted by accounting for the reacted mass of the dominant ArHC and their respective SOA potential using previously reported SOA yields (i.e. $SOA_{predicted} = \sum_i (\Delta NMOC_{i,reacted}(t) * yield_{i,literature})$ assuming a: constant yield as a function of OH exposure and suspended OA loading, yield data as reviewed in (Bruns et al., 2016)). As shown in

the bottom panels of Figure 5 (a and b) this does not provide a closure between reacted precursor mass and SOA mass formed. Because SOA yields are a function of the suspended absorptive mass (partitioning theory ((Pankow, 1994);(Donahue et al., 2006)), we further investigate the agreement between reacted SOA forming precursors and the formed SOA mass as SOA yield curves (discussed in the following Section 3.6).

## 3.6 Effective SOA yields

In this Section, we provide a comparison of effective SOA yields ($Y_e$), while Section 3.7 discusses differences in the chemical composition of for the SC and OFR experiments. In Figure 6a we present yields for GDI1-3 vehicle exhaust (SC



and OFR-from-SC) together with single precursor yields (SC and OFR experiments reported in the literature and new data from our OFR). Note that for GDI exhaust, $Y_e$ assumes solely BENZ, TOL, XYL/EBENZ, C3BENZ, C4BENZ, NAPH, STY, C1STY as SOA precursors. Data are presented as a function of suspended OA for all experimental conditions of cold-started GDI1-3 (i.e. for full cW, cE; and Ph 1 (cW)), while GDI4 or hot engine conditions, i.e. Ph 2-4 (cW) are not included
in the analysis due to limited experimental statistics.

For the GDI exhaust comparisons we find that
- GDI vehicle exhaust aged in the SC and the OFR result in similar effective yield curves for the two systems, with a trend for lower yields obtained in SC experiments (Figure 6c, see Section 3.6.1 for a detailed discussion).
- GDI vehicle exhaust effective SOA yields (SC and OFR), appear higher than our reference measurements conducted with the most relevant SOA precursors (ArHC) present in the vehicle exhaust. Potential reasons for the discrepancy are discussed in detail further below (Figure 6a, see Section 3.6.2).

We performed separate OFR experiments with toluene, $o$-xylene and 1,2,4-TMB as appropriate surrogates. Their effective
SOA yields obtained by our OFR measurements are in agreement with previously published SOA yield curves for ArHC photo chemically aged in an analogous OFR ($m$-xylene, by (Ahlberg et al., 2017)) and in a SC (benzene, toluene, o-xylene, from (Li et al., 2016a) and (Li et al., 2016b), shown in Figure 6b). All yields (for vehicle exhaust as well as for single precursors) increase with the suspended OA concentration, and range up to 0.8-1 for OFR vehicle experiments above 300 μg m$^{-3}$. In the atmospherically more relevant concentration range of 10 to 100 μg m$^{-3}$, the effective yield spreads from a few
percent (below 15%) up to 20-50% (at 100 μg m$^{-3}$).

**3.6.1 SC vs. OFR yields of vehicle exhaust (Figure 6c).** Aging of GDI vehicle exhaust in the SC and the OFR resulted in similar SOA yield curves, with a trend towards lower yields for SC experiments (Figure 6c). The contribution of SOA precursors to the reacted ArHC fraction is comparable between the systems (Figure S4). OFR-derived effective yields for
GDI vehicle exhaust appear higher than our SC-derived yields. One reason might be higher initial levels of NO in the SC experiments compared to the OFR, suppressing the SOA yields, as recently discussed by (Zhao et al., 2017)). However, as discussed in the Methods Section, we expect both SC and OFR yields to be underestimated, by factors of approximately 1.5-2 (SC) and 1.25 (OFR) ((Platt et al., 2017);(Zhang et al., 2014);(Palm et al., 2016)) due to vapor wall losses. Corrections would reduce the discrepancy between the two systems.
Our SC-derived yields themselves are variable (Figure 6c). While effective yields are presented vs. OA as assumed absorptive mass here, we find significant formation of $NH_4NO_3$ during SC experiments (Figure S10b, Table S4; $NO_3$ is used as a surrogate for $NH_4NO_3$; $NO_3$/OA ratios are significantly higher in the SC (4.00±2.11) compared to the OFR (0.43±0.26)). Presenting SC yields as a function of the sum of OA+$NO_3$+$NH_4$ in Figure S13 appears to bring them into better agreement



and indicates a relationship between the yields and $NH_4NO_3$ for three experiments (and a correlated trend of higher yields at higher initial NOx levels, which is contradicted by (Zhao et al., 2017)). The high concentrations of $NH_4NO_3$ formed in 3 of those experiments (several hundreds of $\mu g\ m^{-3}$) is outside our $CO_2^+$-AMS interference calibration and data of those three experiments may still be associated with a positive mass bias even after correction ((Pieber et al., 2016)). Neglecting a

potential additional $CO_2^+$-AMS interference, the data indicate that inorganic nitrate (as well as the associated water) may contribute to the absorptive mass (Stirnweis et al., 2017)), however, a detailed analysis is beyond the scope of our study.

**3.6.2 SOA yield of vehicle exhaust vs. single precursors (Figure 6a).** GDI vehicle exhaust effective SOA yields (SC and OFR), appear higher than our reference measurements conducted with the most relevant SOA precursors (ArHC) present in

the vehicle exhaust. However, we can explain a significant fraction (> 0.5) of our obtained vehicle exhaust effective yields with the mix of reacted o-xylene and toluene (OXYL/TOL, 3:1) as presented in Figure 6a. This generally indicates that we are able to identify the most relevant SOA precursors in the vehicle exhaust. The discrepancy between the effective SOA yields for GDI and the measured yields of major precursors may result from various reasons:

- Missing precursor (unidentified/undetected in NMOC analysis, see Section 3.4):

Our calculated effective SOA yields assumes that all our relevant SOA precursors found in the exhaust are identified and their decay quantified, as defined in Eq. (2). We are able to explain only 65%±15% of the total non-methane hydrocarbon signal with the carbon found in the NMOCs (Section 3.4). This leaves about 35% additional carbon to be attributed to other molecules. While both aromatic ((Odum et al., 1997);(Ng et al., 2007b);(Hildebrandt et al., 2009);(Loza et al., 2012);(Platt et al., 2014)) and aliphatic (especially alkanes) ((Lim and Ziemann, 2005);(Loza et al.,

2014)) species are found in vehicle exhaust and may form SOA, aliphatic species do so only if their carbon chain is sufficiently long and does not substantially fragment during reaction. Short-chain alkanes ($<C_8$) are expected to have only low SOA yields at typical ambient OA levels (Jordan et al., 2008). ArHC (starting from the simplest with $C_6$) instead have been shown to produce highly oxygenated multifunctional organics with only few OH attacks ((Molteni et al., 2016);(Schwantes et al., 2016)), making them efficient SOA precursors with high SOA yields, especially under "low

NO" conditions (Ng et al., 2007b). As discussed above, ArHC dominate the total gas-phase organic compounds as determined by the PTR-ToF-MS. The identified ArHC have saturation vapor pressures ($C^*$) at or above $10^6\ \mu g\ m^{-3}$ (VOCs) with a small contribution from aromatics (such as naphthalene) in the IVOC range ($C^*=10^2$-$10^6\ \mu g\ m^{-3}$), rather than SVOC ($C^*\ 10^{-1}$-$10^2\ \mu g\ m^{-3}$) or lower volatility ($C^* <10^{-1}\ \mu g\ m^{-3}$) compounds (Pandis et al., 2013). The larger contribution of VOCs than IVOCs to SOA is consistent with the results by (Zhao et al., 2016), who found a ratio of ~10

in the VOC-to-IVOC ratio for gasoline exhaust. The other detected or postulated organic compounds in our exhaust samples (e.g. short-chain alkanes, small oxygenated molecules) are not expected to contribute significantly to the SOA mass.

In sum, missing SOA precursors might thus comprise



- o 1) additional ArHC (e.g. (Nordin et al., 2013)) which are a) unidentified or b) present only in small quantities and hence not taken into account in the effective yield analysis (such as the oxygenated ArHC because they are present at only roughly 1% of our total NMOC).
- o 2) short-chain alkanes/alkenes which contribute a significant fraction to the total carbon in the gas-phase, but have low SOA yields and
- o 3) long-chain alkanes and alkyl-substituted mono-aromatics which are not well detected by the PTR-ToF-MS technique due to low protonation affinity or substantial fragmentation.

Also MTBE is present in significant amounts in the exhaust. It has currently not been considered as a significant SOA precursor, due to its small carbon number and high volatility, but should be investigated in future work.

- • Reference SOA yields aren't chosen correctly:
  - o Aromatic isomers show a distribution of yields based on carbon number, number of aromatic rings, and degree and location of substitution, which are not fully covered by the reference compounds selected for testing. Isomers present in the exhaust may enhance the effective SOA yield relative to the reference measurements.
  - o Benzene contributes less than 10% to the reacted NMOCs (Section 3.5 and Figure S4) and was therefore not tested separately in our OFR. However, its SOA yield has been reported to exceed that of alkylated analogous compounds, such as xylenes or higher alkylated benzenes ((Li et al., 2017);(Bruns et al., 2016)). Benzene may hence contribute to the enhanced effective SOA yield relative to the reference measurements. The same is true for oxygenated ArHC which were not considered in our analysis due to their relatively low contribution to the NMOC composition.

- • Differences in the experimental conditions of single/mixed aromatics vs. the more complex vehicle exhaust:
  - o The influence of NO on SOA yields has been previously addressed in the literature for biogenic and anthropogenic sources (e.g. (Ng et al., 2007b);(Ng et al., 2007a)), and generally indicates that at higher NO conditions, lower SOA yields are observed. (Zhao et al., 2017) showed this recently also for gasoline exhaust.. We choose NO-free conditions as comparison points for our yields (based on the discussion in Section 3.5). Hence our SOA reference yields for comparison are an upper estimate in this regard, and choosing a high-NO reference would make the discrepancy to the vehicle exhaust even bigger).
  - o The influence of other exhaust constituents which are absent in our reference measurements, such as eBC acting as a seed, $NH_3$ and the formed $NH_4NO_3$, the presence of $NO_2$ or chemical processing by unwanted formation of $NO_3$-radicals ((Schwantes et al., 2016)) on SOA yields is insufficiently addressed in the literature to discuss in detail, but a potential influence cannot be excluded in our work.



### 3.7 SOA composition of SC and OFR

The bulk OA elemental O:C and H:C ratios for SOA formed from GDI vehicle exhaust for the SC and OFR-from-SC experiments at varied OH exposure (OFR UV intensity) are shown in Figure 7. In all cases, the SOA composition shifts towards higher O:C and lower H:C ratios as a function of OH exposure. However, the OFR yields higher H:C values, with

decreasing divergences at higher OH exposures. The end point of the SC experiments in terms of max. OH exposure corresponds to the 70% UV setting in the OFR on average. Concerning the O:C ratio we find agreement with OFR-from-SC at similar OH exposure for three SC experiments, while the other three experiments yield relatively higher O:C ratios at the same OH exposure. The three SC experiments with higher O:C ratios also had higher SOA yields than the other experiments (see Figure S13 for detailed yields and Figure S16 for the differences in the O:C ratios of these experiments). Those three

experiments are characterized by a higher absolute as well as relative $NH_4NO_3$ concentration, which, as noted above is outside our $CO_2^+$-AMS interference calibration and data may still be associated with a positive bias towards higher O:C even after correction ((Pieber et al., 2016)). Hence we consider our data in general agreement with regards to the O:C ratio as a function of OH exposure in OFR and SC. This does not apply for the H:C ratio, however.  Possible reasons for differences in the products between OFR and SC in terms of the bulk elemental composition ratio include differences in the aging

conditions (oxidant concentrations, ratios of $OH/O_3$, presence/absence/amount of $NO_x$, ratios of $NO/NO_2$, (unwanted) presence of $NO_3$ radicals, presence of secondary $NH_4NO_3$ and associated water acting as additional absorptive mass and leading to chemical differences of the products) and other experimental uncertainties (loss of secondary vapors to the walls or loss by UV interaction at different rates between SC and OFR, please refer also to the discussion in section 3.6). Further investigation of those aspects requires information on a molecular level and should be the focus of future comparison studies

between the two systems.

### 4 Conclusions

We studied exhaust from modern GDI vehicles as a function of driving cycles, individual phases thereof and engine temperature (cold-started, hot-started), and evaluated the effect of retrofitted, prototype GPFs on the primary and secondary emissions. We present a detailed analysis of primary NMOC composition from PTR-ToF-MS measurements and the

associated SOA formation potential evaluated by SC and OFR experiments, and provide a quantitative link between the NMOC fraction and the observed SOA. We also provide OFR-obtained ArHC SOA yield curves for toluene, *o*-xylene and 1,2,4-TMB.

For all GDI vehicles, the dominant fraction of NMOC emissions is released during the starting phase of cold-started vehicle tests, before after-treatment systems are hot. NMOC emissions of cold-started vehicles are dominated by aromatic

hydrocarbons, especially toluene, xylenes/ethylbenzenes, and C3-alkyl-benzenes and benzene. SOA formation is likewise governed by the cold-start. These results are independent of testing protocol, demonstrating that the performance of after-




treatment systems and not the driving behavior governs these emissions. It appears that GDI4, which is in line with Euro 6 regulations regarding its NMHC emissions, has a reduced overall and cold-start NMHC EF, and that its emissions during hot-engine conditions contribute a bigger relative fraction to the total compared to GDI1-3. Additionally, by trend, oxygenated ArHCs have a slightly enhanced fraction in GDI4 compared to GDI1-3 exhaust.

GPF application efficiently removes eBC, which is the dominant component of primary PM, but has small effects on the minor POA fraction. The volatile POA fraction passes through the filter in the vapor phase and later condenses when the exhaust is emitted and cooled; hence POA emission factors are not significantly reduced. NMOC emissions and SOA formation are unaffected by the tested GPFs, both catalytically inactive and with catalytically active coating. This means that retrofitting GDI vehicles with GPFs will likely result in an important reduction of the total primary PM emitted (removal of

refractory material), but will not (or only to a small extent) reduce NMHC (or NMOC) emissions including ArHC and will not directly lead to a reduction of the SOA formation. Future work on so-called "4-way catalysts", i.e. a TWC catalyst directly applied onto a GPF and installed at the location of the current TWC for simultaneous filtration of particulates and catalytic conversion of NMHC (or NMOC) should be conducted to understand whether reductions of SOA precursors, SOA production, and semi-volatile primary PM can be achieved.

Effective SOA yields as a function of suspended OA mass concentration appear rather higher for the OFR than the SC. We believe that the differences are in parts due to unaccounted experimental losses, as well as a potential influence of $NO_x$-chemistry in our SC experiments. Trends in the atomic O:C and H:C ratios of the bulk OA mass suggest that differences in SC and OFR aging or operational conditions affect the chemical composition of the formed SOA. These divergences cannot be unambiguously attributed to a specific process. Further investigation of those aspects requires information on a molecular

level and should be the focus of future comparison studies between the two systems.

Based on gas phase compositional analysis, SOA precursors from GDI vehicles are likely dominated by a few aromatic hydrocarbons. While a large fraction (> 0.5) of the SOA formed can be attributed to the identified aromatic precursors, the effective SOA yield determined for GDI vehicle exhaust appear relatively higher than the sum of the yields of the major aromatic precursors identified (toluene, $o$-xylene, 1,2,4-TMB) under comparable OA loadings. This may have diverse

reasons including the presence of other aromatic isomers, unaccounted for precursors (which cannot be detected by PTR-ToF-MS measurements), the influence of the complex exhaust gas matrix ($NO_x$, secondary $NH_4NO_3$, eBC, other constituents) compared to single precursor testing, or experimental uncertainties.

**Supporting Information**

Provided as noted in the main text.





## Acknowledgements

This work was funded by the CCEM project GASOMEP (http://www.ccem.ch/gasomep). SMP acknowledges support from the Swiss National Science Foundation (SNF Project 140590). JGS acknowledges support from the SNF starting grant BSSGI0_155846. We would like to thank René Richter for his invaluable technical support as well as the staff of the

Laboratories for IC-Engines and Exhaust Emission Control of the Berne University of Applied Sciences in Biel, who conducted the vehicle testing.

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





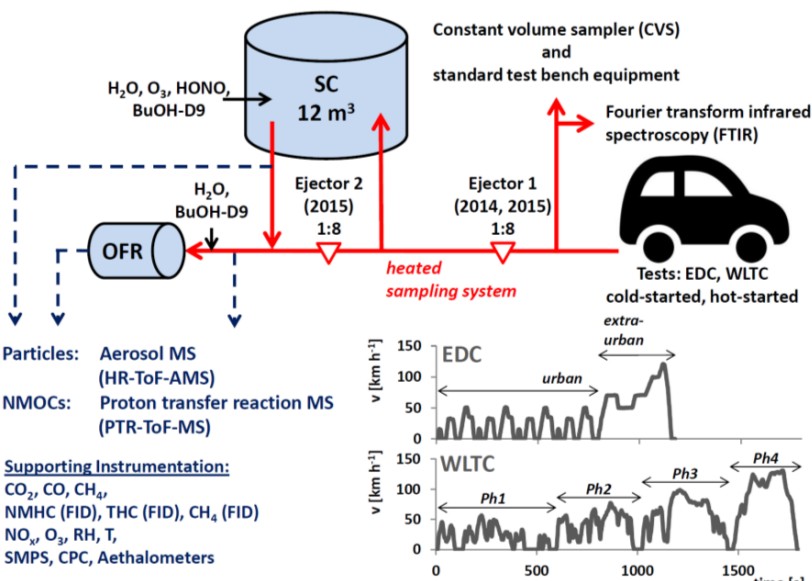

**Figure 1. Schematic (not to scale) of the experimental set-up.** Vehicles were driven over regulatory driving cycles (EDC and WLTC,
for which speed profiles are shown in the figure) on a chassis dynamometer test bench. Emissions were sampled through a heated dilution
and sampling system using 1 or 2 ejector dilutors into the PSI mobile SC (Platt et al., 2013) and the potential aerosol mass (PAM)
oxidation flow reactor (OFR) (Bruns et al., 2015). Instrumentation for characterization of fresh and photo-chemically aged emissions is
listed. The raw exhaust was also sampled at the tailpipe using standard test bench equipment to monitor regulatory species (diluted in a
constant volume sampler, CVS) and unregulated emissions (with Fourier-Transformed Infrared Spectroscopy, FTIR).





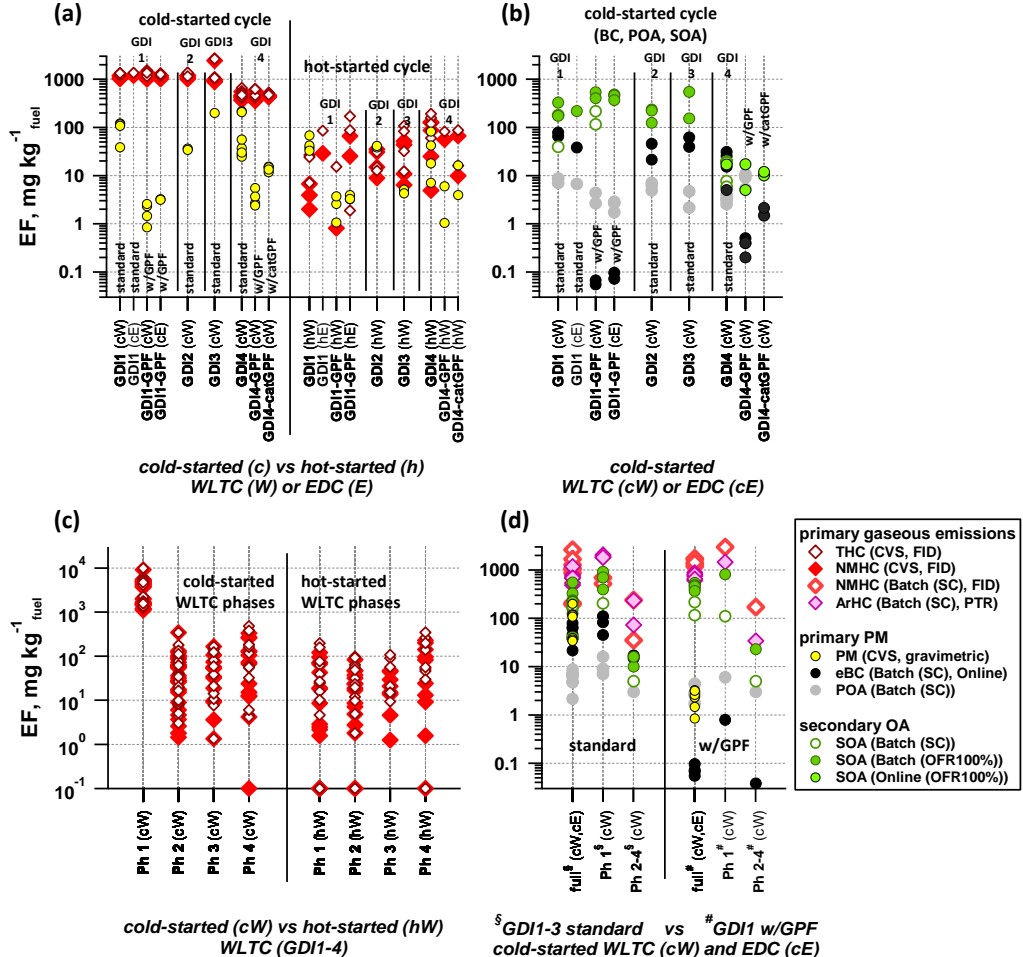

**Figure 2. Emission factors (EF) of pollutants from cold-started ("c") and hot-started ("h") test cycles (WLTC ("W") and EDC ("E")).** Individual cW and hW phases are indicated as "Ph" 1-4. **(a)** EFs of total and non-methane hydrocarbons (THC, NMHC) and primary gravimetric particulate matter (PM) from CVS measurements over entire test cycles for different vehicle configuration and test conditions, **(b)** EFs of primary PM (equivalent black carbon (eBC) and primary organic aerosol (POA)), and secondary organic aerosol (SOA) formed during photochemical aging in SC, OFR-from-SC experiments and during online operation of the OFR (OFR at 100% UV intensity) per vehicle configuration for cold-started test cycles (note that POA EFs for GDI4-catGPF (cW) are not available). **(c)** EFs of the same cW and hW experiments presented in (a) separated into individual cycle phases. **(d)** EFs of primary gravimetric PM, POA and eBC, NMHC and aromatic hydrocarbons (ArHC) and SOA full cW and cE, compared to individual phases of cW. Note that the EF for eBC for Ph 2-4 (cW) is 17 mg kg$^{-1}_{fuel}$ and that the data point is hidden behind the SOA data points in the graphical presentation. **(a-d)** EF calculation is detailed in the SI. The time-resolved SOA profile from online OFR measurements conducted on GDI4 in 2015 (standard and catGPF) is available in the Supporting Information (Figure S14).





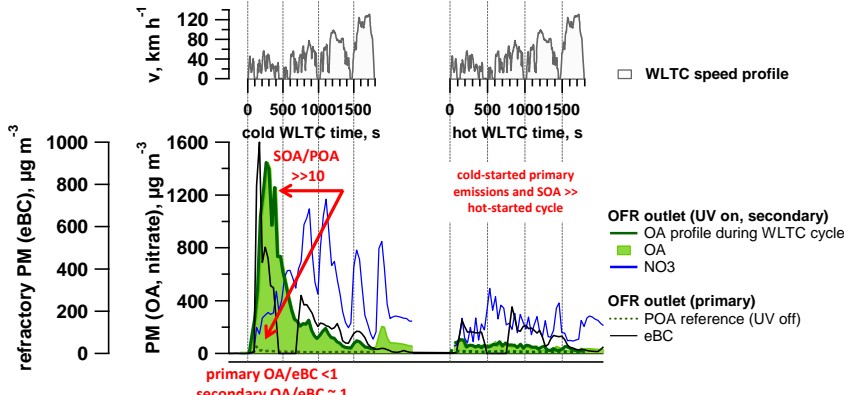

**Figure 3. Time-resolved aging of cold- and hot-started emissions (WLTC) (GDI1, standard configuration, Expt A2, extended version in Supporting Information, Figure S6).** Top: WLTC speed profile. Bottom: OA profile during WLTC presenting the OA measurement during the 30 min driving test with OFR at 100% UV intensity; due to a delay in the OFR the signal after the WLTC is finished is displayed as well), nitrate aerosol (inorganic, ammonium nitrate, displayed is only NO$_3$), as well as POA and equivalent black carbon (eBC). Further experiments (A1 (a repeat of GDI1 in standard configuration, Supporting Information, Figure S5, and B1 (Supporting Information, Figure S7) and B2 (Supporting Information, Figure S8), which are experiments of GDI1 equipped with GPF) are presented in the SI. Time-resolved profiles of GDI4 in standard configuration and with catGPF are given in Figure S14.



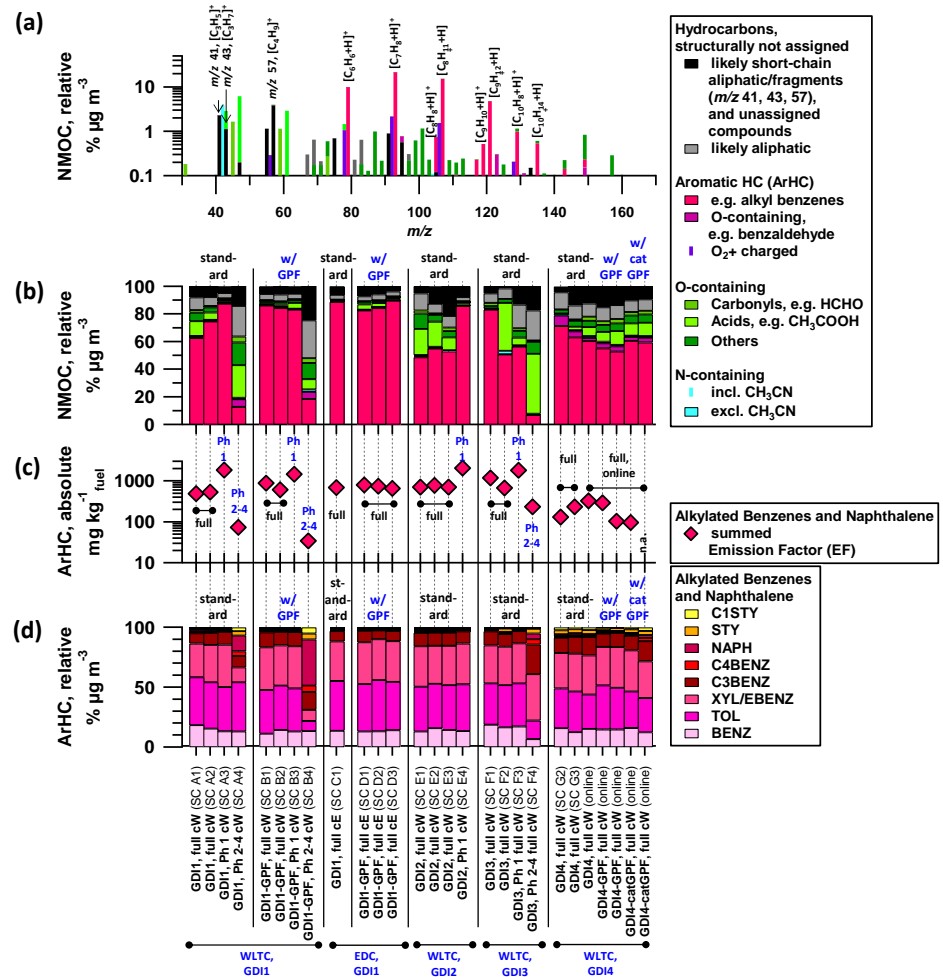

**Figure 4. PTR-ToF-MS derived NMOC composition ("cold-started cycles"). Data are collected by batch sampling ("SC") or during online measurements ("online"). (a)** PTR-ToF-MS mass spectrum of emissions from GDI1 in standard configuration sampled into the SC during a cold-started WLTC (cW). **(b)** Relative composition of the PTR-ToF-MS derived NMOC fraction, **(c)** total ArHC EFs, and **(d)** relative contribution of the 8 dominant ArHC. **(b-c)** Data correspond to vehicle exhaust for GDI1 (expt. A-D), GDI2 (expt. E), GDI3 (expt. F) and GDI4 (expt. G) sampled into the SC during full cW and cE driving tests, or individual phases of cW, or measured "online". The identifier in parenthesis specifies individual SC experiments (see also Supporting Information, Table S4-S7, for SC experimental conditions). Note that the total NMOC levels for Ph 2-4 (cW) are about 1/10 of full cW and Ph 1 (cW) concentrations only and measurements are close to the background measurements (signal not significantly different from 3 standard deviations of the background measurement).

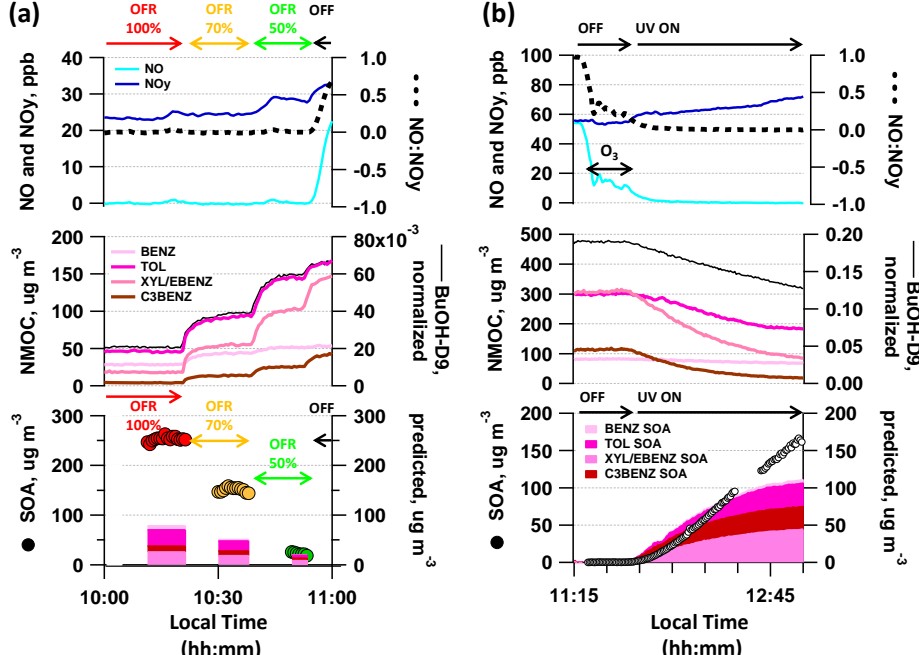

**Figure 5. SOA precursor behavior in OFR and SC during photo-chemistry.** Example of decay of organic vapors (selected aromatic NMOCs: benzene (BENZ), toluene (TOL), *o-/m-/p*-xylene (XYL) or ethylbenzene (EBENZ), C3-benzenes (C3BENZ)) upon
5  photochemistry and associated SOA formation in **(a)** OFR (sampling from SC batch at different UV intensities, displayed is expt D3) and **(b)** aging in SC (displayed is expt B1). **(a-b)** UV on/off (and UV intensities for OFR) and $O_3$ injection (for SC) are indicated. Also displayed is the $NO:NO_y$ ratio during the experiments, as well as the decay of the OH tracer BuOH-D9. "Predicted" refers to predicted SOA mass (purple color) from the reacted NMOC vapors and a constant literature based SOA yield applied to it (as reviewed in (Bruns et al., 2016) (BENZ: 0.32, TOL: 0.27, XYL/EBENZ: 0.20, C3-BENZ: 0.32) Note: ammonium nitrate ($NH_4NO_3$) is also formed upon photo-
10  oxidation due to the presence of $NO_x$ and $NH_3$, but is not displayed here, see Supporting Information, Figure S10 instead. Reacted ArHC fractions upon OH exposure in the SC and OFR are provided in the Supporting Information, Figure S4. Local time is given in intervals of **(a)** 30 min and **(b)** 15 min.





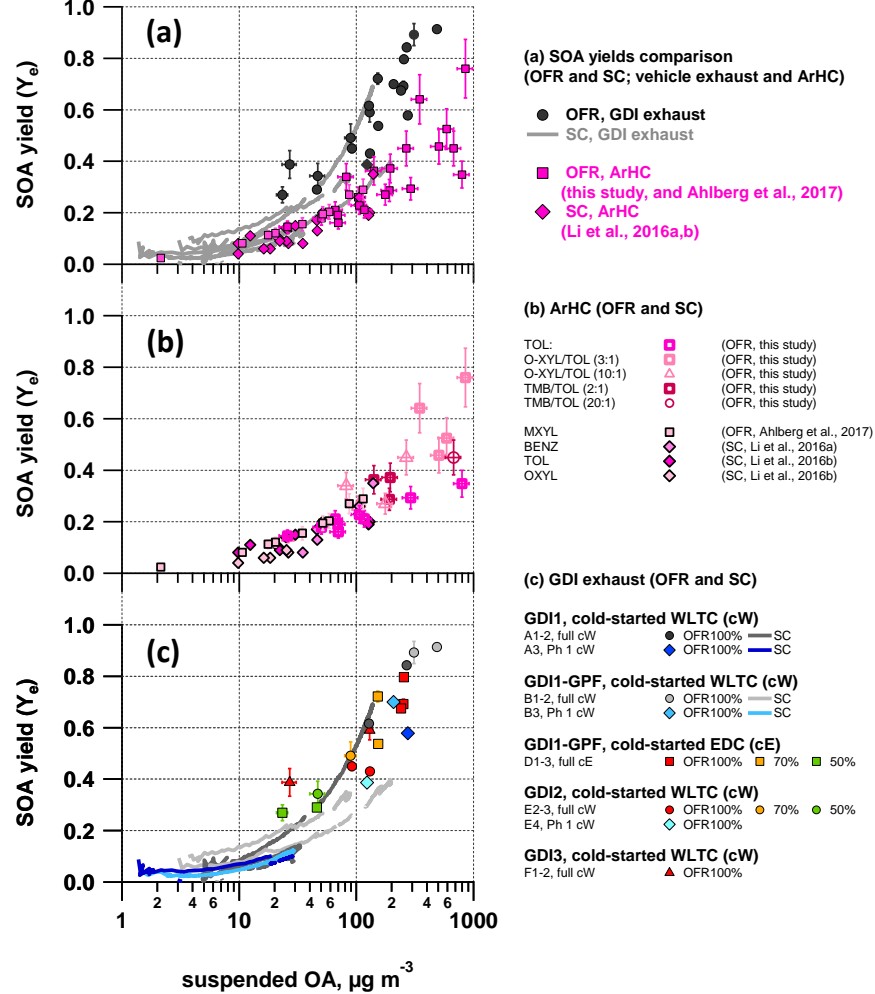

**Figure 6. Effective SOA yields.** Vehicle exhaust from GDI1-3 (full cW, full cE, Ph1 (cW)) photo-chemically aged in the SC and OFR-from-SC compared to effective SOA yields from selected ArHC (toluene, *o*-xylene, 1,2,4-TMB) photo-chemically aged in our OFR (this study, w/o NO; additional *m*-xylene data from (Ahlberg et al., 2017)) and in a SC (data from literature:
5      benzene, toluene, *o*-xylene from (Li et al., 2016a) and (Li et al., 2016b), w/o NO)). **(a)** all data combined, **(b)** OFR (average±15% measurement variability) and SC yields of single ArHC or mixtures, **(c)** vehicle exhaust photo-chemically aged in SC and OFR-from-SC (average±1SD of AMS OA measurement during stable conditions). Note that error bars on data from OFR represent the variability of the measurement rather than the total uncertainty of the data. SC yield curves are presented in more detail in SI Figure S13 **(a-c)** OH data are given in the legend of Figure 7 and summarized here: OH exposures (Barmet et
10      al., 2012) range up to $1.4 \times 10^{11}$ molec cm$^{-3}$ s, in ~2 hours of SC photochemistry experiments (average [OH]=$2 \times 10^{7}$ molec cm$^{-3}$). OFR100%: [OH]=$(2.7-5.2) \times 10^{9}$ molec cm$^{-3}$; [OH]$_{exp}$=$(3.0-5.8) \times 10^{11}$ molec cm$^{-3}$ s (at ~8 ppm O$_3$). OFR70%: [OH]=$(1.4-2.2) \times 10^{9}$ molec cm$^{-3}$; [OH]$_{exp}$=$(1.6-2.5) \times 10^{11}$ molec cm$^{-3}$ s (at ~3 ppm O$_3$). OFR50%: [OH]=$(0.28-0.44) \times 10^{9}$ molec cm$^{-3}$;





[OH]$_{exp}$=(0.31-0.49)x10$^{11}$ molec cm$^{-3}$ s (at ~0.7 ppm O$_3$). The max. OH exposure in the SC experiments corresponds to the range of green to orange colored OFR data points in panel (c).

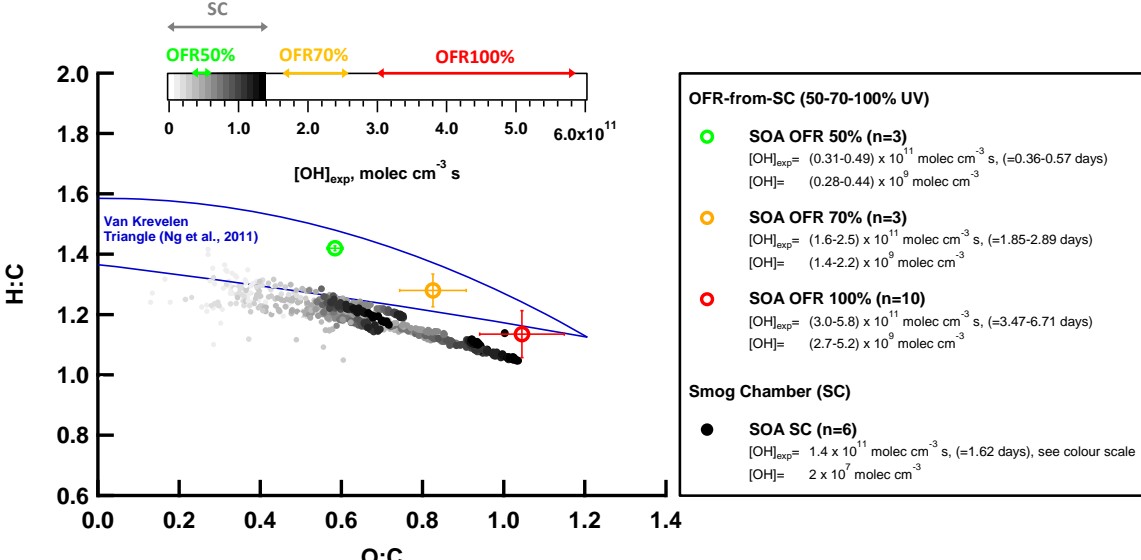

**Figure 7. Bulk OA composition of SC and OFR SOA.** Van-Krevelen plot (O:C vs H:C ratio) for SOA formed during SC expts (n=6, GDI1 standard and w/GPF, cW and Ph 1 (cW)) and OFR-from-SC data points (n=10, GDI1 standard and w/GPF, full cW, full cE, Ph 1 (cW)) at different OFR UV settings (100%, 70%, 50%). The POA contribution was subtracted from the total OA bulk composition; SOA/POA ratios are far above a factor of 10. OH and O$_3$ information is provided in Figure 6. The Aiken parameterization ((Aiken et al., 2007); (Aiken et al., 2008)) has been applied to HR fitted data in order to allow for a better comparison with previously published data. Lines indicate the Van-Krevelen (VK) space typical for ambient AMS measurements as presented by (Ng et al., 2011). Error bars represent one standard deviation corresponding to the average for several experiments presented and reflect measurement variability rather than total uncertainties. [OH]$_{exp}$ in days refers to an assumed average ambient [OH] of 10$^6$ molec cm$^{-3}$. Figure S16 provides details on single SC experiments.



5 **Table 1. Vehicles** (details in Table S1, Supporting Information) and tests (*n* gives the number of driving tests conducted; EDC tests were only conducted with GDI1 and GDI1 w/GPF).

| Vehicle Code | Vehicle Type | Expt. Set | cold-started WLTC | hot-started WLTC | cold-started EDC | hot-started EDC |
|---|---|---|---|---|---|---|
| **GDI1** | Opel Insignia; Euro 5, standard configuration | 2014 (I) | *n* =4 | *n* =4 | *n* =1 | *n* =1 |
| **GDI1 w/GPF** | Opel Insignia; Euro 5, with retrofitted GPF (underfloor) | 2014 (I) | *n*=4 | *n* =4 | *n* =3 | *n* =3 |
| **GDI2** | Opel Zafira Tourer, Euro 5 | 2015 (II) | *n* =4 | *n* =4 | -- | -- |
| **GDI3** | VW Golf Plus, Euro 4 | 2015 (II) | *n* =4 | *n* =4 | -- | -- |
| **GDI4 (2014)** | Volvo V60, Euro 5, standard configuration | 2014 (I) | *n* =4 | *n* =4 | -- | -- |
| **GDI4 (2015)** | Volvo V60, Euro 5, standard configuration | 2015 (II) | *n* =3 | *n* =1 | -- | -- |
| **GDI4 w/GPF** | Volvo V60, Euro 5, with retrofitted GPF (underfloor) | 2015 (II) | *n* =4 | *n* =2 | -- | -- |
| **GDI4 w/catGPF** | Volvo V60, Euro 5, with retrofitted catGPF (underfloor) | 2015 (II) | *n* =4 | *n* =2 | -- | -- |

**Table 2. NMOC information** (list of dominant peaks).

| Ion, $m/z$ | Chem. Formula | Assignment | Denotation | $k_{H3O+}$[a] $cm^3 s^{-1}$ | $k_{OH}$[b] $cm^3$ $molec^{-1}$ $s^{-1}$ |
|---|---|---|---|---|---|
| 79 | $[C_6H_6+H]^+$ | benzene | BENZ | $1.93 \times 10^{-9}$ | $1.22 \times 10^{-12}$ |
| 93 | $[C_7H_8+H]^+$ | toluene | TOL | $2.08 \times 10^{-9}$ | $5.63 \times 10^{-12}$ |
| 107 | $[C_8H_{10}+H]^+$ | *o-/m-/p*-xylene, ethylbenzene | XYL/E-BENZ | $2.26 \times 10^{-9}$ | $(7\text{-}23) \times 10^{-12}$ |
| 121 | $[C_8H_{12}+H]^+$ | $C_3$-alkyl-benzenes | C3BENZ | $2.39 \times 10^{-9}$ | $(6\text{-}57) \times 10^{-12}$ |
| 135 | $[C_{10}H_{14}+H]^+$ | $C_4$-alkyl-benzenes | C4BENZ | $2.50 \times 10^{-9}$ | $(5\text{-}15) \times 10^{-12}$ |
| 129 | $[C_{10}H_8+H]^+$ | naphthalene | NAPH | $2.45 \times 10^{-9}$ | $23 \times 10^{-12}$ |
| 105 | $[C_8H_8+H]^+$ | styrene | STY | $2.27 \times 10^{-9}$ | $28 \times 10^{-12}$ |
| 119 | $[C_9H_{10}+H]^+$ | methyl-styrene | C1STY | $2.00 \times 10^{-9}$ | $(51\text{-}57) \times 10^{-12}$ |
| 41 | $[C_3H_5]^+$ | HC fragment | - | $2.00 \times 10^{-9}$ | n.a. |
| 43 | $[C_3H_7]^+$ | HC fragment | - | $2.00 \times 10^{-9}$ | n.a. |
| 57 | $[C_4H_9]^+$ | HC fragment | - | $2.00 \times 10^{-9}$ | n.a. |

10 Ions are referred to with their integer mass-to-charge ($m/z$) ratio for simplicity, but are identified based on the HR derived exact $m/z$ instead. n.a.=not applicable. [a]$k_{H3O+}$ from (Cappellin et al., 2012), [b]$k_{OH}$ from (Atkinson and Arey, 2003), range corresponds to isomers.