# Peer review of "Gas phase composition and secondary organic aerosol formation from standard and particle filter-retrofitted gasoline direct injection vehicles investigated in a batch and flow reactor"

_Atmospheric Chemistry and Physics, 2017_

## Referee Comment (RC1) · Anonymous Referee #1 · 23 Dec 2017

This paper evaluates the gas phase composition and secondary organic formation from gasoline vehicles in a batch and flow reactor. Excusive results about primary emission factors, gas vapor composition and SOA formation are shown. This paper is well written and organized. The paper describes a large amount of data, but critical evaluation and analysis is missing, that is needed to have confidence on the quantification of results. Therefore I recommend that the paper may be published in ACP after addressing the major revisions below.

Major revisions:

1) The analysis of vapor losses to walls is inadequate. Enormous progress has been made on this area recently, and there is no excuse to ignore those corrections in current studies. Comparability with past studies that were performed when vapor wall losses were not understood is not an excuse to ignore this major issue. Comparability with future studies, for which all the good ones will include analysis of this effect, should be the relevant criterion. It will benefit the citation of this paper for showing both results.

This is an essential correction which could introduce large bias (a factor of 3-4 in Zhang et al. 2014) and is related to multiple key calculation including SOA yield, NMHC composition etc. Taken vapor loss corrections into account, comparison results can be more accurately assessed, especially crucial for some main focuses of this paper: SOA yield between SC and OFR, SOA yield of vehicle exhaust vs single precursors. As the authors stated in page 21 line 27-29 "we expect both SC and OFR yields to be underestimated, by factors of approximately 1.5-2 (SC) and 1.25 (OFR) ((Platt et al., 2017);(Zhang et al., 2014);(Palm et al., 2016)) due to vapor wall losses. Corrections would reduce the discrepancy between the two systems." Thus a gas vapor wall loss correction is needed for this study. The model of Krechmer et al. (2016) can be used for the SC, and has been recently shown ((Ye et al., 2016); also at AAAR 2017) to be consistent among all Teflon chambers. The model of Palm et al. (2016) should be applied to the OFR.

2) The comparison on SOA formation between OFR and SC should also consider gas vapor loss in the tubing (Pagonis et al., 2017). The different tubing lengths and materials can also result in a SOA yield difference between SC and OFR. A vapor loss calculator can be found in Pagonis et al. (2017). Please clarify the tubing material and length as well.

Page 16 line 17-18: In addition to the residence time of OFR, this SOA delay was also possibly caused by the delayed gas vapor in the tube, as suggested in Pagonis et al. (2017).

3) The particle losses due to heating in the sampling line and the hotter temperature in SC when UV light is on should be addressed. The aerosol loss due to tubing (under no heating condition) needs to be estimated as well. The model for aerosol loss calculation in the tubing can be found in (von der Weiden et al., 2009). The particle loss due to heating can be experimentally determined.

4) Page 10 line 20-30: The SOA photolysis in OFR should be considered as well. For example in OFR with full UV light setting (100%), half of the SOA from toluene SOA (OH chemistry) or naphthalene SOA (OH chemistry) can be photolyzed under 254 UV light with a low quantum yield of 0.1, as shown in Fig. 8b in (Peng et al., 2016). This photolysis effect on SOA formation and SOA yield calculation under 50% and 100% UV light setting should be considered.

5) Page 14 line 1-2. How did the mass quantification between AMS and SMPS compare? It is essential to document this comparison in the form of scatterplots and regressions. Collection efficiency (CE) vary with chemical composition and aerosol phase (Middlebrook et al., 2012). CE for AMS quantification should vary in this study since relative OA and NO3 fraction in total aerosol changed a lot. Why do the authors choose a CE ~1 here. I would expect a slight variation on RIE since there were POA dominated periods during the studies, for which RIE may be higher (Jimenez et al., 2016; Murphy, 2016). Attention needs to be paid for the size cut differences between AMS and SMPS as well, when nucleation was happened.

6) Schematic of the sampling strategy is confusing. I did not get the timing for the sampling strategy. Was the UV light setting is constant during vehicle testing cycle? Then the background of aerosol and gases under UV light is off in SC and OFR was obtained by repeating the testing cycle of the cars? Sufficient detail needs to be provided, that would enable someone else to repeat the experiments, as it is standard in scientific publications.

Other revisions:

Page 5 Line 8: EDC was defined as "older" low-road European Driving cycle (EDC), which is inconsistent with the definition of "New" EDC in the abbreviation/definition list.

Page 7 line 26-27: Please specify the dilution factor for smog chamber (SC) and oxidation flow reactor (OFR).

Page 8 line 14-15: What is the aerosol background level in SC and OFR experiments with clean air when the UV light is on? If it is high, this background needs to be subtracted in the calculation of formed SOA mass concentration.

Page 8 line 26-29: Figure S1 shows the aerosol and gas-phase species are sampled through the same tube in the center of OFR. If it is true, there will be large loss either on VOCs species (if stainless or copper tube is used) or on aerosols (if Teflon tube is used)

Page 9 line 26-27: The particle loss can be determined by measuring the aerosol concentration before and after OFR when the UV-light is off. The aerosol concentration before the OFR can also be roughly determined with aerosols in the SC chamber if the dilution factors and volatility of OA are known in OFR and SC. There have also been some reports from FIREX (Jesse Kroll's group) that particles containing BC can be charged by the UV lights and be lost much faster, also see (Federer et al., 1983). Was this effect evaluated?

Page 10 line 16: "1000-5000 nm2 cm-3" Is this unit true? These values indicate the particle surface areas in this study are very small, which is inconsistent with mass values reported in Fig. 3. It is ~$10^{^6}$ times less than those in the typical chamber studies, e.g. (Zhang et al., 2014).

Page 11 line 19-24: Does the OH exposure estimated from BuOH-D9 agree with the OH exposure calculated based on Peng et al. (2015)? A plot showing the comparison of these two methods will beneficial for readers to understand how accurate of the OH exposure used here.

Page 12 line 2: Please specify the dilution factor.

Page 12 line 5-7: Why heat the sample before CPC? Why 300 $^{o}$C.

Page 14 line 19-24: These descriptions cannot be found in the Fig. 1 e.g.. No graph compares "Ph 1 of cW and hW vs. Ph 2-4 of cW and hW" in line 20.

Page 15 line 2-3: It is hard to draw such a conclusion based on Fig. 2b. A scatter plot between POA+BC vs PM is needed or at least please give the value POA+BC.

Page 15 line 9: Please give the value ranges for "previous finding".

Page 15 line 16-19: A smaller vapor loss in the OFR is also a possible explanation.

Page 18 line 5: publishing year is required for "Jordan et al"

Page 20 line 1: How to define the "high NO condition".

Page 20 line 21: No OH exposure is shown in Fig. 6.

Page 21 line 5: Please clarify "Limited experimental statistics"

Page 21 line 30-32: The fragmentation effect on aerosol phase under high OH exposure in OFR should also be considered.

Page 22 line 12: "This generally indicates that we are able to identify the most relevant SOA precursors in the vehicle exhaust." This statement is not true. The SOA yield in SC and OFR was calculated based on a larger group of ArHC than merely OXYL/TOL. The author should calculated the SOA formation based on OXYL/TOL consumption in the SC (and OFR) and yield from OXYL/TOL experiment. Then compare the calculated SOA to the SOA formed in the SC.

Page 24 line 13-14: To conclude this, it is better to plot a graph showing O:C comparison as a function of OH exposure between SC and OFR. The SC shows similar O:C ratios with OH exposure of 1.2*10$^{11}$ molec cm-3 s to the those in OFR under 4.5 molec cm-3 s, which seems not agreeable. The different vapor losses between the SC and OFR might also a reason.

Fig. 2(b) POA point is missing for GDI4-catGPF (CW)

Fig. 5 Better to show the OH exposure range as well.

**References:**

Federer, B., Burtscher, H., Schimidt-Ott, A., and Siegmann, H. C. (1983). Photoelectric charging and detection of ultrafine particles, Atmospheric Environment (1967). 17, 655-657, https://doi.org/10.1016/0004-6981(83)90140-3

Jimenez, J. L., Canagaratna, M. R., Drewnick, F., Allan, J. D., Alfarra, M. R., Middlebrook, A. M., Slowik, J. G., Zhang, Q., Coe, H., Jayne, J. T., and Worsnop, D. R. (2016). Comment on "The effects of molecular weight and thermal decomposition on the sensitivity of a thermal desorption aerosol mass spectrometer", Aerosol Sci Tech. 50, i-xv, 10.1080/02786826.2016.1205728

Krechmer, J. E., Pagonis, D., Ziemann, P. J., and Jimenez, J. L. (2016). Quantification of Gas-Wall Partitioning in Teflon Environmental Chambers Using Rapid Bursts of Low-Volatility Oxidized Species Generated in Situ, Environ Sci Technol. 50, 5757-5765, 10.1021/acs.est.6b00606

Middlebrook, A. M., Bahreini, R., Jimenez, J. L., and Canagaratna, M. R. (2012). Evaluation of Composition-Dependent Collection Efficiencies for the Aerodyne Aerosol Mass Spectrometer using Field Data, Aerosol Sci Tech. 46, 258-271, 10.1080/02786826.2011.620041

Murphy, D. M. (2016). The effects of molecular weight and thermal decomposition on the sensitivity of a thermal desorption aerosol mass spectrometer, Aerosol Sci Tech. 50, 118-125, 10.1080/02786826.2015.1136403

Pagonis, D., Krechmer, J. E., de Gouw, J., Jimenez, J. L., and Ziemann, P. J. (2017). Effects of gas–wall partitioning in Teflon tubing and instrumentation on time-resolved measurements of gas-phase organic compounds, Atmos. Meas. Tech. 10, 4687-4696, 10.5194/amt-10-4687-2017

Palm, B. B., Campuzano-Jost, P., Ortega, A. M., Day, D. A., Kaser, L., Jud, W., Karl, T., Hansel, A., Hunter, J. F., Cross, E. S., Kroll, J. H., Peng, Z., Brune, W. H., and Jimenez, J. L. (2016). In situ secondary organic aerosol formation from ambient pine forest air using an oxidation flow reactor, Atmos. Chem. Phys. 16, 2943-2970, 10.5194/acp-16-2943-2016

Peng, Z., Day, D. A., Stark, H., Li, R., Lee-Taylor, J., Palm, B. B., Brune, W. H., and Jimenez, J. L. (2015). HOx radical chemistry in oxidation flow reactors with low-pressure mercury lamps systematically examined by modeling, Atmos. Meas. Tech. 8, 4863-4890, 10.5194/amt-8-4863-2015

Peng, Z., Day, D. A., Ortega, A. M., Palm, B. B., Hu, W., Stark, H., Li, R., Tsigaridis, K., Brune, W. H., and Jimenez, J. L. (2016). Non-OH chemistry in oxidation flow reactors for the study of atmospheric chemistry systematically examined by modeling, Atmos. Chem. Phys. 16, 4283-4305, 10.5194/acp-16-4283-2016

von der Weiden, S. L., Drewnick, F., and Borrmann, S. (2009). Particle Loss Calculator – a new software tool for the assessment of the performance of aerosol inlet systems, Atmos. Meas. Tech. 2, 479-494, 10.5194/amt-2-479-2009

Ye, P., Ding, X., Hakala, J., Hofbauer, V., Robinson, E. S., and Donahue, N. M. (2016). Vapor wall loss of semi-volatile organic compounds in a Teflon chamber, Aerosol Sci Tech. 50, 822-834, 10.1080/02786826.2016.1195905

Zhang, X., Cappa, C. D., Jathar, S. H., McVay, R. C., Ensberg, J. J., Kleeman, M. J., and Seinfeld, J. H. (2014). Influence of vapor wall loss in laboratory chambers on yields of secondary organic aerosol, Proceedings of the National Academy of Sciences, 10.1073/pnas.1404727111

---

## Referee Comment (RC2) · Anonymous Referee #2 · 7 Jan 2018

This paper describes measurements of primary emissions and secondary organic aerosol from four modern gasoline direct injection (GDI) engine equipped vehicles. The market share of GDI vehicles is rapidly increasing in the US and other countries, displacing the more traditional port fuel injection engine equipped vehicles. This paper represents probably most systematic study of SOA formation from GDI vehicles to date. The paper also investigates the impact of adding a retrofitted gasoline particulate filter (GPF) to two of the vehicles. While others have investigated the effect of this technology on primary emissions, I am not aware of previous studies that have investigated

its impact on SOA formation. The paper also uses an oxidation flow reactor (OFR) and a smog chamber to investigate SOA formation, comparing the results between the two systems and with measurements made with individual compounds.

The paper shows SOA formation dominates primary PM emissions (consistent with previous studies, though the SOA/POA ratio seem much larger than previous studies). The paper demonstrates that the majority of SOA formation is formed from cold-start emissions, which is not surprising but I have not seen it demonstrated before. The paper shows that somewhat more than half of the SOA formation appears due to 8 single-ring aromatics. Finally the paper shows GPF reduces primary PM emissions but does not reduce the SOA formation (or non-methane organic compound emissions).

Overall the paper is well written and very comprehensive. The experiments appear to have been carefully conducted (though I agree with the other reviewer's concerns on treatment of wall loss), with results from repeated experiments shown (it would be nice to describe the precision a bit more). I think this paper makes a nice contribution and recommend that it be published in ACP after addressing the following comments.

Specific comments:

Abstract and a few other places "a large fraction (>0.5)" These statements refer to the mass closure of the SOA production based on measured precursors. 1 is greater than 0.5. The authors need to be more quantitative; e.g. give a range (or some other metric such as median and interquartile range). Figure 6 suggests that this ratio likely varies with OA concentrations. Figure 5 suggests poor closure for 100% and 70% OFR conditions, but good closure for 50% OFR. SC has better closure at short timescales. The authors need to be more quantitative about the mass closure.

Retrofitted GPF – How was this done? How representative is it of how a true OEM designed and installed GPF-system would operate? It is hard to simply add a control system to a vehicle for which it was not designed (I have seen tests with a retrofit GPF hanging off the back of a car!), therefore I am always concerned about how representative the performance of researcher retrofitted system versus what might be done by a vehicle manufacturer. This is not to say that they are not seeing some effect of the GPF but it may be (much) less than the performance of bottom up engineered system. For example, I was surprised that the catalyzed GPF did not further reduce the NMOC emissions – what was the operating temperature of the GPF? Anyways I think the GPF results are interesting, but a bit more detail on how the retrofit was done, specifically limitations of researcher retrofitted systems should be acknowledged. Unless they can document that the retrofitted is representative of OEM designed and installed systems the conclusion section (∼line 10 on page 25) is too strong.

The dramatically higher NMOC emissions and SOA production from cold start is important. Can the authors quantify how much more important it is than hot start (e.g. using an analysis similar to that Saliba et al. EST 2017 10.1021/acs.est.6b06509 to compare hot and cold start emissions).

The conclusion section largely repeats conclusion from earlier in the paper. The paper would be improved if they put the results in context with the growing literature in this area. In particular I was interested if the results are consistent with the existing body of knowledge on SOA formation for PFI vehicle exhaust. My sense is that it is.

You tested the vehicles using two different cycles? Were there any cycle dependencies or was cold start just dominant?

Figure 2 – The y-axes are five orders of magnitude. This illustrates large changes, but changes of a factor of 2 or 3 can also be interesting. For most tests it does appear that the GPF is reducing the POA emissions, but not as dramatically as the EC. I did not get that impression reading the text but it does appear in the figure. More attention needs to be paid to these trends.

Figure 2 – The SOA production seems surprisingly high. For GDI1 the total NMHC (most of which are not SOA precursors) is around 1000 mg/kg (Figure 2a). The SOA production is between 200 and 600 mg/kg (Figure 2a) – the SOA production from the

GPF equipped experiments with GDI1 seem incredibly high. This suggests an effective of SOA of 20-60% of the total NMHC emissions of which less than half is aromatics (Figure 4b is misleading because the NMOC measured by PTR is only 65% of NMHC measured by FID). The SOA production seems higher than previous studies of modern vehicles (they are more similar to 25 year old vehicles). I guess Figure 6 suggest SOA yields are "reasonable", but I was confused looking at Figure 2 (maybe it is just the log scale with 5 orders of magnitude). Are there background issues?

Figure 2c – There is a lot of vehicle to vehicle variability (2+ orders of magnitude). Are the reductions between cold and hot start consistent across vehicles? Plotting ratios may be more informative. What is up with the experiments with an NMHC emission rate of 0.1 mg/kg? Are those valid data?

Figure 2 is very busy (especially panel d). It is basically impossible to sort out the trends. Pick the key points you want to make and plot just that data. The SOA production appears surprisingly close to the NMHC emissions (it even exceeds it for some vehicles).

Additional ArHC (page 23). The analysis in Zhao et al. (EST 50(8): 4554-4563 2016) suggests some of the IVOCs are alkylated single ring aromatics larger than those included in the analysis here. How does including IVOC component measured by Zhao et al. change the analysis? His analysis suggests that IVOCs contribute somewhat less than half of the SOA in gasoline vehicle exhaust.

Page 3 "diesel PM emissions have been greatly reduced." – This is true for new diesel particulate filter (DPF) equipped vehicles but there are a lot of old diesels on the road, especially in Europe so human exposure to diesel particles has probably not yet been greatly reduced. Eventually it will be when the fleet is completely turned over. May want to refine this statement.

Page 4 "modern diesel vehicles" Modern is too generic. You should be more precise catalyzed-DPF equipped diesel vehicles. I don't necessarily think modern = DPF.

Page 8 "experiment. Control experiments were conducted regularly in the SC to estimate the contribution of the SC background to SOA formation." Please provide another sentence or two here that describes results from control experiments. How much SOA was formed in controls and how does it compare to what is measured in an experiment with vehicle exhaust. Did you run control experiments with the OFR – what were the background levels in that system?

Section 2.2.6 – I am pretty sure that you are calculated yields using the reacted aromatic mass in the denominator however this statement is confusing "the ratio of the SOA mass to the reacted SOA-forming mass, delta_NMOCreacted" My understanding is that delta_NMOCreacted is not the same as the reacted aromatic mass. This needs to be cleaned up to avoid confusion. May also want to state this in the caption for Figure 6 to reminder reader of how yields are calculated.

Using "NMOC" to describe the sum of the PTR measurements is confusing as it is measuring less than 2/3rds of the organic gas emissions as measured with FID. This limitation needs to be stated more clearly (it is in the intro but the reader will likely forget – e.g. adding to caption of Figure 4 would be good and in other places in the main text when you discuss NMOC.

Emissions data from tests in mg/kg-fuel needs to be provided in tables in supplemental.

---

## Author Comment (AC1) · 15 Apr 2018

This paper evaluates the gas phase composition and secondary organic formation from gasoline vehicles in a batch and flow reactor. Excusive results about primary emission factors, gas vapor composition and SOA formation are shown. This paper is well written and organized. The paper describes a large amount of data, but critical evaluation and analysis is missing, that is needed to have confidence on the quantification of results. Therefore I recommend that the paper may be published in ACP after addressing the major revisions below.
**Author Response:** We agree with referee 1 that our manuscript describes an extensive data set on GDI vehicles with novel after-treatment systems, including a comprehensive analysis of the gas- and particle phase, as well as the SOA formation. We are confident, that our manuscript at its initial stage includes an extended discussion of experimental uncertainties in the main text and provides additional critical evaluation in detail in main text and supporting information. We provide answers to RC1 and modifications to our manuscript to the best of our abilities.

**Major revisions:**

**RC1-1:** 1) The analysis of vapor losses to walls is inadequate. Enormous progress has been made on this area recently, and there is no excuse to ignore those corrections in current studies. Comparability with past studies that were performed when vapor wall losses were not understood is not an excuse to ignore this major issue. Comparability with future studies, for which all the good ones will include analysis of this effect, should be the relevant criterion. It will benefit the citation of this paper for showing both results. This is an essential correction which could introduce large bias (a factor of 3-4 in Zhang et al. 2014) and is related to multiple key calculation including SOA yield, NMHC composition etc. Taken vapor loss corrections into account, comparison results can be more accurately assessed, especially crucial for some main focuses of this paper: SOA yield between SC and OFR, SOA yield of vehicle exhaust vs single precursors. As the authors stated in page 21 line 27-29 "we expect both SC and OFR yields to be underestimated, by factors of approximately 1.5-2 (SC) and 1.25 (OFR) ((Platt et al., 2017);(Zhang et al., 2014);(Palm et al., 2016)) due to vapor wall losses. Corrections would reduce the discrepancy between the two systems." Thus a gas vapor wall loss correction is needed for this study. The model of Krechmer et al. (2016) can be used for the SC, and has been recently shown ((Ye et al., 2016); also at AAAR 2017) to be consistent among all Teflon chambers. The model of Palm et al. (2016) should be applied to the OFR.
**Author Response:** We agree with the need to address vapor wall losses in smog chamber (SC) and oxidation flow reactor (OFR) studies. For this reason, we have provided estimates for vapor wall losses for both systems in the initial version of the manuscript, as the referee mentions in the second part of this comment. We believe that our transparent approach to provide the data along with the expected correction factor which we have determined for this specific smog chamber, using gasoline vehicle emissions in our related publication (please refer to Platt et al., 2017), is a valid option, as dedicated experiments to study the exact losses during the presented experiments have been missing. It would be misleading to apply correction factors from literature. We would like to note, that the correction factor determined in Platt et al., 2017 (1.5-2) is in line with reports by others (e.g. Zhang et al., 2014, reports a factor of 1.1-4.2 underestimation, La et al., 2016, found a factor of 1.1.-6). Generally, we disagree that gas/vapor loss corrections are transferrable from one chamber to the next
* * *
without specific characterization experiments, and don't see this as a conclusion in previous literature reports either (see Krechmer et al., 2016).

Regarding the OFR, the model from Palm et al., 2016, has been applied in the initial version of our manuscript and the corresponding correction factor is already stated in the manuscript.

Loss corrections on our primary NMOC composition are not needed: The compounds of interest for SOA formation which we were able to identify by PTR-ToF-MS in our emissions mix are mainly BTEX, C3-Benzenes and Naphthalene. Based on their saturation mass concentration, these compounds are classified as VOCs or correspond to the upper end of the IVOC range, and are therefore not expected to be impacted significantly by losses to SC walls on a time-scale relevant to our SOA study. We performed measurements with emissions containing these substances to test their loss to chamber walls without oxidation chemistry initiated, and have monitored the aromatic composition and absolute concentrations of our identified species over the course of several hours: it did not change significantly enough to impact our findings, i.e. during experiments lasting between 2 and 6-8 hours the stability of relative gas phase composition (determined as the change in the ratio of relevant SOA-precursors to benzene) changed only by 3% (after 2 hours, which is a typical time-scale for the presented experiments) to 7-12% (at 6-8 hours).

**Text modifications:** We have modified the section (2.2.5) to include also additional references to newer literature and report the possible spread for SOA yield underestimations better. The new text reads as follows:

- *"a robust strategy for their determination and correction remains challenging (Krechmer et al., 2016). In our previous work, we estimated that vapor wall losses may cause SOA yields to be underestimated for the SC used herein (assessed based on gasoline vehicle exhaust SOA, see Platt et al., 2017, by a factor 1.5-2 for our experimental conditions), comparable to suggestions by others, e.g. a factor of 1.1.-4.2 by (Zhang et al., 2014) and 1.1-6 (La et al., 2016). Data correction would increase SOA yields on average by a factor 1.5-2."*
- *"Given the high SOA concentration and hence particle surface ($(1-5)x10^9$ $nm^2$ $cm^{-3}$ based on the SMPS size distribution of SOA), at least 80% of the formed LVOC was calculated to partition to the pre-existing OA mass based on the model by Palm et al., 2016. Data correction would increase SOA yields by a factor of 1.25 on average."*

**RC1-2:** 2) The comparison on SOA formation between OFR and SC should also consider gas vapor loss in the tubing (Pagonis et al., 2017). The different tubing lengths and materials can also result in a SOA yield difference between SC and OFR. A vapor loss calculator can be found in Pagonis et al. (2017). Please clarify the tubing material and length as well. Page 16 line 17-18: In addition to the residence time of OFR, this SOA delay was also possibly caused by the delayed gas vapor in the tube, as suggested in Pagonis et al. (2017).

**Author Response:** We agree that this should be specified.

Tubing material and length: Tubing to sample direct emissions from the vehicle tailpipe for a) injection into the SC or online-OFR, or b) direct gas-phase measurements are made of SilcoTek®-coated steel (12 mm diameter), temperature controlled at 140°C, operated under high flows (30 L min$^{-1}$), and of roughly 8 m length. Ejector dilutor 1 is placed in a temperature controlled housing (200°C), and ejector dilutor 1 is operated at 80°C. Instruments sampling either a) from the SC or b) behind the OFR, or c) directly, are connected via separated tubing for gas-phase and particle phase. Particle-phase tubing is exclusively made of stainless steel, no copper tubing is used. Sampling lines are of 6 mm diameter, and up to 2 m length. Support pumps are used at the instrument inlets, to minimize sampling residence time by increasing the flow rate. Similar approach is used for gas-phase sampling (total tubing length to reach all of the instrument inlets, which are also equipped with support pumps is up to 2 m). Tubing for gas-phase sampling is made of either SilcoTek®-coated steel or Teflon, temperature controlled at 60°C where necessary (i.e. for PTR-ToF-MS measurements and FID), to avoid losses of the VOC and IVOC species relevant to our SOA-study and PTR-ToF-MS analysis (including essentially BTEX, C3-Benzenes, Naphthalenes, and eventually phenolic compounds, benzaldehyde). SilcoTek®-coating and Teflon are also suitable for the sampling of species known to be easily retained on surfaces, such as formaldehyde, acetic acid, acetaldehyde, for which, in addition to the uncertainties of PTR-ToF-MS analysis, also tubing losses may induce a slight shift in our gas-composition analysis.

We are confident that the differences in SOA yields obtained in the SC and OFR are not caused by losses on sampling lines between the SC and OFR. This sampling system was made of a combination of SilcoTec® coated steel with carbon-coated Teflon (i.e. electrically conductive Teflon suitable for simultaneous gas- and particle phase sampling), and the total length between SC and OFR inlet was roughly 35 cm (6 mm diameter, ca. 8 L min$^{-1}$ flow). Additionally, all measurements from the dark smog chamber (which is the basis for gas-phase composition data), were performed for at least 10-15 minutes, to reach a stable signal. We experimentally determined potential losses in the sampling from dark smog chamber and sampling through dark OFR and observed a reduced mass of species by less than 5% and no change in the composition of the SOA precursors, which, in our case, is the determining factor.

Page 16, Line 17-18: Our statement is in line with findings of other researcher (Zhao et al., 2018), who state "After the vehicle was turned off at the end of bags 2 and 3 it took approximately 3 min for the OA signal at the PAM reactor outlet to return to background levels. This delay reflects the time it takes for the exhaust to pass through the PAM reactor.", in analogy to our interpretation. However, we do generally agree that delays due to retention on sampling lines can cause a shift in the signal, and hence add this also specifically to our manuscript.

**Text modifications:** Tubing material, length, temperature and flow rates are specified in the SI as follows:

- *Tubing to sample direct emissions from the vehicle tailpipe for injection into the SC or online-OFR, or direct gas-phase measurements are made of SilcoTek®-coated steel (12 mm diameter), temperature controlled at 140°C and operated under high flows (30 L min$^{-1}$) to avoid substantial losses over the sampling length of roughly 8 m. Ejector dilutor 1 is placed in a temperature controlled housing (200°C), and ejector dilutor 1 is operated at 80°C.*

- *Instruments sampling either from the SC, behind the OFR, or directly from the dilution system are connected via specific tubing for gas-phase and particle phase. Particle-phase tubing is made of stainless steel (6 mm diameter), and up to 2 m length. Support pumps are used at the instrument inlets, to minimize sampling residence time by increasing the flow rate. Total tubing length to reach all of the gas-phase instrument inlets, which are likewise equipped with support pumps is up to 2 m. Tubing is made of Teflon or SilcoTek®-coated steel. The sampling line of the PTR-ToF-MS instrument and FID is temperature controlled at 60°C.*

- *SilcoTek®-coating and Teflon are also suitable for the sampling of species known to be easily retained on surfaces, such as formaldehyde, acetic acid, acetaldehyde, for which, in addition to the uncertainties of PTR-ToF-MS analysis, also tubing losses may induce a slight shift in our gas-composition analysis.*

- *The sampling system between the SC and OFR (for OFR-from-SC experiments) was made of a combination of SilcoTec® coated steel and conductive Teflon tubing, suitable for simultaneous gas- and particle phase sampling. The total length between SC and OFR inlet was roughly 35 cm (6 mm diameter, ca. 8 L min$^{-1}$ flow). Additionally, all measurements from the dark SC batch sample were performed for at least 10 minutes, to reach a stable signal."*

Signal delay was adressed as follows:

*"The latter is related to a delay of the OFR signal by the residence time in the reactor, as also observed by others (Zhao et al., 2018), and might potentially also be caused by a delay of SOA forming species which are retained on surfaces (Pagonis et al., 2017)."*

**RC1-3:** 3) The particle losses due to heating in the sampling line and the hotter temperature in SC when UV light is on should be addressed. The aerosol loss due to tubing (under no heating condition) needs to be estimated as well. The model for aerosol loss calculation in the tubing can be found in (von der Weiden et al., 2009). The particle loss due to heating can be experimentally determined.

**Author Response:** Particle losses in the SC are assessed with our data as presented in our initial manuscript, which takes all factors leading to particle losses, including temperature effects into account.

**Text modifications:** "The main losses of particles are due to (1) diffusion, (2) electrostatic deposition and (3) gravitational settling, *which are in turn affected by temperature changes due to the UV lights.* Wall losses of particles in the SC were accounted for using the method described in (Weitkamp et al.,

2007) and (Hildebrandt et al., 2009), *which accounts for all these loss processes simultaneously, including the aforementioned temperature effects.*"

**RC1-4:** 4) Page 10 line 20-30: The SOA photolysis in OFR should be considered as well. For example in OFR with full UV light setting (100%), half of the SOA from toluene SOA (OH chemistry) or naphthalene SOA (OH chemistry) can be photolyzed under 254 UV light with a low quantum yield of 0.1, as shown in Fig. 8b in (Peng et al., 2016). This photolysis effect on SOA formation and SOA yield calculation under 50% and 100% UV light setting should be considered.

**Author Response:** Non-OH losses of SOA precursors via photolysis within the OFR was assessed for all our experimental conditions (OFR-from-SC, online-OFR 2014 and online-OFR 2015) as described in the discussion manuscript (section 2.2.5 and the corresponding information in the SI).

We summarize our findings and implications here: Given that non-OH losses strictly only imply a non-OH induced reaction of the compounds, but do not rule out any SOA formation from the obtained reaction/photolysis products, no corrections for any non-OH reaction can be made in our view (as already stated in our manuscript) regarding the formation of SOA. However, we agree to note specifically that the observed SOA may not only be related to OH-induced aging but also to UV-induced reactions that produce SOA, (in addition to potential $O_3$-induced SOA, as well as $NO_3$-induced SOA). Our discussion paper contains this information already in the SI (page 3, line 5) as follows *"This only refers to the reactive interaction of OH vs. the excitation by UV, and does not allow conclusions on the formation of SOA. Also chemistry initiated by UV185 or UV254 may lead to the formation of SOA. Additionally, it does not suggest any conclusions about the interaction of O3 with double bonds made available by first ring-opening reactions." (...) "Potential effects of O3 on first generation products are not taken into account. Under those diluted conditions (initial NO < 100 ppb), we regard the experiments in OFR as low NO conditions (Peng and Jimenez, 2017). The dominant SOA precursors found in the exhaust are not reactive towards NO3 radicals that can be formed in the OFR; potential effects on first generation products are not taken into account."*

In this specific comment (RC1-4), the referee addresses in addition the destruction of previously formed SOA by photolysis, which we have not previously addressed in our manuscript. We address this here: Our OFR photon-flux is $(1.2-2.3) \times 10^{15}$ photons $cm^{-2}$ at 185 nm and $(1.7-2.9) \times 10^{17}$ photons $cm^{-2}$ at 254 nm (100% UV setting, which is the maximum and hence inducing maximum photolysis impact). Those photon-fluxes can be considered "medium" if comparing to Peng et al., 2016, Figure 8. The estimated SOA-photolysis is <1% for naphthalene at a quantum yield of 0.1 and <5% at a quantum yield of 1, for 185 nm. For 254 nm, the estimated SOA-photolysis is around 20% at a quantum yield of 0.1, but reaches 60-80% if a quantum yield of 1 is assumed, according to Figure 8 in Peng et al., 2016. Significant uncertainties are associated with this assessment: 1) we do not have any precise information about photon fluxes of our OFR, 2) dependent on the assumption of the quantum yield, we obtain a result which ranges from insignificant to significant, 3) as a full description of the chemical composition of the SOA is missing, it is not clear which effects photon-interactions will induce and whether photolytic destruction of molecules will lead to a loss of SOA mass and to which extent; hence, this assessment is at the current point in time of speculative nature only. Future work should address this in further detail before any corrections can be applied. Our conclusion is in line with Peng et al., 2016, who state: "Thus, to our current knowledge, lack of solid information on quantum yields of SOA components with multiple carbonyls and hydroxyls at 254nm prevents a clear assessment of SOA photolysis in OFRs at the medium and high UV.

**Text modifications:** *"... Also chemistry initiated by UV185 or UV254 may lead to the formation of SOA, and likewise photons may also lead to the destruction of OH-formed SOA; both processes deserve attention in future research."* was added to the corresponding SI section, and "*The results and implications of photon-induced effects on SOA formation or destruction are discussed in the SI.*" was added to the main text.

**RC1-5:** 5) Page 14 line 1-2. How did the mass quantification between AMS and SMPS compare? It is essential to document this comparison in the form of scatterplots and regressions. Collection efficiency (CE) vary with chemical composition and aerosol phase (Middlebrook et al., 2012). CE for AMS quantification should vary in this study since relative OA and NO3 fraction in total aerosol changed a lot. Why do the authors choose a CE ~1 here. I would expect a slight variation on RIE

since there were POA dominated periods during the studies, for which RIE may be higher (Jimenez et al., 2016; Murphy, 2016). Attention needs to be paid for the size cut differences between AMS and SMPS as well, when nucleation was happened.

**Author Response:** CE: Typical assumptions for ambient aerosols assume CE=0.5, which is related to large extents to the fact that ambient aerosol contains ammonium sulfate and solid or glassy organics, which lead to significant bounce on the AMS vaporizer. Ammonium-Nitrate and Organic Aerosol mixtures are not expected to bounce significantly, and hence we have used a CE=1. Middlebrook et al., 2012, observed a CE close to 1 (0.8) for ambient aerosol containing ammonium nitrate, but no ammonium sulfate.

RIE: In principle, the RIEs of organic material may vary with the molecular weight of the parent molecule. However, most compounds undergo extensive thermal decomposition in the AMS and ionize as much smaller molecules, which have similar ionization cross-sections and thus similar RIEs. In the current study, the observed POA is dominated by aliphatic hydrocarbons, which have been shown in other studies to have RIEs of approximately 1.4, the same as SOA (Jimenez et al., 2016). For example, there is no significant difference in the decane, diesel fuel, and lubricating oil despite a factor of 2 differences in molecular weight. Therefore we do not expect significant differences between POA and SOA RIEs in this study. Further, even if the POA and SOA RIEs were different, the very low POA/SOA ratios observed in this study would prevent a significant bias in the results, even if the POA RIE was as high as that suggested by Murphy et al., 2016 (~4 assuming the molecular weight of lubricating oil), which we note again would be in conflict with experimental evidence (Jimenez et al., 2016), our conclusions won't be affected. An RIE of 1.4 for SOA remains reasonable, and hence, any conclusions on SOA yields are neither effected by applying the standard assumption of RIE=1.4 to our data set.

Lens-cut-off/Nucleation: AMS pToF size distributions are described in the SI. As provided in the main text, no lens-transmission analysis was performed.

**Text modifications:** The new text reads as follows:

*"We used a collection efficiency of 1, as upon photochemistry, significant amounts of $NH_4NO_3$ were formed, and under those $(NH_4)_2SO_4$ –free conditions, our aerosol mixture is not expected to bounce significantly. No corrections for lens transmission were performed, pTOF distributions are provided in Figure S10."*

**RC1-6:** 6) Schematic of the sampling strategy is confusing. I did not get the timing for the sampling strategy. Was the UV light setting is constant during vehicle testing cycle? Then the background of aerosol and gases under UV light is off in SC and OFR was obtained by repeating the testing cycle of the cars? Sufficient detail needs to be provided, that would enable someone else to repeat the experiments, as it is standard in scientific publications.

**Author Response:** We have provided a lengthy (9 pages) description of our experimental set-up, strategy and conditions on Page 5-14 of the discussion paper (section 2 "Experimental"), which we believe allows for a full repeat of our experiments, specifically, this is true for our detailed description in section 2.2.1. Additionally, we refer to Platt et al., 2013 and 2017, in which we have published our experimental set-up earlier in further detail. Further, our photochemistry sampling scheme is discussed in section 3.5, along with representation of a typical experiment in Figure 5. We provide further information on background levels in our answer to RC2 and RC1-9 below, and have added additional information on control experiments to the text (see RC1-9). We have clarified this lengthy description to avoid confusions.

In the following our specific explanations:

- SC sampling: Background measurements were conducted prior each experiment and emissions were injected thereafter during a full cold-started test cycle or selected phases thereof. During this injection phase, emissions in the SC were only monitored with a limited set of experiments. The mass spectrometers for gas- and particle phase characterization were monitoring the direct emissions sampling (online) and online-OFR instead. When the cold-started emissions test was completed, mass spectrometers were set to characteriz the emissions sample collected in the SC for about 15 minutes; thereafter, the hot-started vehicle test was conducted for which instruments were again disconnected from the SC, to monitor

- the online OFR and direct emissions instead. Meanwhile, no photochemistry was initiated in the SC.
- OFR-from-SC (also "batch-OFR")¨
  OFR-from-SC experiments refers to experiment that included aging a sample of emissions in the OFR reactor at fixed conditions, with UV lamps set to either 50, 70 or 100% intensity and with UV off (dark). Emissions which were pre-collected in the dark SC used as a buffer/storage volume were sampled via a short sampling system described in detail within answer to RC1-2 from the SC into the OFR. Sampling was conducted until the SOA formation in the OFR reached a stable signal, and conditions were kept stable for several minutes from this point onwards. Prior and after UV on conditions, fresh emissions (OFR UV off as well as directly in the SC) were characterized to allow for estimating the POA contribution to the OA mass measured under UV on conditions, as well as to allow for calculating the reacted SOA precursor mass. OFR cleaning with clean air under UV on conditions was performed prior to the measurements to minimize any background contamination.
- Direct emissions sampling (online): PTR-ToF-MS data were collected at the inlet of the OFR. These data were used for selected experiments (e.g. data on GDI4 in 2015, labelled "online"). For all other vehicles and experiments we present the average composition and emissions factors from samples that were collected in the SC (w/o photochemical aging). Essentially, integrating the direct emissions sampling (online) vs. the composition determined in the SC yielded comparable results (Figure 4, GDI4, SC vs. online).
- Online-OFR: yes, here we have tested vehicles up to 4 times with cold-started and 4 times with hot-started experiments (as stated in Table 1). Only one cold-started vehicle test could be conducted each day, hence, typically 3 cold-started vehicle tests were performed on 3 consecutive days with UV lights switched on in the OFR, and 3 hot-started vehicle tests with UV lights switched on. One a 4[th] experiment day, we conducted the same set of experiment with OFR- UV lights switched off, to determine primary OA emissions and calculate the SOA. Essentially, much less POA was found then SOA (please see our online-OFR experiments, one of which is provided in the main text (Figure 3), as well as the corresponding figures in the SI). Note, however, that only for selected experiments (GDI4 in 2015), these online-OFR data were used in a quantitative way. For all other vehicles and experiments, we rely on SC and OFR-from-SC data (regarding SOA), as well as batch sampled experiments for primary NMOC composition. Essentially, integrating the online OFR-SOA for GDI 4 yielded comparable results to the SC (Figure 2b, GDI-4 SC vs. online).

**Text modifications:** We have made small modifications to shorten and clarified this section which has become more logic to facilitate experiment repeats now; information on backgrounds and control experiments was added as described in answer to RC1-9.

**Other revisions:**

**RC1-7:** Page 5 Line 8: EDC was defined as "older" low-road European Driving cycle (EDC), which is inconsistent with the definition of "New" EDC in the abbreviation/definition list.
**Author Response:** We have made the modifications to the abbreviations list, the modified text is provided in the following. The so-called "New European Driving Cycle", is by now the "older" cycle and overtaken by the recent "WLTC", which is, why nowadays, the NEDC is referred to as EDC, although the strict definition is "NEDC".
**Text modifications:** *"EDC =European Driving Cycle (previously known as the "New European Driving Cycle")"*

**RC1-8:** Page 7 line 26-27: Please specify the dilution factor for smog chamber (SC) and oxidation flow reactor (OFR).
**Author Response:** Dilution factors are provided in main text (Figure 1) and SI for our experimental set-up in the initial version of the manuscript. The final dilution value for the SC is determined by the volume sampled during a driving cycle. The decisive characteristic of a SC experiment are the concentration levels, rather than the dilution ratio, which are provided in Table S4-S7 of the initial SI.

**Text modifications:** Adjustments were made in section 2.2.2 and 2.2.3 to add the specific values for SC and OFR. "*Concentration-levels of our SC experiments were representative for urban ambient conditions, as reported in Table S4-S7.*" and "*…the diluted exhaust (either 1 or 2 ejector dilutors, each at a dilution ratio of 1:8)...*"

**RC1-9:** Page 8 line 14-15: What is the aerosol background level in SC and OFR experiments with clean air when the UV light is on? If it is high, this background needs to be subtracted in the calculation of formed SOA mass concentration.

**Author Response:** The background was generally low (with UV off, as well as UV on, and the SOA background was insignificant compared to our typical experiments ($<1$ and $<2$ ug m$^{-3}$ for SC and OFR respectively, during SC and OFR-from-SC experiments; $< 10$ ug m$^{-3}$ during online OFR experiments (compared to $100 - 2000$ ug m$^{-3}$ SOA formed during these experiments, this is not significant). See also our response to RC-2. As we noted in the text, however, Ph2-4 experiments in the SC were close to the background measurements with the PTR-ToF-MS and hence we have noted this specifically in the discussion paper (figure caption of figure 4 "*Note that the total NMOC levels for Ph 2-4 (cW) are about 1/10 of full cW and Ph 1 (cW) concentrations only and measurements are close to the background measurements (signal not significantly different from 3 standard deviations of the background measurement*").

**Text modifications:**
- *Background measurements of the clean chamber were conducted prior to each experiment, and was insignificant compared to our measurements except for Ph2-4 or GDI4 experiments as stated in the results. Photochemistry control experiments were conducted regularly to estimate the contribution of the SC background to SOA formation; these experiments were conducted after the standard cleaning procedure. Instead of vehicle exhaust, pure air was used as a sample and ammonium sulfate (50 ug m$^{-3}$) injected as seed. Other experimental procedures were in line with the typical vehicle experiments. We found a SOA background of $< 1$ µg m$^{-3}$, which is below the SOA concentrations formed during vehicle exhaust aging Concentration-levels of our SC experiments were representative for urban ambient conditions, as reported in Table S4-S7.*"
- "*Background levels were $<2$ µg m$^{-3}$ SOA before OFR-from-SC experiments (when sampling from cleaned SC) and $<10$ µg m$^{-3}$ when sampling diluted (1:8) test bench room air prior online-experiments.*"

**RC1-10:** Page 8 line 26-29: Figure S1 shows the aerosol and gas-phase species are sampled through the same tube in the center of OFR. If it is true, there will be large loss either on VOCs species (if stainless or copper tube is used) or on aerosols (if Teflon tube is used)

**Author Response:** We described our sampling system and tubing materials, length and flows in our response to RC1-major revisions, and have discussed implications for losses of the species of interest within our study in this response. We kindly ask the referee and editor to refer to this section.

**Text modifications:** As provided in "RC1-major revisions", we have added a description in the SI.

**RC1-11:** Page 9 line 26-27: The particle loss can be determined by measuring the aerosol concentration before and after OFR when the UV-light is off. The aerosol concentration before the OFR can also be roughly determined with aerosols in the SC chamber if the dilution factors and volatility of OA are known in OFR and SC. There have also been some reports from FIREX (Jesse Kroll's group) that particles containing BC can be charged by the UV lights and be lost much faster, also see (Federer et al., 1983). Was this effect evaluated?

**Author Response:** The referee refers to a sentence where we state that particle losses in the OFR were evaluated by comparing eBC concentrations before and after the OFR during experiments. We find this test to provide a more realistic evaluation of the OFR performance than the lights-off tests suggested by the referee due to differences in temperature and potential losses due to charging of BC particles by UV light, as suggested by the referee. As noted in the original manuscript, observed eBC losses were negligible, consistent with previous characterization of this OFR for similarly-sized particles (Lambe et al., 2011).

Although we have not specifically investigated the effects of UV charging, the fact that

overall BC losses are negligible across the OFR suggests that this loss process is also negligible for our experimental set-up.

**Text modifications:** No major adjustments were needed, we have modified wording to highlight that we are using an experimentally determined transmission under actual operating conditions ("in-situ"). The section reads now as follows: *"A comparison of eBC mass before and after the OFR indicated no significant losses during UV on or UV off periods (experimentally determined transmission was equal to 1).Consequently no further correction was applied. Particle wall losses in the OFR have been quantified previously by Lambe et al., 2011, who reported at least 80% transmission efficiency through the OFR for particles of mobility diameter ($d_m$) > 150 nm (Lambe et al., 2011). The particles measured behind the OFR in our study had a median vacuum aerodynamic diameter ($d_{va}$) between 200-400 nm based on HR-ToF-AMS measurements (size distributions are provided in Figure S9), which correspond to $d_m$ > 150 nm when assuming spherical particles and an OA density of 1.2 g $cm^{-3}$ (Turpin et al., 2001) supporting our experimentally determined transmission efficiency. "*

**RC1-12:** Page 10 line 16: "1000-5000 $nm^2$ $cm^{-3}$" Is this unit true? These values indicate the particle surface areas in this study are very small, which is inconsistent with mass values reported in Fig. 3. It is ~10^6 times less than those in the typical chamber studies, e.g. (Zhang et al., 2014).

**Author Response:** We thank the referee for spotting this typo. The values should indeed read (1-5)x$10^9$ $nm^2$ $cm^{-3}$. The particle surface area noted here corresponds to the OFR-from-SC aging experiments (i.e. an average SOA mass value of 100 µg $m^{-3}$), and no primary eBC seed (b/c of GPF installation) was present. We have revised the corresponding section to clarify this point further.

**Text modifications:** We have corrected the stated values to read (1-5)x$10^9$.

**RC1-13:** Page 11 line 19-24: Does the OH exposure estimated from BuOH-D9 agree with the OH exposure calculated based on Peng et al. (2015)? A plot showing the comparison of these two methods will beneficial for readers to understand how accurate of the OH exposure used here.

**Author Response:** The comparison requested by RC1 can be found in the SI of the current version of the discussion manuscript, as also highlighted in the main text. We copy-paste the information here:

*"Based on these input parameters, the model (Peng et al., 2016) predicted an $[OH]_{exposure}$ (OH concentration integrated over time, see discussion in main text "OH exposure estimation", in molec $cm^{-3}$ s) in the OFR of*
*UV100%:      $[OH]_{exposure}=(10-13)x10^{11}$*
*UV70%:       $[OH]_{exposure}=(2.4-3.1)x10^{11}$*
*UV50%:       $[OH]_{exposure}=(0.35-0.48)x10^{11}$.*
*The estimated $[OH]_{exposure}$ (in molec $cm^{-3}$ s) and OH concentration (in molec $cm^{-3}$), [OH], based on the experimental measurements of the decay of BuOH-D9 correspond instead to*
*UV100%:      $[OH]_{exposure} =(3.0-5.8)x10^{11}$, i.e. [OH]= (2.7-5.2)x$10^9$*
*UV70%:       $[OH]_{exposure} =(1.6-2.5)x10^{11}$, i.e. [OH]=(1.4-2.2)x$10^9$*
*UV50%:       $[OH]_{exposure} =(0.31-0.49)x10^{11}$, i.e. [OH]=(0.28-0.44)x$10^9$"*

It appears that the model is able to re-produce our experimental measurements correctly at lower UV-intensity of the OFR reactor (i.e. the same order of magnitude is achieved), but fails to predict the OH exposure correctly at the higher UV intensity, where it over predicts the OH exposure). We believe the experimentally determined tracer-based method (d9-BuOH here) is more accurate because it is specific to the current system whereas the model is a generalized parametrization that here uses only the $O_3$-concentration as an input to determine the photon flux of the UV lamps. Therefore the tracer-based method is used throughout the manuscript and we strongly encourage other users of the OFR reactors to supply a proper tracer (such as BuOH-D9, see Barmet et al., 2012) for the experimental in-situ determination of the OH exposure whenever possible.

[Figure]

**Figure R1-1.** OH exposure predicted by the Peng-model vs. our experimentally determined OH exposure (data as stated in the SI). This plot was not added to the SI, as the data are compared in the text.

**Text modifications:** In the main text we have added the following: "T*he tracer-based OH exposure calculations are generally in good agreement with exposures predicted by the model, except at the highest OH exposures where the tracer method is approximately a factor of 2 higher. Tracer-based OH exposures are used throughout this analysis, as these measurements are specific to the current OFR system.*"

**RC1-14:** Page 12 line 2: Please specify the dilution factor.
**Author Response:** The dilution ratio in the CVS-dilution tunnel is variable and was controlled by means of the CO2-analysis, and is at a range of 8 during high engines loads to 30-40 at idle conditions.
**Text modifications:** We added the information as described above.

**RC1-15:** Page 12 line 5-7: Why heat the sample before CPC? Why 300 oC.
**Author Response:** These settings are based on the PMP- Particle Measurement Program of the ECE GRPE Group; thermo-conditioning is a pre-requisite in order to measure only non-volatile particles.
**Text modifications:** We noted that we are not presenting any data from the CPC instrument in our manuscript and have therefore removed this statement from the main text, however.

**RC1-16:** Page 14 line 19-24: These descriptions cannot be found in the Fig. 1 e.g. No graph compares "Ph 1 of cW and hW vs. Ph 2-4 of cW and hW" in line 20.
**Author Response:** We are unable to follow the referee's argument. As we noted in the text, these comparison discuss the data presented in Figure 2, panel a and c. Specifically, the NMHC comparison for Ph 1 of cW and hW vs. Ph 2,3 and 4 of cW and hW are derived by looking at the data presented in Figure 2 panel c, and are discussed in the text.
**Text modifications:** No modifications were made specific to this request. However, we have revised Figure 2 along with suggestions by Referee 2 and have hence modified parts of section 3.1, which the Referee 1 refers to here; please refer to our response to RC-2.

**RC1-17:** Page 15 line 2-3: It is hard to draw such a conclusion based on Fig. 2b. A scatter plot between POA+BC vs PM is needed or at least please give the value POA+BC.
**Author Response:** We agree with the referee's argument, and have added a comparison plot in the SI to support our statement "*PM measured in the batch samples (sum of eBC and POA, Figure 2b) compares generally well with the gravimetric PM analysis of filters sampled from the CVS (Figure 2a).*"
**Text modifications:** As noted above, we have added an additional Figure to the SI and refer to it in the main text. *"PM measured in the batch samples (sum of eBC and POA) are compared with gravimetric PM analysis of filters sampled from the CVS in Figure S16."*

[Figure]

**Figure R1-2:** Comparison of POA+eBC from batch SC sampling to gravimetric PM measurements from filter samples taken from the CVS, added to the SI; the figure caption reads:
*"POA and eBC measurements in the SC batch sample compared to gravimetric PM measurements from the CVS."*

**RC1-18:** Page 15 line 9: Please give the value ranges for "previous finding".
**Author Response:** This information is provided in the current version of the discussion paper few lines further down ("*median 60, range ~10-400 mg kg$^{-1}$ $_{fuel}$*"), i.e. on Page 15, line 12.
**Text modifications:** We have revised this paragraph to remove redundancies and make the range of previous findings easier to grasp for the reader, along with suggestion by RC-2.

**RC1-19:** Page 15 line 16-19: A smaller vapor loss in the OFR is also a possible explanation.
**Author Response:** We agree with the referee (as also indicated by our provided vapor correction factors 1.25 for the OFR and 1.5-2 for the SC) and have revised the statement.
**Text modifications:** As described above, we have put a reference to the subsections which discuss those issues at other locations within the manuscript.

**RC1-20:** Page 18 line 5: publishing year is required for "Jordan et al"
**Author Response:** Thanks for the hint; the reference was corrected.
**Text modifications:** "*Jordan et al.*" was revised to "*Jordan et al., 2011*".

**RC1-21:** Page 20 line 1: How to define the "high NO condition".
**Author Response:** The answer to the definitions is provided in the publication (Peng and Jimenez, 2017) which is cited along with above statement. We have clarified this now in our manuscript .
**Text modifications:** The statement *""high NO" conditions may be reached in the OFR ((Peng and Jimenez, 2017))."* was modified to read now *""high NO" conditions may be reached in the OFR as defined by (Peng and Jimenez, 2017)."*

**RC1-22:** Page 20 line 21: No OH exposure is shown in Fig. 6.
**Author Response:** We refer to the OH exposure data which are noted in the caption to Figure 6. We have clarified this statement.
**Text modifications:** The statement *"...OH exposure data at the end point of SC experiments and for the OFR are provided in Figure 6 and Figure 7)."* was revised and reads now *"OH exposure data at the end point of SC experiments and for the OFR are provided in caption to Figure 6, and Figure 7)."*

**RC1-23:** Page 21 line 5: Please clarify "Limited experimental statistics"

**Author Response:** We conducted 2 experiments with Ph2-4 emissions aged in the SC (1 with GDI1 with standard configuration and 1 with GDI1 equipped with a GPF). Additionally, Ph2-4 SOA-precursor emissions collected in the SC were close to background concentrations (as already discussed within this answer to referees as well as noted in caption to Figure 4). Therefore, the available data are not sufficient to allow a reliable SOA yield analysis for Ph2-4 analysis. Others (Zhao et al., 2018) have recently published a SOA yield comparison from an OFR data set on cold- and hot-engine emissions, and also discuss the potential background effects in their publication.

**Text modifications:** We have re-adjusted the main text, to read as follows: *"Data are presented as a function of suspended OA for all experimental conditions of cold-started GDI1-3 (i.e. for full cW, cE; and Ph 1 (cW)), while GDI4 or hot engine conditions, i.e. Ph 2-4 (cW) are not included in our the analysis, as this data set includes only two experiments with concentrations levels close to our background measurements; a discussion of SOA yields from cold- and hot-engine emissions has recently been published by Zhao et al., 2018 for an OFR data set)."*

**RC1-24:** Page 21 line 30-32: The fragmentation effect on aerosol phase under high OH exposure in OFR should also be considered.

**Author Response:** We are unable to follow the referee's argument. Page 21 line 30-32 discusses discrepancies among SC experiments, not OFR. Page 21 line 21 onwards discusses discrepancies between OFR and SC experiments, which, noting that they were conducted at somewhat different OA loadings, let us conclude that OFR-yields tend to be higher than SC yields, especially at higher OH exposures. Fragmentation (over functionalization) would tend to a) yield lower OFR SOA yields and b) as discussed in our previous publication (Bruns et al., 2015) and by Lambe et al., 2012, would yield higher O:C. We do not find any of these effects in our data set, hence, while we may be looking at compensating effects, we don't see results that seem to be driven into any direction by fragmentation only.

**Text modifications:** No modifications are required.

**RC1-25:** Page 22 line 12: "This generally indicates that we are able to identify the most relevant SOA precursors in the vehicle exhaust." This statement is not true. The SOA yield in SC and OFR was calculated based on a larger group of ArHC than merely OXYL/TOL. The author should calculated the SOA formation based on OXYL/TOL consumption in the SC (and OFR) and yield from OXYL/TOL experiment. Then compare the calculated SOA to the SOA formed in the SC.

**Author Response:** We agree with the referee's observation. We have used two different ways to approach the mass closure in our manuscript: 1) with a forward closure as presented in the initial Figure 5 as an example for 1 experiment (i.e. the reacted aromatic SOA precursor mass was weighted by a fixed SOA yield from the literature, which is the procedure suggested by the Referee 1 on RC1-25), and 2) via an indirect approach normalizing the formed SOA to the reacted ArHC mass (i.e. deriving an "effective yield" combining all reacted species), which allows to present the result as a function of OA loading. Because it appears confusing to have two different methods to address this, we decided to remove the SOA-closure from initial Figure 5 (bottom panels), and instead present data as "effective yield closure" (Figure 6) only.

**Text modifications:** We have revised all text sections to allow this modification; instead of a mass closure we discuss the results now in terms of a yield-closure in the original section 3.6. We have removed the statements on the mass closure from the text.

[Figure]

**Figure R1-3**: New Figure 5 with modified lower panels; the new figure caption reads as follows:
*"**Typical OFR-from-SC and SC photochemistry experiment.** Decay of dominant SOA precursors (benzene (BENZ), toluene (TOL), o-/m-/p-xylene (XYL) or ethylbenzene (EBENZ), C3-benzenes (C3BENZ)) upon photochemistry and associated SOA formation in (**a**) OFR (sampling from SC batch at different UV intensities, displayed is expt D3) and (**b**) SC (displayed is expt B1). (**a-b**) UV status and $O_3$ are indicated along with the NO:$NO_y$ ratio and the OH tracer BuOH-D9. Reacted ArHC fractions are provided in the SI per experiment, see Figure S4. Local time is given in intervals of (**a**) 30 min and (**b**) 15 min."*

**RC1-26:** Page 24 line 13-14: To conclude this, it is better to plot a graph showing O:C comparison as a function of OH exposure between SC and OFR. The SC shows similar O:C ratios with OH exposure of $1.2*10^{11}$ molec cm$^{-3}$ s to the those in OFR under 4.5 molec cm$^{-3}$ s, which seems not agreeable. The different vapor losses between the SC and OFR might also a reason.

**Author Response:** Our OH exposures are provided as color code; hence the data requested by the referee are already presented in the main text Figure 7 and initial SI Figure S16ab. However, while all our 6 SC experiments yield a similar end-point OH exposure as the 70% UV intensity setting in the OFR (marked in organe), there are 3 SC experiments which appear to have been conducted under conditions yielding higher $NO_3$/OA ratios and we believe that these experiments have reached exceptionally high O/C ratios compared to the other 3 SC experiments, despite no difference in their OH exposure. To facilitate the understanding of this plot, we have prepared an additional plot of O/C ratio vs. OH exposure as suggested by the referee.

**Text modifications:** We have replaced our initial Figure 7 and adjusted the corresponding text.
* * *
[Figure]

**Figure R1-4**: New version of Figure 7, splitting the experiments into a) and b) by their NO₃/OA ratio, as previously addressed in Figure S16 and in addition panel c) showing the plot requested by the referee (O:C vs. OH exposure); the new figure caption reads as follows:

*"**Bulk OA composition of SC and OFR SOA. a-b)**Van-Krevelen plot (O:C vs. H:C) for SOA formed during SC expts (n=6, GDI1 standard and w/GPF, cW and Ph 1 (cW)) and OFR-from-SC data points (n=10, GDI1 standard and w/GPF, full cW, full cE, Ph 1 (cW)) at different OFR UV settings (100%, 70%, 50%). a) shows SC Expt (A2, A3, B3; Table S4) and b) SC Expt (A1, B1, B2; Table S4), experiments which are characterized by a NH₄NO₃ is outside our CO₂⁺-AMS interference calibration range (Pieber et al., 2016). The POA contribution was subtracted from the total OA bulk composition; SOA/POA ratios are >> 10. The Aiken parameterization (Aiken et al., 2007; Aiken et al., 2008) has been applied to HR fitted data. Lines indicate the Van-Krevelen (VK) space typical for ambient AMS measurements (Ng et al., 2011). Error bars represent one standard deviation of measurement variability. (**c**) O:C of a) and b) as a function of [OH] exposure. [OH]ₑₓₚ in days refers to an assumed average ambient [OH] of 10⁶ molec cm⁻³."*

**RC1-27:** Fig. 2(b) POA point is missing for GDI4-catGPF (CW)
**Author Response:** The data point is now visible Figure 2.
**Text modifications:** The revised version of Figure 2 is provided in the answer to RC-2.

**RC1-28:** Fig. 5 Better to show the OH exposure range as well.
**Author Response:** Figure 5 presents one example experiment, and the OH-exposure information is provided in the figure caption to our earlier Figure 6 and Figure 7, as already described in the initial manuscript.
**Text modifications:** As described in our response above, no further modifications were made.

**References by RC1**

See acp-2017-942-RC1 for references mentioned by Referee 1.

**References to Author's Response:**

Aiken, A. C., DeCarlo, P. F., and Jimenez, J. L.: Elemental analysis of organic species with electron ionization high-resolution mass spectrometry, Anal. Chem., 79, 8350-8358, 10.1021/ac071150w, 2007.

Aiken, A. C., Decarlo, P. F., Kroll, J. H., Worsnop, D. R., Huffman, J. A., Docherty, K. S., Ulbrich, I. M., Mohr, C., Kimmel, J. R., Sueper, D., Sun, Y., Zhang, Q., Trimborn, A., Northway, M., Ziemann, P. J., Canagaratna, M. R., Onasch, T. B., Alfarra, M. R., Prevot, A. S. H., Dommen, J., Duplissy, J., Metzger, A., Baltensperger, U., and Jimenez, J. L.: O/C and OM/OC ratios of primary, secondary, and ambient organic aerosols with high-resolution time-of-flight aerosol mass spectrometry, Environ. Sci. Technol., 42, 4478-4485, 10.1021/es703009q, 2008.

Barmet, P., Dommen, J., DeCarlo, P. F., Tritscher, T., Praplan, A. P., Platt, S. M., Prévôt, A. S. H., Donahue, N. M., and Baltensperger, U.: OH clock determination by proton transfer reaction mass spectrometry at an environmental chamber, Atmos. Meas. Tech., 5, 647-656, 10.5194/amt-5-647-2012, 2012.

Bruns, E. A., El Haddad, I., Keller, A., Klein, F., Kumar, N. K., Pieber, S. M., Corbin, J. C., Slowik, J. G., Brune, W. H., Baltensperger, U., and Prévôt, A. S. H.: Inter-comparison of laboratory smog chamber and flow reactor systems on organic aerosol yield and composition, Atmos. Meas. Tech., 8, 2315-2332, 10.5194/amt-8-2315-2015, 2015.

Hildebrandt, L., Donahue, N. M., and Pandis, S. N.: High formation of secondary organic aerosol from the photo-oxidation of toluene, Atmos. Chem. Phys., 9, 2973-2986, 10.5194/acp-9-2973-2009, 2009.

Jimenez, J. L., Canagaratna, M. R., Drewnick, F., Allan, J. D., Alfarra, M. R., Middlebrook, A. M., Slowik, J. G., Zhang, Q., Coe, H., Jayne, J. T., and Worsnop, D. R. (2016). Comment on "The effects of molecular weight and thermal decomposition on the sensitivity of a thermal desorption aerosol mass spectrometer", Aerosol Sci Tech. 50, i-xv, 10.1080/02786826.2016.1205728

Jordan, A., Jaksch, S., Jürschik, S., Edtbauer, A., Agarwal, B., Hanel, G., Hartungen, E., Seehauser, H., Märk, L., Sulzer, P., and Märk, T. D.: $H_3O^+$, $NO^+$ and $O_2^+$ as precursor ions in PTR-MS: isomeric VOC compounds and reactions with different chemical groups, 5th International Conference on Proton Transfer Reaction Mass Spectrometry and its Applications, 2011.

Krechmer, J. E., Pagonis, D., Ziemann, P. J., and Jimenez, J. L. (2016). Quantification of Gas-Wall Partitioning in Teflon Environmental Chambers Using Rapid Bursts of Low-Volatility Oxidized Species Generated in Situ, Environ Sci Technol. 50, 5757-5765, 10.1021/acs.est.6b00606

La, Y. S.; Camredon, M.; Ziemann, P. J.; Valorso, R.; Matsunaga, A.; Lannuque, V.; Lee-Taylor, J.; Hodzic, A.; Madronich, S.; Aumont, B. Impact of chamber wall loss of gaseous organic compounds on secondary organic aerosol formation: explicit modeling of SOA formation from alkane and alkene oxidation. Atmos. Chem. Phys. 2016, 16, 1417−1431.

Lambe, A. T., Ahern, A. T., Williams, L. R., Slowik, J. G., Wong, J. P. S., Abbatt, J. P. D., Brune, W. H., Ng, N. L., Wright, J. P., Croasdale, D. R., Worsnop, D. R., Davidovits, P., and Onasch, T. B.: Characterization of aerosol photooxidation flow reactors: heterogeneous oxidation, secondary organic aerosol formation and cloud condensation nuclei activity measurements, Atmos. Meas. Tech., 4, 445-461, 10.5194/amt-4-445-2011, 2011.

Middlebrook, A. M., Bahreini, R., Jimenez, J. L., and Canagaratna, M. R. (2012). Evaluation of Composition-Dependent Collection Efficiencies for the Aerodyne Aerosol Mass Spectrometer using Field Data, Aerosol Sci Tech. 46, 258-271, 10.1080/02786826.2011.620041

Murphy, D. M. (2016). The effects of molecular weight and thermal decomposition on the sensitivity of a thermal desorption aerosol mass spectrometer, Aerosol Sci Tech 50, 118-125, 10.1080/02786826.2015.1136403

Ng, N. L., Canagaratna, M. R., Jimenez, J. L., Chhabra, P. S., Seinfeld, J. H., and Worsnop, D. R.: Changes in organic aerosol composition with aging inferred from aerosol mass spectra, Atmos. Chem. Phys., 11, 6465-6474, 10.5194/acp-11-6465-2011, 2011.

Pagonis, D., Krechmer, J. E., de Gouw, J., Jimenez, J. L., and Ziemann, P. J. (2017). Effects of gas–wall partitioning in Teflon tubing and instrumentation on time-resolved measurements of gas-phase organic compounds, Atmos. Meas. Tech. 10, 4687-4696, 10.5194/amt-10-4687-2017

Palm, B. B., Campuzano-Jost, P., Ortega, A. M., Day, D. A., Kaser, L., Jud, W., Karl, T., Hansel, A., Hunter, J. F., Cross, E. S., Kroll, J. H., Peng, Z., Brune, W. H., and Jimenez, J. L. (2016). In situ secondary organic aerosol formation from ambient pine forest air using an oxidation flow reactor, Atmos. Chem. Phys. 16, 2943-2970, 10.5194/acp-16-2943-2016

Peng, Z., Day, D. A., Ortega, A. M., Palm, B. B., Hu, W., Stark, H., Li, R., Tsigaridis, K., Brune, W. H., and Jimenez, J. L. (2016). Non-OH chemistry in oxidation flow reactors for the study of atmospheric chemistry systematically examined by modeling, Atmos. Chem. Phys. 16, 4283-4305, 10.5194/acp-16-4283-2016

Peng, Z., Day, D. A., Stark, H., Li, R., Lee-Taylor, J., Palm, B. B., Brune, W. H., and Jimenez, J. L. (2015). HOx radical chemistry in oxidation flow reactors with low-pressure mercury lamps systematically examined by modeling, Atmos. Meas. Tech. 8, 4863-4890, 10.5194/amt-8-4863-2015

Pieber, S. M., El Haddad, I., Slowik, J. G., Canagaratna, M. R., Jayne, J. T., Platt, S. M., Bozzetti, C., Daellenbach, K. R., Frohlich, R., Vlachou, A., Klein, F., Dommen, J., Miljevic, B., Jimenez, J. L., Worsnop, D. R., Baltensperger, U., and Prévôt, A. S. H.: Inorganic Salt Interference on $CO_2^+$ in Aerodyne AMS and ACSM Organic Aerosol Composition Studies, Environ Sci Technol, 50, 10494-10503, 10.1021/acs.est.6b01035, 2016.

Platt, S. M., El Haddad, I., Pieber, S. M., Zardini, A. A., Suarez-Bertoa, R., Clairotte, M., Daellenbach, K. R., Huang, R. J., Slowik, J. G., Hellebust, S., Temime-Roussel, B., Marchand, N., de Gouw, J., Jimenez, J. L., Hayes, P. L., Robinson, A. L., Baltensperger, U., Astorga, C., and Prévôt, A. S. H.: Gasoline cars produce more carbonaceous particulate matter than modern filter-equipped diesel cars, Sci Rep, 7, 4926, 10.1038/s41598-017-03714-9, 2017.

Turpin, B. J., and Lim, H.-J.: Species Contributions to PM2.5 Mass Concentrations: Revisiting Common Assumptions for Estimating Organic Mass, Aerosol Sci. Technol., 35, 602-610, 10.1080/02786820119445, 2001.

Weitkamp, E. A., Sage, A. M., Pierce, J. R., Donahue, N. M., and Robinson, A. L.: Organic aerosol formation from photochemical oxidation of diesel exhaust in a smog chamber, Environ. Sci. Technol., 41, 6969-6975, 10.1021/es070193r, 2007.

Zhang, X., Cappa, C. D., Jathar, S. H., McVay, R. C., Ensberg, J. J., Kleeman, M. J., and Seinfeld, J. H. (2014). Influence of vapor wall loss in laboratory chambers on yields of secondary organic aerosol, Proceedings of the National Academy of Sciences, 10.1073/pnas.1404727111

Zhao, Y., Lambe, A. T., Saleh, R., Saliba, G., and Robinson, A. L.: Secondary Organic Aerosol Production from Gasoline Vehicle Exhaust: Effects of Engine Technology, Cold Start, and Emission Certification Standard, Environ Sci Technol, 52, 1253-1261, 10.1021/acs.est.7b05045, 2018.

---

## Author Comment (AC2) · 15 Apr 2018

**Referee Comments #2 (acp-2017-942-RC2-supplement) and author response.**
**Simone M. Pieber et al.**

We thank the editor and referees for their comments. To guide the review process we have copied the referee comments in black text. Our responses are in regular blue font. We have responded to all the referee comments and made alterations to our paper (*in italic text*) and removed redundancies for clarification. Along with the revision we suggest a slightly changed title: "*Gas phase composition and secondary organic aerosol formation from gasoline direct injection vehicles with prototype particle filters investigated in a batch and flow reactor*"

**General:**

**RC2-1:** This paper describes measurements of primary emissions and secondary organic aerosol from four modern gasoline direct injection (GDI) engine equipped vehicles. The market share of GDI vehicles is rapidly increasing in the US and other countries, displacing the more traditional port fuel injection engine equipped vehicles. This paper represents probably most systematic study of SOA formation from GDI vehicles to date. The paper also investigates the impact of adding a retrofitted gasoline particulate filter (GPF) to two of the vehicles. While others have investigated the effect of this technology on primary emissions, I am not aware of previous studies that have investigated its impact on SOA formation. The paper also uses an oxidation flow reactor (OFR) and a smog chamber to investigate SOA formation, comparing the results between the two systems and with measurements made with individual compounds. The paper shows SOA formation dominates primary PM emissions (consistent with previous studies, though the SOA/POA ratio seem much larger than previous studies). The paper demonstrates that the majority of SOA formation is formed from cold-start emissions, which is not surprising but I have not seen it demonstrated before. The paper shows that somewhat more than half of the SOA formation appears due to 8 single-ring aromatics. Finally the paper shows GPF reduces primary PM emissions but does not reduce the SOA formation (or non-methane organic compound emissions). Overall the paper is well written and very comprehensive. The experiments appear to have been carefully conducted (though I agree with the other reviewer's concerns on treatment of wall loss), with results from repeated experiments shown (it would be nice to describe the precision a bit more). I think this paper makes a nice contribution and recommend that it be published in ACP after addressing the following comments.

**Author Response:** We thank the referee for the positive feedback and address the specific comments in the following. We provide answers to RC2 and modifications to our manuscript to the best of our abilities and have condensed and clarified the text where possible.

**Specific comments**

**RC2-2:** Abstract and a few other places "a large fraction (>0.5)" These statements refer to the mass closure of the SOA production based on measured precursors. 1 is greater than 0.5. The authors need to be more quantitative; e.g. give a range (or some other metric such as median and interquartile range).

**Author Response:** We agree with the referee's comment and have reformulated the statement in the abstract. As the assessment of the contribution is complex and condition-dependent, we prefer to remove the quantitative information in the abstract and rather keep the discussion of this issue in the results and discussion section 3.6 where we can address the details of this closure.

**Text modifications:** Abstract: "*A significant fraction of the SOA production was explained by those compounds, based on investigation of reacted NMOC mass and comparison of effective SOA yield curves with those of toluene, o-xylene and 1,2,4-trimethylbenzene determined in our OFR within this study and others from literature. Remaining discrepancies may result from diverse reasons including apart from unaccounted precursors also aging conditions, uncertainties of SOA yields for the aromatic hydrocarbons with different degrees of substitution, as well as experimental uncertainties in the assessment of particle and vapor wall losses.* "

**RC2-3:** Figure 6 suggests that this ratio likely varies with OA concentrations. Figure 5 suggests poor closure for 100% and 70% OFR conditions, but good closure for 50% OFR. SC has better closure at short timescales. The authors need to be more quantitative about the mass closure.

**Author Response:** We agree with the referee's observation. We have used two different ways to approach the closure in our manuscript: 1) with a forward mass closure as presented in initial Figure 5 as an example for 1 experiment (i.e. the reacted aromatic SOA precursor mass was weighted by a fixed SOA yield from the literature), and 2) via an indirect approach normalizing the formed SOA to the reacted ArHC mass (i.e. deriving an "effective yield" combining all reacted species), which allows to present the result as a function of OA loading, takes partitioning into account and is interpreted by comparing to the yields of classes of compounds (such as ArHC). As also Referee 1 noted that this is confusing, we have decided to remove the SOA-mass closure from initial Figure 5 (bottom panels), and instead present all our data only as a "yield closure" (method 2, Figure 6).

**Text modifications:** We have revised all text sections to allow this modification; instead of a mass closure we discuss the results now in terms of the yield-closure presented in section 3.6.

**RC2-4:** Retrofitted GPF – How was this done? How representative is it of how a true OEM designed and installed GPF-system would operate? It is hard to simply add a control system to a vehicle for which it was not designed (I have seen tests with a retrofit GPF hanging off the back of a car!), therefore I am always concerned about how representative the performance of researcher retrofitted system versus what might be done by a vehicle manufacturer. This is not to say that they are not seeing some effect of the GPF but it may be (much) less than the performance of bottom up engineered system. For example, I was surprised that the catalyzed GPF did not further reduce the NMOC emissions – what was the operating temperature of the GPF? Anyways I think the GPF results are interesting, but a bit more detail on how the retrofit was done, specifically limitations of researcher retrofitted systems should be acknowledged. Unless they can document that the retrofitted is representative of OEM designed and installed systems the conclusion section (_line 10 on page 25) is too strong.

**Author Response:** We agree that validity of the experiments to transfer our results to bottom up engineered systems is a crucial point. We have highlighted that our GPF system was a "retrofit" in our discussion paper, and have described the system in section 2.1.1 (page 6, line 5 onwards). It is indeed difficult to judge how such systems will be implemented by manufacturers and there will be also variability between different vehicles. For our experiments, which we designed with experts from the industry, we replaced the "muffler" which was located ca. 60 cm downstream of the three-way-catalyst (TWC) with the GPF. Pictures are provided in Figure R2-1 which has been added to the SI of our revised manuscript. The GPF was not externally heated, and its temperature dependent on the exhaust conditions, in the same way the temperature of the TWC is dependent on this.

We believe our retrofit is installed in a location that is representative for real-world retrofitting of GDI vehicles which are on the market currently, and that our experiments represent also the typical temperature conditions that can be found in such retrofits. An evaluation of this system on primary particle number emissions was published previously by a sub-set of our collaboration team (Czerwinski et al., 2017). The PCFEs of this investigation yielded ≥98% for GPF1 on either GDI1 or GDI4, and ≥86% for catGPF on GDI4, and indicate somewhat lower performance of the catGPF on primary PN reductions. This does not seem to be dependent on the location of installation in a vehicle, given that GPF1 on GDI4 showed good performance.

We don't believe that the generally somewhat lower performance of catGPF on primary PN is caused the limited effect on NMOCs and associated SOA. Instead, we believe that given that the vehicle is already equipped with a TWC, it is unlikely to see additional effects on NMOC reductions by a 2$^{nd}$ catalytic system during cold-starts. This is, because under cold-start conditions the NMOCs that pass the TWC will also pass the catGPF, which, at this point will likewise not have reached light-off temperatures to efficiently remove NMOCs.

We agree, that differently engineered systems (especially catGPFs that will be meant to replace the TWC), should be investigated in the future to see whether further NMOC reduction can be achieved.

[Figure]

**Figure R2-1**: *"**Pictures** of **a**) original "muffler" and GPF in comparison, **b**) retrofitted GPF, installed underfloor in replacement to "muffler"."*

**Text modifications:** We have modified above mentioned text sections to specify the distance between TWC and GPF (replacing muffler), which was roughly 60 cm. Along with modifications within the manuscript:

- *"GDI1 was studied i) in standard configuration, and ii) equipped with a prototype gasoline particle filter (GPF (cordierite, porosity 50%, pore size 19 μm, 2000 cells per square inch)), installed at the muffler ("underfloor"), which was located 60 cm downstream of the TWC. It's filtration quality at this configuration is equivalent to the best available technology for DPFs (personal communication by the manufacturer; particle number reductions by the application of the GPF are further assessed in Czerwinski et al., 2017, and yield PCFE ≥98%)."*
- *"GDI4 was retrofitted with i) the previously tested GPF (as above: cordierite, porosity 50%, and pore size 19 μm, 2000 cells per square inch, PCFE ≥98%), as well as ii) a Pd/Rh catalytically coated GPF (catGPF) (installed at the muffler, underfloor, while keeping the original TWC in the original position; the PCFE was ≥86%)."*

For the catGPF, additionally we had already the following statement provided in the initial version of the experimental section, which we kept in its original format:

- *"For the retrofitted catGPF, the primary purpose of the catalytically active coating is the constant self-cleaning of deposited carbonaceous material on the particle filter (personal communication with manufacturer). In future applications, such catalytic coating on a GPF might replace the existing TWC in GDI vehicles, or specifically, the TWC can be replaced with a GPF carrying the TWC coating."*

Along with this, however, we have modified the conclusions section to state more explicit, that GPFs carrying the TWC coating meant to replace the initial TWC will possibly lead to better NMOC removals than the current TWCs, and that research addressing this should be conducted:

- *"GPF application efficiently removes eBC, which is the dominant component of primary PM, and also shows small effects on the minor POA fraction. The volatile POA fraction passes through the filter in the vapor phase and later condenses when the exhaust is emitted and cooled; hence POA emission factors are not as significantly reduced as refractory PM. NMOC emissions and SOA formation are unaffected by the tested GPFs. This is particularly true when the GPF is catalytically inactive, and at cold-started driving cycles for catalytically active GPFs (i.e. when emissions pass through the TWC and the catGPF before light-off temperatures are reached.. This means that retrofitting GDI vehicles with GPFs will likely result in an important reduction of the total primary PM emitted (removal of refractory material), but will (under conditions similar to our experiments only to a small extent reduce NMHC (or NMOC) emissions including ArHC, and thereby not directly lead to a reduction of SOA. Future work on so-called "4-way catalysts", i.e. a TWC catalyst directly applied onto a GPF and installed at the location of the current TWC for simultaneous filtration of particulates and catalytic conversion of NMHC (or NMOC) should be conducted, to understand whether reductions of SOA precursors, SOA production, and semi-volatile primary PM can be achieved with further optimized systems."*

**RC2-5:** The dramatically higher NMOC emissions and SOA production from cold start is important. Can the authors quantify how much more important it is than hot start (e.g. using an analysis similar to that Saliba et al. EST 2017 10.1021/acs.est.6b06509 to compare hot and cold start emissions).

**Author Response:** We have provided a comparison of cold-started and hot-started cycles in Figure 2 (discussion paper), but had not given a comparison of the values previously, this is a highly interesting point. We now provide ratios of Ph1-SOA emission factors over Ph2-4-SOA emission factors in the manuscript, which is similar to Saliba et al., 2017-approach, and indicates that the cold-start is 20-50 times more important. However, it needs to be noted that this information should only carefully be transferred to the ambient air, additional parameters (e.g. how long/far will a vehicle be driven for during a real journey and what's the ambient temperature, i.e. "how cold" is the vehicle) need to be considered (see also Platt et al., 2017). In addition to our own ratio, we also provide a reference to a recent publication by Zhao et al., 2018, who also compare SOA from cold- and hot engine conditions.

**Text modifications:** *"Data are presented as a function of suspended OA for all experimental conditions of cold-started GDI1-3 (i.e. for full cW, cE; and Ph 1 (cW)), while GDI4 or hot engine conditions, i.e. Ph 2-4 (cW) are not included in our the analysis, as this data set includes only two experiments with concentrations levels close to our background measurements; a discussion of SOA yields from cold- and hot-engine emissions has recently been published by Zhao et al., 2018 for an OFR data set)."*

We have also added the following and updated the conclusions.

- *"Hot-engine emissions (Ph 2-4 sampling from cold-started WLTC, as presented in Figure 2d) also resulted in SOA formation, which was, however, 20-50 times lower in terms of EFs than SOA formed from Ph 1 sampling of a cold-started WLTC. This is in line with the trends indicated by the phase-dependent NMHC emissions (Figure 2c).*
- *"Future work should investigate the quantitative use of online OFR data in further detail for additional quantification of cold- and hot-start contribution of SOA to the total SOA burden; a discussion of the associated technical issues (i.e. changes in OH-exposure and condensational sink as well as the equilibration time inside the OFR reactor) has been recently published by Zhao et al., 2018."*

**RC2-6:** The conclusion section largely repeats conclusion from earlier in the paper. The paper would be improved if they put the results in context with the growing literature in this area. In particular I was interested if the results are consistent with the existing body of knowledge on SOA formation for PFI vehicle exhaust. My sense is that it is. You tested the vehicles using two different cycles? Were there any cycle dependencies or was cold start just dominant?

**Author Response:** We have modified and shortened the manuscript and provided additional comparison with literature. Figure 2a/b includes indeed data from WLTC and EDC cycle, comparing the cold/hot cycles. We don't observe any significant cycle-dependent differences which are larger than vehicle-by-vehicle or test-by-test variability, especially during cold-started cycles. As the EDC cycle was tested only as the full cycle and not split into separate phases, no explicit analysis of the SOA contribution to the total cycle can be made in comparison to the WLTC. This information has already been provided in the discussion paper and is now also stated more clearly in the conclusions section. Further points are discussed along the referee comments below specifically to Figure 2.

**Text modifications:**

- *"While no drastic cycle-dependencies (WLTC vs. EDC) were observable from our tests (especially during cold-started cycles), EFs of primary NMHC and THC were reduced by a factor of 90 under hot-started conditions."*
- *"Emissions of all cold-started vehicles, technologies and driving tests showed significant SOA formation upon photochemical oxidation (Figure 2b), in line with other studies on GDI as well as PFI systems (Platt et al., 2017; Gordon et al., 2014; Saliba et al., 2017; Zhao et al., 2018).*
- *"Overall, the SOA potential (in terms of an emission factor) of the tested vehicles agreed with recent literature reports from both, GDI and port fuel injection systems (PFI)."*

**RC2-7:** Figure 2 – The y-axes are five orders of magnitude. This illustrates large changes, but changes of a factor of 2 or 3 can also be interesting. For most tests it does appear that the GPF is reducing the POA emissions, but not as dramatically as the EC. I did not get that impression reading the text but it does appear in the figure. More attention needs to be paid to these trends.

**Author Response:** We agree that Figure 2 is packed with details, and we have revised it (see below). Further, we refer to our related publication (Munoz et al., 2018), which discusses the difference between cold- and hot-started cycle emissions in detail for GDIs in standard configuration, regarding CO, NOx, particle number and genotoxic PAHs. We agree on the observation of POA removal with the GPF and have adjusted our statements:

**Text modifications:** *"Retrofitted GPFs (including catGPF behind the standard TWC) appeared also to reduce the POA fraction.",* is added in the results and have also updated the abstract ("*GPF retrofitting was found to greatly decrease primary particulate matter (PM) through removal of eBC, showed partial removal of the minor POA fraction, ...*") and conclusions ("*GPF application efficiently removes eBC, which is the dominant component of primary PM, and also shows small effects on the minor POA fraction.*").
We also added the following: *"A detailed discussion on emissions of CO, NOx, particle number and genotoxic PAHs from cold- vs. hot-started cycle driven GDI vehicles in standard configuration can be found in our related publication by Munoz et al., 2018.*

**RC2-8:** Figure 2 – The SOA production seems surprisingly high. For GDI1 the total NMHC (most of which are not SOA precursors) is around 1000 mg/kg (Figure 2a). The SOA production is between 200 and 600 mg/kg (Figure 2a) – the SOA production from the GPF equipped experiments with GDI1 seem incredibly high. This suggests an effective of SOA of 20-60% of the total NMHC emissions of which less than half is aromatics (Figure 4b is misleading because the NMOC measured by PTR is only 65% of NMHC measured by FID). The SOA production seems higher than previous studies of modern vehicles (they are more similar to 25 year old vehicles). I guess Figure 6 suggest SOA yields are "reasonable", but I was confused looking at Figure 2 (maybe it is just the log scale with 5 orders of magnitude). Are there background issues?

**Author Response:** We agree with the very high SOA production of especially GDI1-3, but specifically GDI1. For the corresponding SOA emission factors in relation to previous publications, there are two additional things to consider. 1) the OA mass at which these SOA-emission factors are determined, and 2) that most previous literature is using exclusively data from SC and not OFRs (data points at the upper end of the SOA emission factors in our experiments are derived from OFR experiments). Additionally, previous SC experiments which the referee refers to were conducted typically at lower OH exposures than our experiments, and at different ratio of NO/NO$_2$ or total NO$_x$/VOC, points which are discussed later in the manuscript in the section "SOA yield analysis" and in Zhao et al. 2017, which we have added to our reference list
Regarding background issues, we have provided additional information in our answer to RC1 as well as within other answers herein. Experiments were conducted with high purity air after extensive cleaning (described in the main text). Background was insignificant compared to our vehicle testing data, except for Ph2-4 experiments which were close to background levels in terms of the NMOCs (noted in Figure caption to Figure 4).
**Text modifications:** Figure 2 was modified for clarity (see below).

**RC2-9:** Figure 2c – There is a lot of vehicle to vehicle variability (2+ orders of magnitude). Are the reductions between cold and hot start consistent across vehicles? Plotting ratios may be more informative. What is up with the experiments with an NMHC emission rate of 0.1 mg/kg? Are those valid data?

**Author Response:** Vehicle by vehicle differences between cold- and hot-started cycles can be seen from Figure 2a and we have already provided ratios in our discussion in section "3.1 Pollutants as function of vehicle technology and driving cycle"; We provide median and interquartile ranges for data presented in Figure 2c now; 0.1 mg/kg is the detection limit of our NMHC measurements, and data were below this limit in some cases.
**Text modifications:** Figure 2 was modified for clarity (see below).

**RC2-10:** Figure 2 is very busy (especially panel d). It is basically impossible to sort out the trends. Pick the key points you want to make and plot just that data. The SOA production appears surprisingly close to the NMHC emissions (it even exceeds it for some vehicles).

**Author Response:** Indeed, (OFR-from-SC)-SOA is very close to the FID-based NMHC emission factors determined in the SC. SOA emission factors never exceed the emission factors of aromatic hydrocarbons (which is the relevant information for SOA in our case). Hence, the information provided is fully consistent. We have revised the figure for clarity.

**Text modifications:** The figure was revised, see below, and have clarified section 3.1.

[Figure]

**Figure RC2-1:** new version of Figure 2; new figure caption reads as follows:
*"Emission factors (EF) of pollutants from cold-started ("c") and hot-started ("h") test cycles (WLTC ("W") and EDC ("E")). Individual cW and hW phases are indicated as "Ph" 1-4. (a) Total and non-methane hydrocarbons (THC, NMHC) and primary gravimetric particulate matter (PM) from CVS measurements over entire test cycles for different vehicle configuration and test conditions (average±1SD), (b) primary PM (equivalent black carbon (eBC) and primary organic aerosol (POA)), and secondary organic aerosol (SOA) from SC and OFR-from-SC experiments, and from online OFR operationat 100% UV per vehicle configuration for cold-started test cycles (average±1SD), (c) THC/NMHC of cW and hW experiments from (a) separated into individual cycle phases (median, and P25-P75 range are shown). (d) POA, eBC, aromatic hydrocarbons (ArHC) and SOA over the full cW and cE, compared to individual phases of cW from SC batch experiments and OFR-from-SC (average±1SD). (a-d) EF calculation is detailed in the SI. The time-resolved SOA*

*profile from online OFR measurements conducted on GDI4 in 2015 (standard and catGPF) is provided in Figure S14."*

**RC2-11:** Additional ArHC (page 23). The analysis in Zhao et al. (EST 50(8): 4554-4563 2016) suggests some of the IVOCs are alkylated single ring aromatics larger than those included in the analysis here. How does including IVOC component measured by Zhao et al. change the analysis? His analysis suggests that IVOCs contribute somewhat less than half of the SOA in gasoline vehicle exhaust.

**Author Response:** We agree with the referee that those compounds could give additional SOA mass in our experiments. However, due to the different analytical techniques applied (we use a PTR-ToF-MS vs. the TD-GC-MS technique by Zhao et al., 2016), the data cannot be simply combined, as at the current moment, we are uncertain of which fraction of the IVOC (which makes up ca. 50% of the SOA in Zhao et al., 2016) overlaps with a fraction accounted for in our experiments, as we might be able to see fragments of e.g. alkyl-substituted aromatics that would fall into the IVOC category in Zhao et al., 2016 in our understanding. Ignoring a potential double count here, would likely allow us to conclude that those compounds make up a big fraction of the 50% of the missing mass seen by the FID but not PTR-ToF-MS, and would also bring our yield analysis in closer agreement with the yields determined for the vehicle exhaust (taking these additional compounds into account) would agree with that of single aromatic compounds.

**Text modifications:** We have added small adjustments to make this point more explicit in our manuscript.

**RC2-12:** Page 3 "diesel PM emissions have been greatly reduced." – This is true for new diesel particulate filter (DPF) equipped vehicles but there are a lot of old diesels on the road, especially in Europe so human exposure to diesel particles has probably not yet been greatly reduced. Eventually it will be when the fleet is completely turned over. May want to refine this statement.

**Author Response:** We agree with the comment and have refined our statement to specify we are referring to test bench measurements and emission factors of the recent vehicle fleet, which isn't fully compliant with the fleet on the road.

**Text modifications:** *"Due to the regulatory attention and the improved after-treatment systems, diesel PM emissions from new generation vehicles have been greatly reduced, and fleet modernization will help to reduce their burden in the ambient air."*

**RC2-13:** Page 4 "modern diesel vehicles" Modern is too generic. You should be more precise catalyzed-DPF equipped diesel vehicles. I don't necessarily think modern = DPF.

**Author Response:** We agree with the comment and have specified our terms.

**Text modifications:** *"modern"* was replaced with *"catalyzed-DPF equipped"*

**RC2-14:** Page 8 "experiment. Control experiments were conducted regularly in the SC to estimate the contribution of the SC background to SOA formation." Please provide another sentence or two here that describes results from control experiments. How much SOA was formed in controls and how does it compare to what is measured in an experiment with vehicle exhaust. Did you run control experiments with the OFR – what were the background levels in that system?

**Author Response:** Thanks for the comment. In brief: control experiments were SOA experiments conducted with the SC and OFR after the standard cleaning procedures, and in both cases, SOA formed during control experiments was insignificant compared to SOA formed during vehicle testing, except for Ph2-4 which are close to background levels in the SC as stated in cation to Figure 4, and and eventually GDI4 experiments which formed less SOA. SOA-control experiments with ammonium sulfate as seed in the SC yielded a SOA background < 1 µg m$^{-3}$ after 2 hours of aging (i.e. comparable to the typical vehicle SC SOA experiments). Control experiments with UV on where also conducted in the OFR. When sampling test bench room air through our 1:8 dilution system (prior to online tests during cWLTC and hWLTC), background levels where <10 µg m$^{-3}$ (which is far below the online vehicle SOA measurements of 100-2000 µg m$^{-3}$). When sampling pure air from the cleaned SC before OFR-from-SC experiments, background levels where < 2 µg m$^{-3}$.

**Text modifications:** We have added the following:

- *"Background measurements of the clean chamber were conducted prior to each experiment, and was insignificant compared to our measurements except for Ph2-4 or GDI4 experiments as stated in the results. Photochemistry control experiments were conducted regularly to estimate the contribution of the SC background to SOA formation; these experiments were conducted after the standard cleaning procedure. Instead of vehicle exhaust, pure air was used as a sample and ammonium sulfate (50 ug m$^{-3}$) injected as seed. Other experimental procedures were in line with the typical vehicle experiments. We found a SOA background of < 1 μg m$^{-3}$, which is below the SOA concentrations formed during vehicle exhaust aging Concentration-levels of our SC experiments were representative for urban ambient conditions, as reported in Table S4-S7."*
- *"Background levels were <2 μg m$^{-3}$ SOA before OFR-from-SC experiments (when sampling from cleaned SC) and <10 μg m$^{-3}$ when sampling diluted (1:8) test bench room air prior online-experiments."*

**RC2-15:** Section 2.2.6 – I am pretty sure that you are calculated yields using the reacted aromatic mass in the denominator however this statement is confusing "the ratio of the SOA mass to the reacted SOA-forming mass, delta_NMOCreacted" My understanding is that delta_NMOCreacted is not the same as the reacted aromatic mass. This needs to be cleaned up to avoid confusion. May also want to state this in the caption for Figure 6 to reminder reader of how yields are calculated.

**Author Response:** We calculated yields by normalizing the formed SOA mass to the reacted delta of the 8 selected aromatic hydrocarbons which dominated the identified NMOC fraction. We have revised this statement and clarified this also in the discussion to Figure 6.

**Text modifications:** According above description, the new text reads: *"SOA yields analysis is based on SC and OFR-from-SC experiments with GDI1-3. An effective SOA yield (Ye), was calculated as the ratio of the SOA mass to the reacted SOA-forming species i (in Δμg m$^{-3}$, Eq. (2)). We take into account all our identified SOA precursors (which refers to the 8 dominant aromatic hydrocarbons presented in Figure 4d), neglecting non-reactive and non-SOA forming precursors and assuming that all relevant SOA precursors are measured.*

$$Ye = \frac{\Delta SOA}{\sum_i \Delta SOA\_precursor_{i,reacted}} \qquad\qquad (2)\text{"}$$

**RC2-16:** Using "NMOC" to describe the sum of the PTR measurements is confusing as it is measuring less than 2/3rds of the organic gas emissions as measured with FID. This limitation needs to be stated more clearly (it is in the intro but the reader will likely forget – e.g. adding to caption of Figure 4 would be good and in other places in the main text when you discuss NMOC.

**Author Response:** We agree and have added the information throughout the manuscript and in the caption of Figure 4; further we have shortened section 3.4 and moved detailed discussion of $O_2^+$ charge processes and fragmentation of alkyl-substituted aromatics to the SI to make this section more concise on the point of SOA-precursor identification.

**Text modifications:** As described in our response. The revised figure caption reads as follows:
*"(b) Relative composition of the PTR-ToF-MS derived NMOC fraction (which makes up 65%±15 of the FID-based NMHC signal on a carbon-basis for cW/cE/Ph 1(cW)), (c) total ArHC EFs (which make up 49±8% of the FID-based NMHC signal on a carbon-basis for cW/cE/Ph 1(cW), and (d) relative contribution of the 8 dominant ArHC (correspond to 96.7±3.3% of the total ArHC signal for cW/cE/Ph 1(cW))."*.

**RC2-17:**
Emissions data from tests in mg/kg-fuel needs to be provided in tables in supplemental.

**Author Response:** Emissions data presented in Figure 2 (in mg/kg fuel) can be made available to others upon request. Median values are stated in section 3.1.

**Text modifications:** No modifications were made to the text.

**References to Author's Response:**

Czerwinski, J., Comte, P., Heeb, N., Mayer, A., and Hensel, V.: Nanoparticle Emissions of DI Gasoline Cars with/without GPF, SAE Technical Paper 2017-01-1004, 10.1021/acs.est.7b05045, 2017.

Gordon, T. D., Presto, A. A., May, A. A., Nguyen, N. T., Lipsky, E. M., Donahue, N. M., Gutierrez, A., Zhang, M., Maddox, C., Rieger, P., Chattopadhyay, S., Maldonado, H., Maricq, M. M., and Robinson, A. L.: Secondary organic aerosol formation exceeds primary particulate matter emissions for light-duty gasoline vehicles, Atmos. Chem. Phys., 14, 4661-4678, 10.5194/acp-14-4661-2014, 2014.

Jathar, S. H., Gordona, T. D., Hennigan, C. J., Pye, H. O. T., Pouliot, G., Adams, P. J., Donahue, N. M., and Robinson, A. L.: Unspeciated organic emissions from combustion sources and their influence on the secondary organic aerosol budget in the United States, Proc. Natl. Acad. Sci. U. S. A., 111, 10473-10478, 10.1073/pnas.1323740111, 2014.

Muñoz, M., Haag, R., Honegger, P., Zeyer, K., Mohn, J., Comte, P., Czerwinski, J., and Heeb, N. V.: Co-formation and co-release of genotoxic PAHs, alkyl-PAHs and soot nanoparticles from gasoline direct injection vehicles, Atmos. Environ., 178, 242-254, 10.1016/j.atmosenv.2018.01.050, 2018.

Nordin, E. Z., Eriksson, A. C., Roldin, P., Nilsson, P. T., Carlsson, J. E., Kajos, M. K., Hellén, H., Wittbom, C., Rissler, J., Löndahl, J., Swietlicki, E., Svenningsson, B., Bohgard, M., Kulmala, M., Hallquist, M., and Pagels, J. H.: Secondary organic aerosol formation from idling gasoline passenger vehicle emissions investigated in a smog chamber, Atmos. Chem. Phys., 13, 6101-6116, 10.5194/acp-13-6101-2013, 2013.

Platt, S. M., El Haddad, I., Pieber, S. M., Zardini, A. A., Suarez-Bertoa, R., Clairotte, M., Daellenbach, K. R., Huang, R. J., Slowik, J. G., Hellebust, S., Temime-Roussel, B., Marchand, N., de Gouw, J., Jimenez, J. L., Hayes, P. L., Robinson, A. L., Baltensperger, U., Astorga, C., and Prévôt, A. S. H.: Gasoline cars produce more carbonaceous particulate matter than modern filter-equipped diesel cars, Sci Rep, 7, 4926, 10.1038/s41598-017-03714-9, 2017.

Saliba, G., Saleh, R., Zhao, Y., Presto, A. A., Lambe, A. T., Frodin, B., Sardar, S., Maldonado, H., Maddox, C., May, A. A., Drozd, G. T., Goldstein, A. H., Russell, L. M., Hagen, F. P., and Robinson, A. L.: A comparison of gasoline direct injection (GDI) and port fuel injection (PFI) vehicle emissions: emission certification standards, cold start, secondary organic aerosol formation potential, and potential climate impacts, Environ Sci Technol, 10.1021/acs.est.6b06509, 2017.

Zhao, Y., Lambe, A. T., Saleh, R., Saliba, G., and Robinson, A. L.: Secondary Organic Aerosol Production from Gasoline Vehicle Exhaust: Effects of Engine Technology, Cold Start, and Emission Certification Standard, Environ Sci Technol, 52, 1253-1261, 10.1021/acs.est.7b05045, 2018.

Zhao, Y., Nguyen, N. T., Presto, A. A., Hennigan, C. J., May, A. A., and Robinson, A. L.: Intermediate Volatility Organic Compound Emissions from On-Road Gasoline Vehicles and Small Off-Road Gasoline Engines, Environ Sci Technol, 50, 4554-4563, 10.1021/acs.est.5b06247, 2016.

Zhao, Y., Saleh, R., Saliba, G., Presto, A. A., Gordon, T. D., Drozd, G. T., Goldstein, A. H., Donahue, N. M., and Robinson, A. L.: Reducing secondary organic aerosol formation from gasoline vehicle exhaust, Proc Natl Acad Sci U S A, 114, 6984-6989, 10.1073/pnas.1620911114, 2017.

---

## Author Response (AR1)

**Dear Dr. Hamilton,**

please find attached our response letters and marked-up files, along with the revised manuscript. Upon reflecting on the title, we believe it should finally read: "Gas phase composition and secondary organic aerosol formation from standard and particle filter-retrofitted gasoline direct injection vehicles investigated in a batch and flow reactor", to best cover the revised content of the paper.

We had suggested "Gas phase composition and secondary organic aerosol formation from gasoline direct injection vehicles with prototype particle filters investigated in a batch and flow reactor" in our response letters, which, however, would not be fully correct as the bigger part of our data set is in "standard configuration", not with GPFs.

Further, upon preparing the revised manuscript, we have found further ways to make our language more concise and clarify the manuscript to avoid confusions as suggested by the referees, as you will see in the marked-up version.

During the update of the manuscript, we noted that on Figure 5 which we suggested in the author response, we should replace the SC experiment given (which was initially B1), with a more typical and well understood experiment. We have therefore, replaced B1 with experiment A2. This is now in line with our updates in response to the referee questions on the yield-discussion: We added in the discussion that A1, B1 and B2 are unexpectedly high and potentially impacted by an unaccounted for interference, hence, we think it is more consistent to show one of the well-understood experiments (A2, A3 and B3) in Figure 5 as a typical example. Additionally, we have indicated which data points correspond to A1-3 and B1-3 in Figure 6 and 7 (by adding identifiers compared to the versions given in the response letter and initial manuscript), and have slightly modified colors in Figure 6a and b, to help guide the reader through our discussions in the text. We apologize for inconvenience caused by this late modification, but believe that while they do not alter our conclusions or arguments in the response letter are bringing big improvements and facilitations to the reader. We attach the further improved figures (including the identifiers) here once more.

We hope you find our work significantly improved and are looking forward to your feedback.

Thanks and best regards, Simone Pieber, and co-authors.

Figure 1. Typical OFR-from-SC and SC photochemistry experiment. Decay of dominant SOA precursors (benzene (BENZ), toluene (TOL), o-/m-/p-xylene (XYL) or ethylbenzene (EBENZ), C3-benzenes (C3BENZ)) upon photochemistry and associated SOA formation in (a) OFR (sampling from SC batch at different UV intensities, displayed is expt D3) and (b) SC (displayed is expt A2). (a-b) UV status, O3 and HONO injection are indicated along with the NO:NOy ratio and the OH tracer BuOH-D9. Reacted ArHC fractions are provided in Figure S5 per experiment. Local time is given in intervals of 15 min.

---

## Author Response (AR2)

Co-Editor Decision: Publish subject to minor revisions (review by editor) (01 Jun 2018) by Jacqui Hamilton.

Comments to the Author:

I am happy with the majority of the revisions and think the paper should be published subject to some final minor revisions. **AC:** Please find our response below in blue font, text from the manuscript is provided in *italics font*.

Abstract: The wording of "may have resulted from diverse reasons including, apart from unaccounted precursors also matrix effects." is unclear and should be rewritten. **AC:** The sentence reads now: *"Remaining discrepancies, which were lower in the SC and higher in the OFR, were up to a factor of 2 and may have resulted from diverse reasons including unaccounted precursors and matrix effects. GPF-retrofitting significantly reduced primary PM through removal of refractory eBC and partially removed the minor POA fraction."*

page4, line 5: I think you have an extra ) **AC:** There was an extra space which we removed.

page 9, line 16: change to "can cause significant" **AC:** This was adjusted.

page 14, line 7, change from "reduced" to "reduce" **AC:** We made this modification on page 14, line 27.

Page 15, line 26: space between with and UV **AC:** This was adjusted.

Page 19, line 11: What was dominated by XYL/EBENZ? Reactivity? **AC:** The sum of the reacted ArHC mass (delta reacted species) was dominated by XYL/EBENZ, hence we have rewritten the statement to read as follows: *"At the final OH exposure of $(1.4-5.8)x10^{11}$ molec $cm^{-3}$ $s^{-1}$ the reacted ArHC mass was dominated by XYL/EBENZ (41±3%),"*

Page 20, line 1: This information would be very useful for others looking at similar emissions. Can you put in a table in SI rather than just the figure? **AC:** The data from OFR experiments conducted within our study as presented in Figure 6a,b were added in Table S8.

Table S8. OFR yields from this study as presented in Figure 6 in the main text.

| Compound | OA
µg m$^{-3}$ | OA_err
µg m$^{-3}$ | Ye
µg ug$^{-1}$ | Ye_err
µg ug$^{-1}$ |
|---|---|---|---|---|
| TOL | | | | |
| TOL | 26 | 4 | 0.15 | 0.02 |
| TOL | 50 | 8 | 0.18 | 0.03 |
| TOL | 66 | 10 | 0.21 | 0.03 |
| TOL | 69 | 10 | 0.19 | 0.03 |
| TOL | 70 | 11 | 0.16 | 0.02 |
| TOL | 106 | 16 | 0.23 | 0.03 |
| TOL | 117 | 18 | 0.21 | 0.03 |
| TOL | 291 | 44 | 0.29 | 0.04 |
| TOL | 795 | 119 | 0.35 | 0.05 |
| OXYL/TOL (3:1) | | | | |
| OXYL/TOL (3:1) | 347 | 52 | 0.64 | 0.10 |
| OXYL/TOL (3:1) | 507 | 76 | 0.46 | 0.07 |
| OXYL/TOL (3:1) | 588 | 88 | 0.53 | 0.08 |
| OXYL/TOL (3:1) | 852 | 128 | 0.76 | 0.11 |
| OXYL/TOL (10:1) | | | | |
| OXYL/TOL (10:1) | 26 | 4 | 0.14 | 0.02 |
| OXYL/TOL (10:1) | 82 | 12 | 0.34 | 0.05 |
| OXYL/TOL (10:1) | 104 | 16 | 0.26 | 0.04 |
| OXYL/TOL (10:1) | 176 | 26 | 0.27 | 0.04 |
| OXYL/TOL (10:1) | 266 | 40 | 0.45 | 0.07 |
| TMB/TOL (2:1) | | | | |
| TMB/TOL (2:1) | 141 | 21 | 0.36 | 0.05 |
| TMB/TOL (2:1) | 192 | 29 | 0.29 | 0.04 |
| TMB/TOL (2:1) | 195 | 29 | 0.37 | 0.06 |
| TMB/TOL (20:1) | | | | |
| TMB/TOL (20:1) | 675 | 101 | 0.45 | 0.07 |

Page 20, line 15. This sentence is difficult to understand. "did not suggest inducing any difference" **AC:** We have removed this part of the phrase and kept only *"At average it agreed by a factor of 1.0±0.3."*

Page 20, line 19: Not sure what you mean by "tendency H:C"? **AC:** We removed "by tendency".

Page 21, line 3. I dont understand this sentence "we could match the yields....." Please reword. **AC:** We have rephrased this sentence, and have also adjusted the abstract and conclusions to agree with this as follows.

Main text:

- *GDI vehicle exhaust effective SOA yields (SC and OFR) appeared relatively higher than our reference measurements with specific SOA precursors, by up to a factor of 2, with larger discrepancies for the OFR and smaller discrepancies for the SC. This is detailed further below (Figure 6a, Section 3.6.2).*

*"Effective yields of vehicle exhausts were in the range of those from single precursors, particularly, when considering SC experiment, but with a higher discrepancy for the OFR experiments, Figure 6a). To explain the remaining discrepancy, which was up to a factor 2, we focus on the following two hypotheses:"*

Abstract: *"Remaining discrepancies, which were smaller in the SC and larger in the OFR, were up to a factor of 2 and may have resulted from diverse reasons including unaccounted precursors and matrix effects."*

Conclusions: *"While a significant fraction of the SOA could be attributed to the identified precursors, divergences in the effective SOA yields remained up to a factor of 2 when comparing to specific precursors."*

Page 22, section 3.7: Your explanation for the discrepnacy in the H:C doesn't make sense. Adding an OH to the ring instead of a H doesn't change the H:C ratio. Surely this relates to the amount of ring opened versus ring closed species. A very important but missing factor here is the amount of NO2 present. I assume NO2 is very high since you've titrated the NO with ozone? Please discuss the amount of NO2 in the two systems. At high concentrations of NO2, there will be more competition for the aromatic adduct between O2 and NO2. **AC:** In section 3.5 we stated that NO2 can not be unambiguously quantified with our experimental set-up. However, from Figure 5 you can see that NOy (an upper limit for $NO_2$) is not much exceeding 100 ppb. This means that the reaction of $NO_2$ with the aromatic OH-adduct would be less than 6% compared to $O_2$. If the OH-adduct adds $O_2$ forming a peroxy radical this can terminate via several path way: 1) reaction with NO to a nitrate (addition of 1 H atom compared to precursor); 2) reaction with HO2 to a hydroperoxide (addition of 2 H atoms); 3) reaction with a $RO_2$ forming an alcohol (addition of 2 H atoms) or a carbonyl (no H addition). These pathways are again possible after each OH addition on a C=C double bond of an oxidation product. As shown by Molteni et al. (2018), highly oxygenated low volatility products with 2 and 4 additional H-atoms can be formed this way. The main difference between SC and OFR is the significantly higher OH concentration in the OFR (while exposure remains similar, at least at the lower UV exposure) and the higher NO concentration in the SC. Both effects may influence the termination pathways of $RO_2$ as described above. More information on HOx/ROx cycling in the SC and the OFR would be needed to make firm statements.

We replaced the old text: *"This agreement did not apply for the H:C, however, for which the OFR yielded higher values than the SC. Initially higher NO-levels in the SC and overall higher OH concentration in the OFR (leading to more than one OH addition to the aromatic ring) as discussed in Section 3.6 could explain the observed trends. Further, we speculate that reaction termination with $HO_2$ rather than $RO_2$ would also increase the H:C in the OFR relative to the SC."*

It now reads: *"This agreement did not apply for the H:C, however, for which the OFR yielded higher values than the SC. Oxidation products with two more H-atoms than the precursor are formed when the aromatic-OH adduct adds an oxygen molecule and the peroxy radical then terminates by a reaction with $HO_2$ or $RO_2$. If the oxidation product contains a C=C double bond, this reaction sequence can be repeated leaving a second generation oxidation product with four additional H-atoms. The formation of highly oxygenated low volatility products with 2 and 4 additional H-atoms under high OH concentrations has been shown by Molteni et al. (2018). The higher NO-levels in the SC and the*

*higher peroxy radical concentration in the OFR are critical to which termination pathways of the peroxy radical occur. For example, an enhanced reaction termination with HO₂ rather than RO₂ would increase the H:C in the OFR relative to the SC."*

Figure 5: I cant really see the colour of the NO3 and NH4 circles as the lines are too think. Can this be modified. **AC:** This was adjusted, see Figure below, the figure caption remains as is.

[Figure]

Figure 5.

[revised manuscript text omitted]
 (following the requirements of the PMP- Particle Measurement Program of the ECE GRPE
20 Group).

**S3. Sampling materials and length**

- Tubing to sample direct emissions from the vehicle tailpipe for injection into the SC or online-OFR, or direct gas-phase measurements were made of SilcoTek®-coated steel (12 mm diameter), temperature controlled at 140°C and operated under high flows (30 L $min^{-1}$) to avoid substantial losses over the sampling length of roughly 8 m. Ejector dilutor 1 was
25 placed in a temperature controlled housing (200°C), and ejector dilutor 1 operated at 80°C.
- Instruments sampling either from the SC, behind the OFR, or directly from the dilution system were connected via specific tubing for gas-phase and particle phase. Particle-phase tubing was made of stainless steel (6 mm diameter), and up to 2 m length. Support pumps were used at the instrument inlets, to minimize sampling residence time by increasing the flow rate. Total tubing length to reach all of the gas-phase instrument inlets, which were likewise equipped with

support pumps was up to 2 m. Tubing was made of Teflon or SilcoTek®-coated steel. The sampling line of the PTR-ToF-MS instrument and FID was temperature controlled at 60°C.

- SilcoTek®-coating and Teflon are suitable for sampling of species known to be easily retained on surfaces, such as formaldehyde, acetic acid, acetaldehyde, for which otherwise, in addition to the uncertainties of PTR-ToF-MS analysis, also tubing losses could induce a shift in our gas-composition analysis.

- The sampling system between the SC and OFR (for OFR-from-SC experiments) was made of a combination of SilcoTec® coated steel and conductive Teflon tubing, suitable for simultaneous gas- and particle phase sampling. The total length between SC and OFR inlet was roughly 35 cm (6 mm diameter, ca. 8 L min$^{-1}$ flow). Additionally, all measurements from the dark SC batch sample were performed for at least 10 minutes, to reach a stable signal.

**S4. OFR data quality (OH exposure, non-OH losses and NOx influence)**

Several recent studies (Li et al., 2015;Peng et al., 2016;Peng et al., 2015) have estimated the contribution of alternative reaction processes than OH radical-induced ones in the OFR across a range of operating conditions (residence time, water vapor availability, and external OH reactivity (OHR$_{ext}$), which is the available OH-reactive material). These non-OH processes include reaction with photons (185 nm, 254 nm), and reactions with oxygen allotropes (excited oxygen atoms (O($^1$D)), ground state oxygen atoms (O($^3$P)), ozone (O$_3$)) were identified as relevant loss processes to precursor molecules. Under certain operating conditions, also suppression of OH formation is critical. We applied a previously published model (Li et al., 2015;Peng et al., 2016;Peng et al., 2015) to estimate competing reaction with OH and loss of precursor molecules by non-OH sources, and estimated the influence of NO$_x$ based on Peng and Jimenez, 2017. Details on model input parameters are presented in the following:

(a) **OFR-from-SC** (see results in Figure S11Figure S10). As input to the model we used OHR$_{ext}$=100 s$^{-1}$, [O$_3$]=1.97x10$^{14}$ molec cm$^{-3}$ (corresponding to 8 ppm at 100% UV intensity), a water mixing ratio=0.01 (1% absolute humidity, corresponding to 50% RH at 25°C) and a residence time=100 sec. O$_3$ measured at our reactor output for 70% UV intensity was 0.74x10$^{14}$ molec cm$^{-3}$ (3 ppm), and at 50% UV intensity 0.17x10$^{14}$ molec cm$^{-3}$ (0.7 ppm). OHR$_{ext}$ was calculated following Eq. (S2).

$$OHR_{ext} = \sum_i (c_{NMOC,i} * k_{OH,NMOC,i});$$
$i$=BENZ, TOL, XYL/EBENZ, C3-BENZ, CO, BuOH-D9  (S2)

where $k_{OH}$ of benzene (BENZ), toluene (TOL), xylene/ethylbenzene (XYL/EBENZ), C3-benzene (C3-BENZ) are given in Table 2; here we applied $k_{OH,BENZ}$=1.22x10$^{-12}$, $k_{OH,TOL}$=5.63x10$^{-12}$, $k_{OH,XYL/EBENZ}$=(7-23)x10$^{-12}$, $k_{OH,C3-BENZ}$=(6-57)x10$^{-12}$, $k_{OH,CO}$=1.5x10$^{-13}$ (from IUPAC, 2005), $k_{OH,BuOH-D9}$=3.4x10$^{-12}$ (from Barmet et al., 2012) cm$^3$ molec$^{-1}$ s$^{-1}$ and used a concentration average of expt A1 of $c_{BENZ}$=4x10$^{11}$, $c_{TOL}$=1x10$^{12}$, $c_{XYL/EBENZ}$=8x10$^{11}$, $c_{C3-BENZ}$=2x10$^{11}$, $c_{CO}$=(3-7)x10$^{14}$, $c_{BuOH-}$

$_{D9}$=(3.7-7.4) x$10^{11}$ in molec cm$^{-3}$ as input. This results in an OHR$_{ext}$ of 70-100 s$^{-1}$. Based on these input parameters, the model from (Li et al., 2015) and (Peng et al., 2016;Peng et al., 2015) predicted an [OH]$_{exposure}$ (OH concentration integrated over time, see discussion in main text "OH exposure estimation", in molec cm$^{-3}$ s) in the OFR as follows:

UV100%:          [OH]$_{exposure}$=(10-13)x$10^{11}$

5    UV70%:          [OH]$_{exposure}$=(2.4-3.1)x$10^{11}$

UV50%:          [OH]$_{exposure}$=(0.35-0.48)x$10^{11}$.

The estimated [OH]$_{exposure}$ (in molec cm$^{-3}$ s) and OH concentration (in molec cm$^{-3}$), [OH], based on the experimental measurements of the decay of BuOH-D9 correspond instead to

10    UV100%:          [OH]$_{exposure}$ =(3.0-5.8)x$10^{11}$, i.e. [OH]= (2.7-5.2)x$10^9$

UV70%:          [OH]$_{exposure}$ =(1.6-2.5)x$10^{11}$, i.e. [OH]=(1.4-2.2)x$10^9$

UV50%:          [OH]$_{exposure}$ =(0.31-0.49)x$10^{11}$, i.e. [OH]=(0.28-0.44)x$10^9$

The ratio of OH (measured) to O$_3$ (measured) remained relatively constant at our test points (OH/O$_3$ at 100%: (1.4-2.6)x$10^{-5}$,
15    (1.9-3.0)x$10^{-5}$ at 70%, (1.7-2.6)x$10^{-5}$ at 50%). The corresponding OH information derived from measurements in the SC was an [OH]$_{exposure}$ of 1.4x$10^{11}$ molec cm$^{-3}$ s at the maximum aging time (after around 2 hours), at a constant [OH]= 2x$10^7$ molec cm$^{-3}$.

Non-OH loss analysis (Figure S11Figure S10) predicted losses of aromatic hydrocarbons as SOA precursors between 10 and
20    25% by UV185 nm and UV254 nm, but no impact of O$_3$, (neither O($^1$D) or O($^3$P)) for the OFR-from-SC conditions. This only refers to the reactive interaction of OH vs. the excitation by UV, and does not allow conclusions on the formation of SOA. Also chemistry initiated by UV185 or UV254 may lead to the formation of SOA, and likewise photons may also destruct OH-formed SOA; both processes deserve attention in future research. Additionally, it does not allow 
[revised manuscript text omitted]
 43). Alkyl-substituted monocylic aromatics might hence (together with long-chain aliphatic compounds which might also substantially fragment) be significant contributors to the missing carbon mass (on average 35%), based on a comparison of FID-based and PTR-ToF-MS based measurements.

**SI Figures**

[Figure]

**Figure S1. Pictures** of **a)** original "muffler" and GPF in comparison, **b)** retrofitted GPF, installed underfloor in replacement to "muffler".

[Figure]

**Figure S2. Speed profile of regulatory driving tests.** Speed profile (v, in km h$^{-1}$) versus test time (in seconds) of EDC (new European driving cycle, top) and WLTC (world-wide light duty test cycle, class-3, bottom). While the EDC is characterized by two phase (an urban, and an extra-urban phase of highly repetitive characteristics) and lasts 20 min, the WLTC (class-3) is characterized by four phases at
10  different speed levels (referred to as Phase (Ph) 1-4, or low, medium, high, and extra-high speed, respectively); it contains patterns of disruptive acceleration and deceleration, and lasts 30 min. The WLTC is believed to represent typical driving conditions around the world and was developed based on combination of collected in-use data and suitable weighting factors by an expert group from China, EU, India, Japan, South Korea, Switzerland and the USA.

[Figure]

**Figure S3. OFR schematic (not to scale).** The OFR version deployed here was previously described in Bruns et al., 2015. The reactor is a 0.015 m³, cylindrical glass chamber (0.46 m L, 0.22 m diameter) flanked by two UV lamps on the upper part of the reactor, each with discrete emission lines at 185 and 254 nm (BHK Inc.). The lamps are cooled by a constant flow of air, or $N_2$. The incoming reactant flow is radially dispersed in the OFR by passing through a perforated mesh screen at the inlet flange. The flow through the OFR is determined by the flow pulled by instruments and pumps behind the reactor. The reactor is equipped with an injection system for water vapor ($H_2O$) and NMOCs (notably BuOH-D9, and selected precursor for single molecule testing). Water vapor is provided via a Nafion humidifier. Air is passing on one side of the Nafion membrane, collecting water vapor from the liquid on the other side of the membrane. In addition, other chemicals, such as BuOH-D9 (used as an OH tracer) can be injected by passing a small stream of clean air through a vial containing the liquid NMOC.

[Figure]

**Figure S4. Ammonium nitrate ($NH_4NO_3$) interference on $CO_2^+$ (Pieber et al., 2016).** The $CO_2^+$ signal (RIE=1) vs the $NO_3$ signal (RIE=1) from pure ammonium nitrate ($NH_4NO_3$) aerosol with $d_m$=400 nm from 3 calibration experiments. An orthogonal distance least squares fit yields a slope of $b$=0.035. Corrections were applied via the fragmentation table as noted in the main text.

[Figure]

**Figure S5. Reacted NMOC fraction in the SC (at t=2h after UV on), and the OFR at 100, 70 and 50% UV intensity (8 dominant ArHC).** A-D identifiers refer to individual experiments (GDI 1 only). The final OH exposure in the SC compares to an OH exposure of the OFR at 50-70% UV setting.

[Figure]

**Figure S6. Time-resolved aging of emissions (WLTC) (GDI1, standard configuration, Expt A1).** Cold and hot started WLTC of vehicle GDI1 (standard configuration). $CO_2$. CO, $CH_4$ (as measured by CRDS), THC and $CH_4$ (as measured by FID, note that the THC signal reaches its range limit at 20 ppm) are presented, together with organic aerosol (primary (denoted POA) and total (POA+SOA), denoted as OA. "OA profile during WLTC" highlights the measurement during the driving cycle, whereas OA shows the extended signal taking into account a delay due to the OFR residence time. Secondary nitrate aerosol (inorganic, ammonium nitrate, displayed is only $NO_3$), and primary equivalent black carbon (eBC). Note: data in these graphs are not normalized to $CO_2$, and have slightly different dilution ratios between cold- and hot-started cycle, as indicated by the $CO_2$ time-trace. Data reflect measured concentrations; no dilution corrections are applied. CRDS was diluted by a factor of 10 compared to FID and particle phase measurements.

[Figure]

**Figure S7. Time-resolved aging of emissions (WLTC) (GDI1, standard configuration, Expt A2, extended version of main text Figure 3).** See Figure S6Figure S6 caption for further details.

[Figure]

**Figure S8. Time-resolved aging of emissions (WLTC) (GDI1 w/ GPF, Expt B1).** See Figure S6 caption for further details.

[Figure]

**Figure S9. Time-resolved aging of emissions (WLTC) (GDI1 w/ GPF, Expt B2).** See Figure S6 caption for further details.

[Figure]

**Figure S10. Particle size distributions for experiments from (a) WLTC and (b) EDC, measured behind the OFR-from-SC.** All OFR-from-SC tests leading to typically 200 µg m⁻³ (~100-500 µg m⁻³) SOA formed at 100%, down to ~50 µg m⁻³ for 50% UV conditions. Expt A-D are identifiers for experiments referring to Table S4.

[Figure]

**Figure S11. OFR-from-SC and Online OFR 2015: non-OH loss estimation (OFR model by Peng et al., 2016; settings: "OFR185 Option 2").** Results are presented for OFR-from-SC Expts at 100% UV intensity, i.e. [OH]= 2.7-5.2 $10^9$ molec $cm^{-3}$. (a) $O_3$, (b) 185 nm, (c) 254 nm. Input parameters to "2016-10-12_OFR_Exposures_Estimator_v2.3": $OHR_{ext}$=100 $s^{-1}$, [$O_3$]=1.97 x $10^{14}$ molec $cm^{-3}$ (at 100%), [$O_3$]=0.74 x $10^{14}$ molec $cm^{-3}$ (at 70%), [$O_3$]=0.17 x $10^{14}$ molec $cm^{-3}$ (at 50%), water mixing ratio = 0.01 (1% absolute humidity).

[Figure]

**Figure S12. Online OFR 2014: Non-OH loss estimation (OFR model by Peng et al., 2016; settings: "OFR185 Option 2").** Time-resolved OFR Expts at 100% UV intensity (GDI1, 1 ejector dilution). (a) $O_3$, (b) 185 nm, (c) 254 nm. Input parameters to "2016-10-12_OFR_Exposures_Estimator_v2.3": OHRext=1000 $s^{-1}$, [$O_3$]=1.97x10$^{14}$ molec cm$^{-3}$, water mixing ratio=0.005 (0.5% absolute humidity), residence time=100 s; model-predicted OH-exposure=(5.9)x10$^{10}$ molec cm$^{-3}$ s.

[Figure]

**Figure S13. Effective SOA yields from SC experiments with different assumptions of absorptive mass. (a)** Yields as a function of suspended OA concentration, and **(b)** as a function of the sum of OA, HR-ToF-AMS derived ammonia ($NH_4$) and nitrate ($NO_3$), assuming that $NH_4NO_3$ acts as additional absorptive mass. Identifiers (A1-A3, B1-B3) allow retrieving the SC experimental conditions for each experiment from Table S4-S7.

[Figure]

**Figure S14. Time-resolved SOA from GDI4 in standard configuration and equipped with a prototype, catalytically active GPF.** SOA was generated by exposure of emissions to photochemistry in the OFR during cold-started WLTC test bench experiments.

[Figure]

**Figure S15. Propene fragmentation ratio in the PTR-ToF-MS.** Measurements were conducted at a concentration of around 0-150 ppbv propene ($C_3H_6$), as measured by the FID instrument.

[Figure]

**Figure S16. POA and eBC measurements in the SC batch sample compared to gravimetric PM measurements from the CVS** (a zoomed-in version is embedded in the figure).

**3 SI Tables**

**Table S1. Vehicle specifications.**

| Parameters | GDI1 | GDI2 | GDI3 | GDI4 |
|---|---|---|---|---|
| Vehicle Type | Opel Insignia 1.6 EcoFlex | Opel Zafira Tourer | VW Golf Plus | Volvo V60 T4F |
| Engine code | A16XHT | A16XHT | CAV | B4164T2 |
| Cylinder (number/ arrangement) | 4 / in line | 4 / in line | 4 / in line | 4 / in line |
| Displacement,cm3 | 1598 | 1598 | 1390 | 1596 |
| Power, kW | 125 @ 6000 rpm | 125 @ 6000 rpm | 118 @ 5800 rpm | 132 @ 5700 rpm |
| Torque, Nm | 260 @ 1650-3200 rpm | 260 @ 1650 - 3200 rpm | 240 @ 1500 rpm | 240 @ 1600 rpm |
| Injection type | DI | DI | DI | DI |
| Curb weight, kg | 1701 | 1678 | 1348 - 1362 | 1554 |
| Gross vehicle weight, kg | 2120 | 2360 | 1960 - 1980 | 2110 |
| Drive wheel | Front- wheel drive | Front- wheel drive | Front- wheel drive | Front- wheel drive |
| Gearbox | m6 | m6 | m6 | a6 |
| First registration | 2014 | 22.07.2014 | 01.02.2010 | 27.01.2012 |
| Exhaust | EURO 5b+ | EURO 5b+ | EURO 4 | EURO 5a |
| VIN | YV1FW075BC1043598 | WOLPD9EZ0E2096446 | WVWZZZ1KZ9W844855 | YV1FW075BC1043598 |

**Table S2. Gas-phase instrumentation.**

| Gas phase Instruments | Measured Parameter | Manufacturer | Lower limit (or range) |
|---|---|---|---|
| Picarro Cavity Ring-Down Spectrometer G2401 | $CO_2 + CO + CH_4 + H_2O$ | Picarro | 0-1000 ppmC ($CO_2$) 0-5 ppmC (CO) 0-20 ppmC ($CH_4$) 0-7% ($H_2O$) |
| THC Monitor APHA-370 | Total Hydrocarbon (THC), Non-methane hydrocarbon (NMHC) | Horiba | 0.02-100 ppmC |
| Proton-Transfer-Reaction- Time-of-Flight-Mass Spectrometer (PTR-ToF-8000) | Volatile organic compounds (VOC) | Ionicon Analytik | 10 ppt |

**Table S3. Particle-phase instrumentation.**

[revised manuscript text omitted]

| Compound | OA
$\mu g\ m^{-3}$ | OA_err
$\mu g\ m^{-3}$ | Ye
$\mu g\ ug^{-1}$ | Ye_err
$\mu g\ ug^{-1}$ |
|---|---|---|---|---|
| **TOL** | | | | |
| TOL | 26 | 4 | 0.15 | 0.02 |
| TOL | 50 | 8 | 0.18 | 0.03 |
| TOL | 66 | 10 | 0.21 | 0.03 |
| TOL | 69 | 10 | 0.19 | 0.03 |
| TOL | 70 | 11 | 0.16 | 0.02 |
| TOL | 106 | 16 | 0.23 | 0.03 |
| TOL | 117 | 18 | 0.21 | 0.03 |
| TOL | 291 | 44 | 0.29 | 0.04 |
| TOL | 795 | 119 | 0.35 | 0.05 |
| **OXYL/TOL (3:1)** | | | | |
| OXYL/TOL (3:1) | 347 | 52 | 0.64 | 0.10 |
| OXYL/TOL (3:1) | 507 | 76 | 0.46 | 0.07 |
| OXYL/TOL (3:1) | 588 | 88 | 0.53 | 0.08 |
| OXYL/TOL (3:1) | 852 | 128 | 0.76 | 0.11 |
| **OXYL/TOL (10:1)** | | | | |
| OXYL/TOL (10:1) | 26 | 4 | 0.14 | 0.02 |
| OXYL/TOL (10:1) | 82 | 12 | 0.34 | 0.05 |
| OXYL/TOL (10:1) | 104 | 16 | 0.26 | 0.04 |
| OXYL/TOL (10:1) | 176 | 26 | 0.27 | 0.04 |
| OXYL/TOL (10:1) | 266 | 40 | 0.45 | 0.07 |
| **TMB/TOL (2:1)** | | | | |
| TMB/TOL (2:1) | 141 | 21 | 0.36 | 0.05 |
| TMB/TOL (2:1) | 192 | 29 | 0.29 | 0.04 |
| TMB/TOL (2:1) | 195 | 29 | 0.37 | 0.06 |
| **TMB/TOL (20:1)** | | | | |
| TMB/TOL (20:1) | 675 | 101 | 0.45 | 0.07 |

**References**

[revised manuscript text omitted]